# Variability of glacier albedo and links to annual mass balance for the Gardens of Eden and Allah, Southern Alps, New Zealand

Angus J. Dowson[1,2], Pascal Sirguey[2], Nicolas J. Cullen[1]

[1]School of Geography, University of Otago, Dunedin, 9016, New Zealand
[2]National School of Surveying, University of Otago, Dunedin, 9016, New Zealand

*Correspondence to*: Pascal Sirguey (pascal.sirguey@otago.ac.nz)

**Abstract.** The Gardens of Eden and Allah (GoEA) are two of New Zealand's largest icefields. However, their remote location and protected conservation status has limited access and complicated monitoring and research efforts. To improve our understanding of the spatial and temporal changes in mass balance of these unique icefields, observations from the Moderate Resolution Imaging Spectroradiometer (MODIS) sensors are used to monitor annual minimum glacier-wide albedo ($\bar{\alpha}_{\mathrm{yr}}^{\min}$) over the period 2000-2018. The $\bar{\alpha}_{\mathrm{yr}}^{\min}$ for 12 individual glaciers ranges between 0.42 and 0.70 and can occur as early as mid-January and as late as the end of April. The evolution of the timing of $\bar{\alpha}_{\mathrm{yr}}^{\min}$ indicates a shift to later in the summer over the 19-year period on 10 of the 12 glaciers. However, there is only a weak relationship between the delay in timing and the magnitude of $\bar{\alpha}_{\mathrm{yr}}^{\min}$, which implies that albedo is not necessarily lower if it is delayed. The largest negative departures in $\bar{\alpha}_{\mathrm{yr}}^{\min}$ (lower than average albedo) are consistent with high snow line altitudes (SLA) relative to the long-term average as determined from the End-of-Summer Snowline (EOSS) survey, which has been the benchmark for monitoring glaciers in the Southern Alps for over 40 years. While the record of $\bar{\alpha}_{\mathrm{yr}}^{\min}$ for Vertebrae Col 25, an index glacier of the EOSS survey and one of the GoEA glaciers, explains less than half of the variability observed in the corresponding EOSS SLA ($R^2 = 0.43$, $p = 0.003$), the relationship is stronger when compared to other GoEA glaciers. Angel Glacier has the strongest relationship with EOSS observations at Vertebrae Col 25, accounting for 69% of its variance ($p < 0.001$). A key advantage in using the $\bar{\alpha}_{\mathrm{yr}}^{\min}$ approach is that it enables variability in the response of individual glaciers to be explored, revealing that topographic setting plays a key role in addition to the regional climate signal. The observed variability of individual glacier response at the scale of the GoEA contrasts with the high consistency of responses found by the EOSS record across the Southern Alps, and challenges the suggestion that New Zealand glaciers behave as a "unified climatic unit". MODIS imagery acquired from the Terra and Aqua platforms also provide insights about the spatial and temporal variability of clouds. The frequency of clouds in pixels west of the Main Divide is as high as 90% during summer months, and reach a minimum of 35% in some locations in winter. These complex cloud interactions deserve further attention as they are likely a contributing factor in controlling the spatial and temporal variability of glacier response observed in the GoEA.

**Keywords**. MODIS, albedo, glacier mass balance, remote sensing, New Zealand.

# 1 Introduction

The retreat of mountain glaciers during periods of the 20th and 21st centuries has been widespread and unprecedented (Zemp et al., 2015, 2019). One approach to document this change is to monitor glaciological mass balance, which enables the response of glaciers to climate forcing to be assessed (Hock et al., 2019; Zemp et al., 2019). However, global records of mass balance are sparse as traditional *in situ* glaciological measurements are resource intensive (Cogley et al., 2011), with only 260 glacier-wide results available from an estimated 200,000 glaciers worldwide (Pfeffer et al., 2014; Zemp et al., 2015). In addition, the ongoing cost of maintaining the repetitive measurements means that few long-term records exist, with only 30 of the world's glaciers having an uninterrupted series dating back to 1976 (Zemp et al., 2009). Mass balance records are therefore biased towards the more accessible glaciers in the Northern Hemisphere, while remote and relatively inaccessible glaciers in South America and New Zealand have remained largely out of reach. In this context, there is a need to monitor the state of more mountain glaciers to capture their response to changes in regional and large-scale climate.

Recent developments in the capabilities of satellite and airborne sensors, and improved processing techniques, have provided several alternatives to the *in situ* glaciological approach to derive individual or regional mass balance signals for remote glaciers (Rabatel et al., 2017). For example, mapping the end-of-summer snowline altitude (SLA) provides an estimate of the glacier equilibrium-line altitude (ELA) and accumulation area ratio (AAR). These glacier properties have been used as proxies for annual mass balance and are often less resource intensive to monitor (Chinn et al, 2012; Rabatel et al., 2016; Salinger et al., 2019b). Alternatively, the 'albedo method' uses glacier surface albedo, for example retrieved from satellite imagery captured by the Moderate Resolution Imaging Spectroradiometer (MODIS), to estimate glacier mass balance and/or monitor its variations (Rabatel et al., 2017). Surface albedo plays a governing role in the mass balance of glaciers due to its control on absorbed short-wave radiation (Klok and Oerlemans, 2004; Oerlemans et al., 2009; Pope et al., 2016). The annual minimum glacier-wide albedo ($\bar{\alpha}_{yr}^{min}$) retrieved from MODIS imagery has been found to scale to annual glacier mass balance in the French Alps (Dumont et al., 2012; Davaze et al., 2018), the Arctic (Greuell et al., 2007), the Himalayas (Brun et al., 2015; Zhang et al., 2018) and New Zealand's Southern Alps on Brewster and Park Pass glaciers (Sirguey et al., 2016; Rabatel et al., 2017).

The mountain glaciers of the Southern Alps of New Zealand comprise the largest ice mass in the Southern Hemisphere outside of Antarctica and South America, and are regarded as globally significant (Chinn et al., 2012; Zemp et al., 2015). The Southern Alps are unique as they are surrounded by vast areas of ocean and are subject to both subtropical and polar air masses that are embedded in a prevailing westerly airflow. This unique setting contributes to the humid, maritime climate that influences the glaciers in the Southern Alps, which is reflected by the exceptionally high precipitation rates exceeding 10 m a$^{-1}$ in many alpine regions (Fitzharris et al., 1999). Meteorological experiments on glaciers in the Southern Alps have shown the main source for melt energy is net radiation, driven primarily by net shortwave radiation (e.g. Gillett and Cullen, 2011; Cullen and Conway, 2015), but fluctuations in mass balance over time are sensitive to changes in air temperature and precipitation. Given the importance of air temperature in controlling both melt and the phase of precipitation, the advance and retreat of glaciers in the wet climate of the Southern Alps are particularly sensitive to changes in air temperature (Oerlemans, 1997; Mackintosh et al., 2017). For example, the advance of some fast-responding glaciers in the Southern Alps between 1983 and 2008, during three of the warmest decades in the instrumental record, have been attributed to regional cooling controlled by changes in large-scale atmospheric circulation in the Southern Hemisphere (Mackintosh et al., 2017). The influence of synoptic scale circulation on air mass variability and changes in cloud properties have also been shown to influence mass balance in the Southern Alps (Conway and Cullen, 2016; Cullen et al., 2019).

The first comprehensive glaciological mass balance study in New Zealand's Southern Alps was completed on Ivory Glacier between 1969 and 1975 (Anderton and Chinn, 1973; 1978). Today, comprehensive glaciological mass balance programmes exist on Brewster Glacier (e.g. Cullen et al., 2017) and Rolleston Glacier (Purdie et al., 2015), extending back to 2004 and

2010, respectively. The size of Brewster Glacier (ca. 2 km$^2$) and its comprehensive in-situ record enabled the albedo method with MODIS to be assessed and calibrated to estimate mass balance (Sirguey et al., 2016). With only 0.11 km$^2$ in 2012, Rolleston Glacier is however too small to be captured by MODIS imagery. In addition, the End-of-Summer Snowline (EOSS) survey, which has been the benchmark for monitoring glaciers in the Southern Alps for over 40 years, has used aerial photography to map and retrieve the altitude of the annual snowline at the end of the ablation season (SLA) of 50 'index' glaciers across the Southern Alps since 1977 (Chinn et al., 2012). The SLA is used as a surrogate of the Equilibrium Line Altitude (ELA) whose annual variations inform about changes in the annual balance of the glaciers (Rabatel et al., 2017). Despite is longevity, the EOSS survey continues to face two main challenges: (1) the temporal resolution is limited to a single observation per year in late-summer; the success of which is dependent on transient conditions such as snowfall prior to a flight, and (2) limited resources constrain the total number of index glaciers that can be mapped. Furthermore, the focus on relatively small glaciers in the Southern Alps has meant that only three index glaciers exceed 2 km$^2$. These challenges, combined with the difficulty of establishing a network of glaciological programmes, have resulted in the vast majority of glaciers in New Zealand remaining out of reach of current observation methods.

The EOSS programme indicates that the yearly response of the transient snowline for each of the index glaciers is consistent with the average of the group, with strong to very strong intra-correlations between SLA departures across index glaciers. This consistent response to climate variability inferred from the EOSS record has led to the suggestion that glaciers in the Southern Alps behave as an "unified climatic unit" (Chinn et al., 2012, p. 115). The linear models corresponding to this hypothesis have led to observations from single glaciers, such as Tasman or Brewster Glacier, being used to estimate fluctuations in ice volume for the entire Southern Alps (Salinger et al., 2019a; 2019b). To further explore the validity of this approach, more long-term continuous signals of glacier mass balance are required across a broader topographical range of glaciers, particularly those larger than 2 km$^2$. Given the available approaches to monitor glaciers and the lack of *in situ* data, the albedo method is a desirable alternative to objectively resolve mass balance variability and trends on large, un-monitored New Zealand glaciers.

The Gardens of Eden and Allah (GoEA), located on the Main Divide of the Southern Alps, northeast of Aoraki/Mt Cook National Park, comprise two of the largest icefields in New Zealand (Fig. 1). They form an interesting target to test the climatic unit hypothesis, as the icefields consist of a network of smaller, interconnected glaciers, placed in a critical climatic zone straddling the Main Divide. Their position on the Main Divide, combined with their protected conservation status has greatly constrained the possible methods for data collection and extraction. As a result, these glaciers are yet to be the target of focused research, and their behaviour over the past decades is largely undocumented and poorly understood.

To address this gap in knowledge, the aim of this paper is to characterise the spatial and temporal variability of glacier-wide surface albedo on the GoEA, and to investigate the linkages between this variability and glacier mass balance. To achieve this aim, this research has the following objectives: (1) to determine the timing and magnitude of glacier-wide surface albedo on 12 different outlet glaciers on the GoEA; (2) to compare the spatial and temporal changes in surface albedo to variability in SLA as determined from the EOSS programme; and (3) to characterise variability in cloud cover using MODIS imagery acquired by the Terra and Aqua platforms. Remote sensing with the albedo method provides an opportunity to examine the significant, yet remote and protected GoEA icefields, and provides an opportunity to further assess the skill of the albedo method to remotely monitor the physical processes governing glacier mass balance in the Southern Alps.

## 2 Site description and data

### 2.1 The Gardens of Eden and Allah

#### 2.1.1 Situation

The Garden of Eden and Garden of Allah (referred to collectively as the GoEA) are two of New Zealand's largest icefields (Fig. 1), situated in the Adams Range of the Southern Alps/Kā Tiritiri-o-te-Moana (43º18'S, 170º43'E). The icefields are aligned east to west, spanning across the Main Divide, 56 km northeast of Aoraki/Mt Cook. In this context, the GoEA are considered the perennial ice mass (ca. 36 km$^2$) of the region, bordered by Outram Peak (2399 m), Mt Barlow (2100 m), Mt Kensington (2444 m) and Mt Stoddart (2223 m). The glaciers extend from a maximum elevation of 2543 m a.s.l. (Newton Peak) to ca. 1100 m a.s.l. at the terminus of Colin Campbell Glacier (southeast of Newton Peak).

The icefields are centred on two large rock plateaus (ca. 1900 m a.s.l), with ice flowing down into broad valley glaciers or terminating in dramatic icefalls over the incised valley walls. These outlets provide meltwater to the catchments of the Rangitata River on the east coast, and the Whataroa and Wanganui on the west coast. The positioning of the GoEA across the Main Divide places the icefields in a complex climatic setting. A prevailing westerly airflow combines with strong orographic lifting to generate a large precipitation gradient from west to east across the Main Divide (Henderson and Thompson, 1999). The estimated mean annual rainfall in the area over the period 1972-2016 ranges from 6,000 to 8,500 mm yr$^{-1}$ (Macara, 2017). The mean annual temperature over the glaciated region for the period 1980-2010 is estimated to range from -1.0 to 2.5°C based on an interpolation of mean normal temperature from distant surrounding weather stations and a lapse rate determined from the data of -5.7°C km$^{-1}$. Given the contribution of snow and ice to New Zealand's water resources, changes in the volume of these icefields have the potential to impact the hydrology of these rivers under future climate change (Chinn, 2001).

Despite their significance, research on the GoEA is limited by the remote location of the glaciers, and direct access has been further complicated in recent years by their inclusion in the 466 km$^2$ Adams Wilderness Area (est. 11 February 2014). Helicopter landings are now largely prohibited, effectively ruling out a number of *in situ* monitoring methods. As a result, the only comprehensive data obtained from the area to date are EOSS SLA records for the two Vertebrae glaciers at the western margin of the GoEA (Willsman et al., 2018), which are described in more detail below (Section 2.3). The Garden of Eden was also briefly included in an attempt to detect changes in the ELA of New Zealand glaciers from 15 m-resolution ASTER satellite imagery (Mathieu et al., 2009). The absence of data and the notable extent and volume of ice locked in the GoEA make the region a compelling target for research among the more than 2500 glaciers in the Southern Alps.

#### 2.1.2 Glacier outlines and surface topography

As part of this research, the boundaries of the icefield outlets were redefined, updating the existing New Zealand Glacier Inventory outlines derived during the 1970s (Chinn, 1991), and refining the 2017 Randolph Glacier Inventory (RGI) version 6.0 outlines (Pfeffer et al., 2014). We used a 10 m-resolution Sentinel-2A image captured on 14 March 2016, in late-summer and cloud-free conditions, as the base of our mapping. This image provided a suitable tradeoff between glacier exposure and illumination to define glacier outlines. Perennial snow and ice in the GoEA were initially segmented using the Normalised Difference Snow Index (NDSI) band ratio, derived from bands 3 (Green) and 11 (Shortwave Infrared) of Sentinel-2, with a threshold of 0.8. Despite strong shadowing, a later Sentinel-2 image on 30 April 2016, which appears close to the maximum ablation, provided a visual reference to refine glacier outlines and exclude transient snow patches. Manual edits were made to refine the classification, specifically to correct shaded or debris-covered areas, where detection of snow and ice is impaired. These edits were aided by the interpretation of 0.5 m-resolution orthorectified pan-sharpened Pléiades-1B satellite image acquired on 10 March 2017 as part of the Pléiades Glacier Observatory programme (PGO), and a range of oblique aerial and terrestrial photographs captured during fieldwork on 27/28 January 2018.

The outlet glaciers were then delineated from the wider icefields using topographic divides identified from 20 m elevation contour vectors surveyed by Land Information New Zealand (LINZ), with refinements based on the interpretation of a preliminary high-resolution surface model derived from the Pleiades stereo acquisition. A total of 17 outlet glaciers were identified in the GoEA. However, this was reduced to 12, as: (1) some small adjacent glaciers ($<$ ca. 0.5 km$^2$) were amalgamated into larger units (if the direction of flow was consistent) to create a more suitable target for the MODIS analysis (i.e. O'Neil, Serpent and Cain glaciers), or (2) discrete glaciers $<$ ca. 0.5 km$^2$ were excluded from further analysis (i.e. Vertebrae Col 12 and Wee McGregor Glacier). Elevation, slope and aspect data for these 12 outlet glaciers were then derived from the national 15 m-resolution digital elevation model (DEM, NZSoSDEM v1.0; Columbus et al., 2011).

## 2.2 MODIS products

This study relies primarily on a record of glacier surface albedo retrieved from a time-series of MODIS images. MODIS is one of the key sensors aboard the Terra (EOS/AM-1) and Aqua (EOS/PM-1) satellite platforms, hereafter referred to as MODIS[T] and MODIS[A], respectively. Terra was launched on 18 December 1999 as the flagship of NASA's Earth Observing System (EOS), allowing MODIS[T] to capture near daily, moderate-resolution images of Earth's surface since 25 February 2000. Aqua was launched in May 2002, creating a second daily opportunity to capture MODIS[A] images. Terra descending node crosses the equator at approximately 10:30 am, while Aqua ascending node is crossing at 1:30 pm. Historically, MODIS[T] imagery has been preferred for the albedo method due to the longer record of imagery, as well as failed detectors with MODIS[A] Band 6.

Following Sirguey et al. (2009) and Dumont et al. (2012), the albedo retrieval was completed with the MODImLab software as described in Section 3.1. As yet, the use of MODISA imagery to supplement the MODIST record for snow and ice albedo retrieval has not been explored. Wardle (1986) demonstrated the extraordinary cloudy sky conditions that dominate New Zealand's Southern Alps, greatly reducing the data available to space-borne remote sensors. Wardle (1986) specifically reported that the western flanks of the study area near Cropp River exhibited the highest frequency of days with some clouds (92%), as well as days with heavy clouds (72%). The frequency of cloudy days tends to decrease towards the east, with some or heavy clouds occurring on 72% and 38% of days, respectively. The consideration of MODISA data in this study provides an opportunity to compare surface albedo derived from each sensor, and to inform on key climate variables such as cloud frequency and variability over the Southern Alps. If the albedo derived from MODIS[A] is suitable, despite the degraded Band 6, it can be used to increase the temporal resolution of the albedo method to two observations per day, or to fill gaps in the MODIST record.

We use MODIS Level-1B Collection 6 (C6) swath data products that contain calibrated and geolocated top-of-atmosphere (TOA) radiance counts for all 36 spectral bands. The products include bands 1 and 2 supplied at 250 m nadir resolution (MxD02QKM files, x=O/Y for Terra/Aqua, respectively). Bands 3 to 7 are provided at 500 m spatial resolution (MxD02HKM files), along with an aggregated 500 m version of bands 1 and 2. All 36 spectral bands are provided at 1000 m resolution (MxD021KM files), including the aggregated bands 1-7 at 1000 m. The accompanying MxD03 file contains the geolocation information and relative geometry parameters such as solar zenith ($\theta_s$), solar azimuth ($\varphi_s$), sensor zenith ($\theta_v$) and sensor azimuth ($\varphi_v$), required to take into account sun-target-sensor geometry in the albedo retrieval (Dumont et al., 2012; Sirguey et al., 2016).

The MODIS[T] record over the GoEA consists of 7067 granules between 25 February 2000 and 30 April 2018 captured on 6416 unique days. In addition, MODIS images were selected to compare the surface albedo products from MODIST and MODISA over the GoEA. Image pairs needed to be captured on the same day, under clear sky conditions, and with similar near-nadir sensor zenith to mitigate the effect of different panoramic distortion on pixel footprint between MODIS[T] and MODIS[A]. Four suitable image pairs across separate years near the end of summer were identified as suitable to assess the consistency of albedo retrieval between both sensors, namely 3 March 2009, 9 March 2010, 8 March 2012 and 10 March 2013. At 250 m resolution, this provided a sufficient sample size of 2196 pairs of albedo from matching pixels that were compared with linear regression

and the coefficient of determination ($R^2$). Finally, a full year of MODIS[A] Level-1B images was downloaded between 1 January

and 31 December 2012 (448 granules) to assess data loss compared to the MODIS[T] record.

**2.3 EOSS survey programme**

The national EOSS programme is run by National Institute of Water and Atmospheric Research (NIWA). Aerial photos are captured on the first suitable weather window following 1 March. The mapping of SLA for each of the index glaciers implements three methods summarised hereafter (see Willsman, 2018). (1) When the snowline is clearly visible, it is sketched

from the oblique photograph onto 1:50,000 topographic contour maps of each glacier, or digitized readily from rectified photos to map the accumulation or ablation zones. The snowline elevation is derived from the ablation area and the hypsometric curve of the glacier. (2) The SLA is often determined by applying an "interpolation method" whereby all EOSS photographs of a glacier are arranged into a sequence of increasing area of snow cover (descending order of SLAs); the SLA value is interpolated form that of the adjacent years. (3) When the snowline is obscured, it is interpolated from an interpretation of the degree of

snow cover surrounding the glacier compared to previous years on record.

For each year $i$, $SLA_i$ records delivered by the EOSS programme for the two Vertebrae glaciers in the GoEA provide an independent dataset to make a comparison with the retrieved albedo signal. Vertebrae Col 25 is a 0.7 km$^2$ mountain glacier facing southwest at the western tip of the Garden of Eden (Fig. 1). Vertebrae Col 12 is a smaller cirque glacier (0.21 km$^2$) located directly adjacent, to the north. The SLA was first recorded for the Vertebrae glaciers in 1978, despite the EOSS survey

beginning in 1977, and has been recorded every year since, with the exception of 1979 (no flight), 1984 (no visit), 1987 (no visit), 1990 (no flight) and 1991 (no flight). Of the 35 observations in the 40-year period since 1978, the SLA was digitised eleven times (method 1) and interpolated (method 2 and 3) all other times (Fig. 2).

As part of the EOSS programme methodology (Chinn et al, 2012; Willsman et al., 2018), the $SLA_i$ for each glacier is compared against the long-term or 'steady-state' SLA, termed 'ELA$_0$', which as determined by the EOSS survey is 1864 m a.s.l. and

1840 m a.s.l. for the Vertebrae Col 12 and 25 glaciers, respectively. Vertebrae Col 12 is not considered in this study, as it is not of sufficient size for albedo retrieval with MODIS, however the annual SLA departures from ELA$_0$ for the two glaciers are strongly correlated ($R^2 = 0.95$). The $SLA_i$ departures for Vertebrae Col 25 show eight years with particularly high seasonal snowlines since 1999, indicative of a negative glacier mass balance (in particular 1999, 2008, 2011, 2012 and 2016), preceded by a majority of years with snowlines located near or below ELA$_0$, indicative of a positive mass balance.

Both SLA departure records from Vertebrae Col 12 and 25 correlate strongly to the SLA departures averaged over the remaining index glaciers photographed in a particular year (EOSS$_{Alps}$; $R^2 = 0.86$ and 0.92, respectively, see Willsman et al., 2018). These very strong correlations between individual glacier responses to EOSS$_{Alps}$ suggest that the Vertebrae Col glaciers, like other index glaciers in the EOSS programme, respond uniformly to climate variability. This high degree of intra-correlation in the EOSS record has led to the theory that glaciers in the Southern Alps behave as a "unified climatic unit"

(Chinn et al., 2012). However, we hypothesise that changes in the climate system are likely to result in greater variability in glacier response in the Southern Alps, which may not be resolved and/or detected by the EOSS programme. The ability to derive and discriminate a mass balance signal from a large number of glaciers in the GoEA using the albedo method, which has a high temporal resolution, provides us with a new opportunity to further characterize the spatial variability of glacier behaviour in the GoEA, as well as test the consistency in glacier response predicted by the EOSS programme.

**2.4 Sentinel-2 data**

As EOSS $SLA_i$ records only exist on two of the smaller glaciers in the GoEA, high-resolution Sentinel-2 data are also used to support the MODIS analysis by observing the evolution of the summer snowline across the icefields. Launched on 23 June 2015, Sentinel-2 imagery are available for the 2016, 2017 and 2018 summers, and provide an independent means to assess the

consistency of the albedo products and EOSS survey results. For the three summer periods (1 January - 30 April), sequences
of cloud-free 10 m-resolution Sentinel-2A and 2B Level-1C images are used to observe the evolution of the GoEA surface
until seasonal snow ends the ablation season. There is an expectation that the discolouration of the glacier surface and elevation
of the snowline in the Sentinel-2 images depicting maximum ablation will compare and be consistent with relative changes
determined by the EOSS survey and minimum glacier-wide albedo (Sirguey et al., 2016). Qualitative results of these
observations are presented and discussed for Lambert Glacier, with the elevation of the snowline determined from LINZ 20 m
topographic contours.

## 3 Methods

### 3.1 Retrieving snow and ice albedo

Terra and Aqua MODIS C6 Level-1B products were processed using the MODImLab toolbox (Sirguey et al., 2009).
MODImLab has been widely employed to perform a series of image processing techniques on MODIS time-series data,
yielding snow and ice albedo products at 250 m-resolution (Dumont et al., 2011, 2012; Brun et al., 2015; Sirguey et al., 2016;
Rabatel et al., 2017; Davaze et al., 2018). The operations and processes of MODImLab are described by Sirguey et al. (2009)
and Dumont et al. (2012), and are only covered here in brief.

First, image fusion combines the higher resolution MxD02QKM bands 1 and 2 (250 m) with the lower resolution MxD02HKM
bands 3 to 7 (500 m) to produce seven spectral bands at 250 m spatial resolution (Sirguey et al., 2008). As a result, MODIS
bands used for mapping snow cover (band 4 at 555 nm and 6 at 1640 nm) have a higher spatial resolution than other common
MODIS products (e.g. MxD10 - 500 m). The atmospheric and topographic correction module (ATOPCOR) then corrects
images containing TOA radiance counts into values of ground surface reflectance (Sirguey, 2009). Topographic corrections
are calculated from the NZSoSDEM along with the MxD03 product to account for the relative illumination and viewing
geometry. The DEM is also used to pre-process the sky-view and terrain factor to account for diffuse and terrain-reflected
irradiance over rugged terrain, and the effects of both self and cast shadow.

The MODImLab algorithm has the ability to resolve the sub-pixel snow cover fraction in mountainous terrain (Sirguey et al.,
2009). Spectral unmixing estimates the relative contributions of specific land cover types to the radiometry of each individual
pixel (Masson et al., 2018). Values of spectral albedo (narrowband albedo) for each pixel measured from five MODIS bands
are compared against a Look-Up Table (LUT) generated with DISORT (Stamnes et al., 1988). The LUT contains an array of
spectral albedo values simulated for snow and ice surfaces with varying snow grain size, impurity type and content, and
incident zenith angle (Dumont et al., 2011). The best match of spectral albedo can then be integrated to yield the Blue-Sky
(BS) and White-Sky (WS) broadband albedo for a pixel. BS albedo is the value of broadband albedo corresponding to the
specific ground irradiance. However, diffuse, or WS albedo is preferred in this research, as it allows the surface albedo to be
studied with reduced sensitivity to seasonal changes in illumination conditions (Dumont et al., 2012). Given the small
reflectance of snow and ice targets at 1600 nm and marginal contribution to broadband albedo, the performance of albedo
retrieval from MODIS Aqua data is not expected to be compromised despite the degraded Aqua band 6.

### 3.2 Glacier masks

Having processed the MODIS granules with MODImLab, we then create 250-m raster glacier masks using the glacier outlines
from Section 2.1.2 under which albedo is averaged. Glacier masks are defined from glacier outlines to avoid debris-covered
areas or mixed land-cover pixels (e.g. containing a combination of snow/ice and rock) so as to preserve the integrity of the
snow and ice albedo retrieval. In previous studies, this has been achieved subjectively (e.g. Dumont et al., 2012; Brun et al.,

2015; Davaze et al., 2018), but takes time and introduces variability between users. As a result, we consider an alternative approach to masking that can be more objectively deployed across a large number of glaciers.

From all pixels within the glacier outlines, we use the sub-pixel snow classification produced by spectral unmixing in MODImLab to exclude those with snow-covered-area less than 50% snow, as this threshold is generally used to segment snow from snow-free pixels (Sirguey et al., 2009). This excludes most non-glaciated surfaces from the overall glacier outlines (e.g. debris-covered) and mitigates the effect of mixed-pixels in the glacier-wide albedo. We assess the effectiveness of this masking approach (Mask 1) against a conservative glacier mask created manually (Mask 2) by comparing the glacier-wide albedo $\bar{\alpha}(t)$ from each mask and outlet glacier over the full sequence of MODIS$^T$ imagery. The mean surface albedo series retrieved from each mask, $\bar{\alpha}(t)_{M1}$ and $\bar{\alpha}(t)_{M2}$, are compared using the linear coefficient of determination ($R^2$), root mean square difference (RMSD) and mean difference (MD).

### 3.3 Filtering the MODIS-albedo record

Despite the near-daily capture of MODIS images over the GoEA, cloud cover in the Southern Alps greatly reduces the quantity of available data. Following Sirguey et al. (2016), only MODIS images with no cloud present in any pixels within the glacier mask are retained in the analysis. Cloudy pixels were determined using MODImLab's cloud detection algorithm, based on the original MODIS MOD35 Cloud Product described by Ackerman et al. (1998). Using these products, Brun et al. (2015) and Davaze et al. (2018) demonstrated the successful application of a cloud threshold, whereby images are retained so long as a certain proportion of the glacier surface is cloud-free in the image (>20% and >30% clear surface, respectively). While there is merit in this approach for larger glaciers (e.g. Chhota Shigri Glacier: 15.7 km$^2$ and Mera Glacier: 5.1 km$^2$), the outlet glaciers of the GoEA identified in this analysis are much smaller (average ca. 2.8 km$^2$). On small glaciers, there is a higher probability that large parts of the accumulation or ablation areas will be obscured by clouds, and therefore the surface albedo across the glacier may be misrepresented. Relying on cloud-free conditions over the glacier excluded ca. 66% of the images, resulting in an average of one clear-sky image every 3 days. The classification of clouds in the complete 19-year long inventory of near-daily MODIS images was then leveraged to characterise the spatial variability in cloud frequency of occurrence around the GoEA.

An increase in sensor viewing zenith angle ($\theta_v$) distorts MODIS pixels due to the panoramic effect. The ground sampling distance of MODIS pixels increases from 2- to 5-fold from $\theta_v = 45°$ to $66°$, respectively (Wolfe et al., 1998). Albedo retrieval of mountain glaciers with MODIS is most accurate with $\theta_v < 30°$ (Sirguey et al., 2016). However, Brun et al. (2015) and Sirguey et al. (2016) accepted $\theta_v < 40°$ and $< 45°$, respectively, to increase the number of images available for retrieval. Davaze et al. (2018) confirmed that, while albedo retrieval was most accurate with view angle $\theta_v < 30°$, it performed well until $\theta_v < 45°$. In this study, and given the various configurations and size of glaciers in GoEA, we found that this threshold remained suitable to optimise the number of images and quality of the albedo retrieval.

### 3.4 Calculating glacier-wide albedo

Glacier-wide albedo $\bar{\alpha}(t)$ was then calculated for each MODIS image (at time $t$) as an average of all pixel values within each outlet glacier mask. Following Dumont et al. (2011), $\bar{\alpha}(t)$ was calculated using the WS albedo product, under the anisotropic reflectance model. To reduce the noise of the $\bar{\alpha}(t)$ signal and smooth the time-series, a three-period rolling average was applied (Sirguey et al., 2016), and was preferred to a fixed-day average, which is heavily influenced by large gaps in the series, driven by persistent cloud cover. The $\bar{\alpha}_{yr}^{min}$ was then extracted as the annual minimum glacier-wide value of smoothed $\bar{\alpha}(t)$, from the period corresponding to the end of the ablation season (1 January - 30 April). Due to complex topography and cast shadows in the GoEA, visual confirmation revealed that when $\bar{\alpha}_{yr}^{min}$ is reached outside of this period, it is always due to incorrect albedo retrieval (underestimated) due to inaccurate topographic correction in the shade, as identified by Davaze et al. (2018).

### 3.5 Characterising topographic shading

Shading at the glacier surfaces complicates the topographic correction and has the potential to create errors in the albedo retrieval algorithm. This typically occurs at low sun zenith angles, outside of the late-summer period when $\bar{\alpha}_{yr}^{min}$ is reached. Nonetheless, changes in the distribution of shortwave radiation being received at the glacier surface affects processes such as the evolution of surface albedo. Net shortwave radiation is one of the key components of the glacier surface energy balance (SEB). The intensity of solar radiation received at the surface is a function of the time of year (season) and global position (latitude). At a local scale, this relationship is complicated by topography, slope and aspect. Olson and Rupper (2018) show shading from topography (both cast- and self-shading) is a key factor contributing to the SEB. Therefore, in addition to quantifying glacier slope and aspect (Section 2.1.2), the ATOPCOR module from MODImLab is used to calculate fractional shading at 250 m resolution across the glacier surface at the time of image capture. We use the maximum proportion of surface shading (occurring during the winter solstice, when the solar zenith angle is at its maximum) as a simple metric to compare shading between glaciers. A high maximum percentage of topographic shading indicates a topographically confined glacier, while a low maximum percentage indicates an open, unconfined surface that receives radiation year round (e.g. Fig. 3).

## 4 Results

### 4.1 Assessing the mask performance

Between February 2000 and April 2018, the difference between derived glacier-wide albedo $\bar{\alpha}(t)_{M1}$ and $\bar{\alpha}(t)_{M2}$ is small (RMSD = 0.037; Table 1), with Mask 1 (objective masking approach) typically yielding smaller albedo compared to Mask 2 (MD = -0.012). The linear agreement between the masks is highest during the critical summer period (1 October-31 March), when $\bar{\alpha}_{yr}^{min}$ is expected to be reached. The increased difference during winter months is likely a result of debris-covered pixels (not considered by Mask 2) that become snow-covered beyond 50%, thus included in Mask 1 and raising $\bar{\alpha}(t)$. The relatively small difference between the two methods, particularly during summer, is confirmation of the suitability of the objective glacier masking approach, and is therefore used to produce the following results over the GoEA.

### 4.2 Glacier characteristics

Despite the two icefields being an interconnected ice mass, the glaciers of the GoEA exhibit large differences in their hypsometry (Table 2). The majority of the ice resides slightly above 1900 m a.s.l., close to the average elevation of the plateaus. As with many other glaciers in the Southern Alps, the average surface gradient is steep, with a number of glaciers approaching 20°. In addition, glacier size is relatively small, ranging between 0.44 km$^2$ and 4.44 km$^2$. Lambert Glacier is an exception (9.44 km$^2$), comprising over one quarter of the total surface area (33.89 km$^2$). The glaciers also occupy a range of aspects with variable topographic shading. As expected, glaciers with mean north-facing aspects display lower values of topographic shading than south-facing glaciers (e.g. Angel Glacier, east-northeast, 5.4%; Colin Campbell, south, 82%).

It is anticipated that the large topographic differences between outlet glaciers may drive a large variability in the temporal evolution of glacier surface albedo. To further characterise the contrasting topography of these glaciers, we perform a K-mean cluster analysis based on mean aspect, mean slope and maximum topographic shading derived from the mapped outlines of each glacier. The glacier characteristics (Table 2) when viewed in scatter plots revealed that the 12 outlet glaciers grouped in three identifiable clusters, with the contrasting hypsometry indicated in Fig. 4. Cluster membership for each glacier is provided in Table 2. Glaciers in Cluster 2 are characterised by southerly aspects, steep slopes and incised topography (indicated by topographic shading exceeding 80%). These glaciers contrast to the north- and east-facing, unconfined glaciers in Cluster 1. Cluster 3 shares similar topographical attributes to Cluster 1, although the aspect is primarily west-facing. Importantly, the

glaciers in Clusters 1 and 3 account for 81% of the total surface area of the GoEA , while the two south-facing glaciers in Cluster 2 occupy the remaining 19%.

### 4.3 Annual evolution of MODIS-derived glacier-wide albedo

All 12 glaciers in the GoEA exhibit a marked seasonal evolution of MODIS-derived glacier-wide albedo $\bar{\alpha}(t)$ when averaged within-cluster (Fig. 5a), characterised by a decrease in $\bar{\alpha}(t)$ at the end of the austral winter as controlled by the onset of melt.

The lowering of $\bar{\alpha}(t)$ is primarily due to the melting of snow from the previous winter, exposing the underlying glacier ice and firn. To compound this process, snow and ice that remain through the summer undergoes metamorphism. This process drives an increase in grain size, liquid water content and impurity concentration, which all act to decrease surface albedo. As a result, glacier-wide albedo continues to fall until fresh snow begins to accumulate in late summer and early autumn. The duration of exposure of discoloured glacier ice and firn across the glacier during summer is a critical part of the physical

processes controlling surface ablation. Although each of the three clusters follows a typical pattern of seasonal evolution, the timing and magnitude of key events, such as the $\bar{\alpha}_{yr}^{min}$ and the rate of change of $\bar{\alpha}(t)$, varies between glaciers.

On some glaciers, the seasonality of the albedo signal is complicated by a substantial decrease in $\bar{\alpha}(t)$ centred around the winter solstice (21/22 June). This trend is not unique to the GoEA, and has been identified to be an artefact of widespread surface shading when the sun is at its lowest that compromises the radiometric correction by MODImLab, resulting in incorrect

surface albedo (Rabatel et al., 2017; Davaze et al., 2018). This issue affects both glaciers in cluster 2 (observable in Fig. 5a), and some in cluster 3 (namely, Eve Icefall and Perth Glacier). These glaciers all exhibit a south/southwest aspect and topographic shading greater than 60% (Table 2). As this pitfall develops in winter, it has little or no impact on the summer albedo signal and the identification of $\bar{\alpha}_{yr}^{min}$. We therefore disregard much of the winter signal from these glaciers, and focus on the summer ablation period.

During the ablation period, all three classes share consistent patterns over the 19-year albedo record that deviate from a monotonic decrease. Short-lived increases in $\bar{\alpha}(t)$ occur in mid-October and December, suggesting synoptic-scale atmospheric processes result in late snowfall events, which temporarily change the albedo signal over the glaciers. A similar pattern of late spring or early summer snowfall events were also observed on Brewster Glacier between November and December (Sirguey et al., 2016), which have a strong and coherent weather-system scale signature (Cullen et al., 2019). The occurrence of these

snow events likely play a considerable role in the evolution of albedo through the summer period and in turn, glacier mass balance. Large events have the potential to deposit enough fresh snow at the surface to provide prolonged protection of the otherwise exposed glacier ice during the height of summer.

### 4.4 Annual minimum glacier-wide albedo ($\bar{\alpha}_{yr}^{min}$)

Figure 6 demonstrates the variability of the $\bar{\alpha}_{yr}^{min}$ observed over the 19-year period between individual glaciers. The average

magnitude of $\bar{\alpha}_{yr}^{min}$ varies between 0.50 (Arethusa Glacier) and 0.62 (Eve Icefall), but ranges between 0.42 and 0.70. $\bar{\alpha}_{yr}^{min}$ is typically reached between early-February (Abel Glacier: 37 Julian Days (JD)) and mid-March (Barlow Glacier: 75 JD), but can occur as early as mid-January and as late as the end of April. In addition to a large range in timing between glaciers, the variability in the arrival of $\bar{\alpha}_{yr}^{min}$ on certain glaciers is also large (e.g. $\sigma$ = 29.5 days for Barlow Glacier).

The Spearman's rank coefficient is used to determine the topographic controls on the median value and timing of $\bar{\alpha}_{yr}^{min}$ across

the outlet glaciers (Table 3). To avoid the circular nature of aspect data (i.e. where 0° and 360° are equal), values of mean aspect (in degrees) are converted into Cartesian coordinates and correlated independently. A strong and significant association is found between the magnitude of $\bar{\alpha}_{yr}^{min}$ and the proportion of topographic shading (r = 0.741, p = 0.008). Similarly, strong and significant associations are also found with the timing of $\bar{\alpha}_{yr}^{min}$ (north/south component of glacier aspect: r = 0.805, p <

0.002; proportion of topographic shading: r = -0.782, p = 0.003). These correlations, as well as Fig. 6, suggest that glaciers with northerly aspects and low topographic shading (cluster 1) exhibit a lower and delayed $\bar{\alpha}_{yr}^{min}$. Conversely, glaciers with southerly aspects and heavy topographic shading (cluster 2) typically have a higher $\bar{\alpha}_{yr}^{min}$, which occurs earlier in the year.

Interestingly, Fig. 5d shows that despite a large interannual variability, the timing of $\bar{\alpha}_{yr}^{min}$ appears to have evolved over the period 2010-2018 towards late summer for clusters 1 and 3, while this timing remained relatively unchanged for the two glaciers in cluster 2. However, Figure 5b reveals no significant temporal trend for $\bar{\alpha}_{yr}^{min}$ within each cluster (the Pearson correlation coefficient between $\bar{\alpha}_{yr}^{min}$ and the year yielded p = 0.20, 0.70 and 0.52 for clusters 1, 2 and 3, respectively). A delayed $\bar{\alpha}_{yr}^{min}$ could indicate a longer ablation duration, and consequently correspond to a lower $\bar{\alpha}_{yr}^{min}$. However, we only find a weak relationship between $\bar{\alpha}_{yr}^{min}$ and its timing when aggregating the three clusters ($R^2$=0.26, p < 0.001, see Fig. 5c). This correlation proves to be weaker when considering all glaciers individually, with the Julian Day of the minimum albedo only explaining 11% (p < 0.001) of the variability of $\bar{\alpha}_{yr}^{min}$. Furthermore, this relationship loses significance within clusters (cluster 1: $R^2$=0.20, p = 0.058; cluster 2: $R^2$=0.08, p = 0.248; cluster 3: $R^2$=0.03, p = 0.472).

## 4.5 Gardens of Eden and Allah (2000 – 2018)

We use the same methodology as the EOSS survey to characterise and compare the variability of $\bar{\alpha}_{yr}^{min}$ across the 12 outlet glaciers. $\bar{\alpha}_{yr}^{min}$ across the 12 outlet glaciers are averaged to establish $\bar{\alpha}_{0}^{min}$ (56.5%), from which yearly departures are calculated. The combined 19-year record of $\bar{\alpha}_{yr}^{min}$ variability averaged across the 12 outlet glaciers of the GoEA is shown in Fig. 7. The use of $\bar{\alpha}_{0}^{min}$ follows the theory outlined by the EOSS survey, where the long-term average elevation of the SLA is assumed to approximate a glacier in its equilibrium state. The main limitation of $\bar{\alpha}_{0}^{min}$ is that the record only consists of 19 years of data, as opposed to almost 40-years for the EOSS survey. In addition, there is no compelling proof that glaciers in the Southern Alps have been in equilibrium during either period. Therefore, it is possible that the value of $\bar{\alpha}_{0}^{min}$ proposed here actually represents the GoEA in a state that is not in equilibrium. In this instance, $\bar{\alpha}_{0}^{min}$ is limited to describing relative change over the observation period, as opposed to providing a measure for quantitative mass balance in absolute terms.

Negative departures from $\bar{\alpha}_{0}^{min}$ correspond to years where glacier-wide minimum albedo is lower than average, and in turn a more negative mass balance than average is expected. The opposite is expected for positive departures. Therefore, based on the relationship between $\bar{\alpha}_{yr}^{min}$ and annual mass balance, it can be inferred that these shifts broadly correspond to relative shifts in glacier mass balance. In general, the glaciers of the GoEA appear to have undergone four changes to $\bar{\alpha}_{yr}^{min}$ since 2000. For the first three years of the record (2000-2002), $\bar{\alpha}_{yr}^{min}$ across the GoEA was close to, or below average, followed by a positive shift for 2003-2007, with a maximum positive departure in 2004 (5.11%). Between 2008 and 2013, departures were largely negative, notably in 2008, 2011 and 2013. However, since 2014 the fluctuation in $\bar{\alpha}_{yr}^{min}$ has increased substantially, highlighted by the very positive year in 2017 and very negative years of 2016 and 2018. The departure in 2018 is the most negative departure across the entire MODIS record (4.21% below $\bar{\alpha}_{0}^{min}$). This value is also consistent with observations at Brewster Glacier and confirms the widespread effect of the 2018 summer heatwave on New Zealand glaciers (Salinger et al., 2019a).

The behaviour of the glacier-wide surface albedo anomaly on the GoEA over this 19-year period couples reasonably well with $EOSS_{Alps}$, with about half of the variability explained ($R^2$ = 0.55, p<0.001). Notably, the largest negative departures in $\bar{\alpha}_{yr}^{min}$ are in general consistent with high snowlines observed across the index glaciers of the Southern Alps in 2000, 2008, 2011, 2016 and 2018. Despite the relatively limited length of the series, the agreement between the two methods in years where the snowline departure is positive ($R^2$ = 0.68; n = 9, p=0.006) is much stronger than in years with a negative departure ($R^2$ = 0.22; n = 9, p=0.201). Salinger et al. (2019a) also highlight the overwhelmingly positive departures of the transient snowline during the 2018 summer, with $EOSS_{Alps}$ estimated to be 386 m above the long-term average from linear regression with the EOSS of

Tasman Glacier alone. Since this estimate is derived with a very different methodology than the traditional EOSS survey, it is not included in the current analysis. Nonetheless, such a large positive departure is supported by widespread reports of exceptional ablation of glaciers in the Southern Alps. The effect of this extreme summer on the GoEA glaciers is now also supported by results in this study using the MODIS driven albedo method.

## 4.6 Assessment of $\bar{\alpha}_{yr}^{min}$ on Lambert Glacier (2016-2018)

The full 19-year time-series of $\bar{\alpha}(t)$ retrieved from Lambert Glacier exemplifies the annual variability in $\bar{\alpha}_{yr}^{min}$ (Fig. 8). The raw MODIS-derived values of $\bar{\alpha}(t)$ are indicated in red, along with the three-period rolling average used to illustrate the seasonality of the albedo signal. In lieu of the glaciological mass balance data needed to relate the $\bar{\alpha}_{yr}^{min}$ to quantitative mass balance, we use Sentinel-2 data to independently support the MODIS record. The images in Fig. 9 show the end of the ablation season and maximum altitude reached by the summer snowline observed on Lambert Glacier in Sentinel-2 images during the 2016, 2017, and 2018 summer periods. Figure 9a shows the snowline captured on 30 April 2016, 12 days later than the $\bar{\alpha}_{yr}^{min}$ estimated from the MODIS record (18 April). The snowline can be distinguished by the separation between the bright white snow and discoloured ice and firn, visible above ca. 1900 m and up to 2000 m at some locations. The snowline in the 29 March 2017 image is located at ca. 1820 m, slightly above the large icefall that separates the upper and lower glacier. A much higher proportion of snow covering the glacier surface corresponds to a 6% increase in the $\bar{\alpha}_{yr}^{min}$ during the 2017 summer, and is found to be 10 days later using the albedo method (8 Aril). This is consistent with the lower ablation reported by the EOSS programme in 2017 (Willsman et al., 2018). Despite a relatively large snowfall event brought by cyclone Gita in mid-February 2018, the image from 29 March 2018 reveals high ablation and a similar snowline to 2016, although the proportion of exposed ice and firn appears to be slightly larger. These events are captured in the albedo record, with a short-lived rise in $\bar{\alpha}(t)$ corresponding to Gita during February. $\bar{\alpha}_{yr}^{min}$ is reached 19 days earlier (10 March) than the chosen Sentinel-2 image, with its magnitude lower than in 2016. The Sentinel-2 observations are consistent with the relative changes of $\bar{\alpha}_{yr}^{min}$ captured by the albedo method, as well as the effects of heatwaves on New Zealand glaciers in 2016 and 2018 (Salinger et al., 2019a; 2019b; Willsman et al., 2017).

The Sentinel-2 images shown in Fig. 9 also illustrate the complexity of defining the summer snowline elevation on topographically complex glaciers such as Lambert. Despite the limited number of cloud-free images, the sequence of Sentinel-2 images obtained during the summer and through to the arrival of seasonal snow proved to be key in interpreting the evolution of the snowline. They were found to be available within a period of less than three weeks from $\bar{\alpha}_{yr}^{min}$ determined using the albedo method, while the 2016 and 2017 EOSS surveys captured Vertebrae Col glacier on 11 and 9 March, or 40 and 38 days earlier than $\bar{\alpha}_{yr}^{min}$, respectively (see the albedo record for Vertebrae Col glacier in Fig. 10; note that reports from the EOSS 2018 campaign have not been released to date).

## 4.7 Links between $\bar{\alpha}_{yr}^{min}$ and EOSS

Previous applications of the albedo method to New Zealand glaciers have developed strong relationships ($R^2 = 0.89$ and $R^2 = 0.87$, see Sirguey et al., 2016 and Rabatel et al., 2017) between the MODIS-derived $\bar{\alpha}_{yr}^{min}$ and EOSS $SLA_i$. Across the GoEA, SLA records only exist for both Vertebrae Col glaciers. However, with 0.7 $km^2$ surface area, Vertebrae Col 25 is substantially smaller than the 2 $km^2$ recommended to develop a reliable glacier-wide albedo average with enough MODIS pixels (Sirguey et al., 2016). Although the albedo signal for Vertebrae Col 25 is obtained from only 11 250-m resolution pixels, the time-series appears to reflect the typical seasonal cycle of albedo, albeit with a larger range in the maximum and minimum albedos than those observed from Lambert Glacier (Fig. 10).

Between 2000 and 2017, the MODIS record of $\bar{\alpha}_{yr}^{min}$ for Vertebrae Col 25 explains nearly half of the variability observed in the EOSS SLA$_i$ (Fig. 11; $R^2 = 0.43$, $p = 0.003$). Alternatively, $\bar{\alpha}_{yr}^{min}$ of Angel Glacier exhibits the strongest relationship with EOSS observations at Vertebrae Col 25, accounting for 69% of its variance ($p < 0.001$). Other glaciers in the GoEA, namely Lambert, East Lambert, and Eve, each capture half or more of this variance as well (52%, 50%, and 55%, respectively). Overall, it appears that the relationship between EOSS observations at Vertebrae Col 25 and $\bar{\alpha}_{yr}^{min}$ of each glacier is related to topographic shading ($R^2 = 0.48$, $p = 0.012$), slope ($R^2 = 0.42$, $p = 0.022$), and the north/south component of glacier aspect ($R^2 = 0.34$, $p = 0.045$). In contrast, glacier size ($R^2 = 0.03$, $p = 0.609$) and elevation ($R^2 = 0.20$, $p = 0.142$) exhibit no significant role in determining this relationship. From the clustering analysis of the 12 glaciers in the GoEA, it is evident that changes in $\bar{\alpha}_{yr}^{min}$ over the unconfined glaciers in cluster 1 are consistent to EOSS observations (mean $R^2 = 0.51$). More confined west-facing glaciers of cluster 3 exhibit a weaker relationship (mean $R^2 = 0.36$). Finally, $\bar{\alpha}_{yr}^{min}$ of the two most confined south-facing glaciers in cluster 2 do not have a strong relationship to the EOSS record (mean $R^2 = 0.19$).

## 4.8 The contribution of Aqua MODIS

### 4.8.1 MODIS albedo

Figure 12a illustrates the agreement between the MODImLab 250 m WS albedo retrieved from Terra and Aqua MODIS images across four days (3 March 2009, 9 March 2010, 8 March 2012, 10 March 2013). To account for the role of shadows during the albedo retrieval process, each pixel within the GoEA was assigned to one of four "shading classes", depending on the presence of shade in each image. Pixels in Class 1 were unshaded in both images, Class 2 and Class 3 pixels were only shaded in the Aqua or Terra image respectively, and Class 4 pixels were shaded in both images. Of the total 2196 matching pixels from these image pairs, 83.5% belonged to Class 1. The linear regression for these 1834 pixels is significant ($p < 0.01$), and strong ($R^2 = 0.58$), although slightly askew to a linear 1:1 relationship (Fig. 12a). Individually, each of the four days displays a similar trend and distribution, with $R^2$ values ranging between 0.51 and 0.71, and gradients between 0.75 and 0.85.

Figure 12a also shows an increase in the variability of albedo between the sensors at higher values. Overall, this agreement is consistent with the expected 10% accuracy of the albedo retrieval method (Dumont et al., 2011, 2012; Sirguey et al., 2016). The increased variability coincides with the highest density of points; where 83% of pixel albedo values fall between 0.4 and 0.8. Due to the uneven distribution of albedo across the spectrum, a second linear regression was run on a stratified random sample of 150 Class 1 pixels (Fig. 12b). The linear regression of these resampled data show a much-improved fit ($R^2 = 0.88$, $p < 0.001$) that closely approximates the 1:1 relationship, and demonstrates that the degraded band 6 of MODIS[A] does not compromise MODImLab albedo retrieval. The slightly lower albedo found in Aqua images compared to Terra may also be explained by the timing, as snow and ice surfaces would undergo transformation compatible with a decrease in albedo from morning to afternoon.

To characterise the variability between the sensors in more detail, we present the spatial distribution of averaged residuals between maps of glacier surface albedo from MODIS[T] and MODIS[A] across the GoEA (Fig. 13). Although Fig. 13 confirms the good agreement between the sensors over large parts of the GoEA, it shows large departures around the fringes and near rock outcrops (+0.23/-0.38), often close to steep, complex terrain and involving mixed pixels. In particular, two large steep-sided rock outcrops in the south of Lambert Glacier and the west icefall of Angel Glacier descending from Mt Farrar correspond to MODIS[A] albedo being substantially larger than MODIS[T]. Close inspection of imagery before and after correction, as well as shadow maps reveal the larger extent of cast shadows produced by outcrops in the afternoon and affecting MODIS[A] imagery. Issues of overcorrecting spectral reflectance in cast shadows is known to challenge MODImLab (Davaze et al., 2018), and appears again to cause overestimates of MODIS[A] albedo. Figure 12 also demonstrates this effect with overestimation of albedo

by MODIS$^A$ relative to MODIS$^T$ for Class 2 pixels (MODIS$^A$ pixels in the shade) and conversely for Class 3 pixels (MODIS$^T$ pixels in the shade).

MODIS$^A$ involves a delay in image capture from 10:30 am to 1:30 pm. Towards the winter solstice, this decreases the proportion of shaded pixels in the steep terrain of the Southern Alps. The change in the solar zenith angle caused by the delayed timing of the image capture means pixels on south- and southwest-facing slopes have a higher chance of receiving incident radiation. This effect was seen across the GoEA, where a much lower proportion of the total pixels in the GoEA were shaded in MODIS$^A$ images captured near the winter solstice, compared to MODIS$^T$ (34.5% and 49.8%, respectively). However, in summer and toward April, steep rock outcrops cast large shadows in the afternoon that challenge albedo retrieval with MODIS$^A$. By May, low sun zenith angles cast longer shadows that are not accurately modelled due to the resolution and accuracy of the DEM. Unpredicted shadows yield severe underestimations of surface albedo, in particular affecting confined glaciers of cluster 3. Although MODIS$^A$ images could help capture a better albedo signal in such cases, Figure 5a demonstrates that, over the period of this study, imagery from MODIS$^T$ remained suitable to capture $\bar{\alpha}_{yr}^{min}$ before this issue becomes problematic by May. Nevertheless, despite the computational burden, the systematic processing of MODIS$^A$ imagery may gain merit as the timing of $\bar{\alpha}_{yr}^{min}$ appears to be more often delayed to late summer or early autumn (Fig. 5).

Finally, Lyapustin et al. (2014) documented calibration issues with MODIS Collection 5 data, and concluded that major calibration trends were removed in C6. The stability of the C6 calibration was confirmed by Sayer et al. (2015). In the context of retrieving time series of snow and ice albedo with MODIS, Casey et al. (2017) stressed how C5 data could compromise the detection and interpretation of trends, and concluded that C6 is preferable. In the case of MODIS Terra, albedo time series derived from C6 data by MODImLab shown in Fig. 8 and 10 reveal no visible trend in winter albedo. It seems unlikely, if a trend in winter snow albedo existed, that it would be matched and concealed by a calibration issue of exactly the opposite magnitude. Our results therefore support the alternative hypothesis that there is no detectable trend in winter albedo nor calibration issue over the length of the record produced by this study. This is further supported by the cross-platform agreement shown in Fig. 12.

### 4.8.2 MODIS cloud cover

While MODIS$^A$ potentially provides a means to capture more shadow-free pixels across the GoEA, its use is inhibited by daily development of cloud cover over the Southern Alps. Figure 14 shows MODIS$^A$ images display a consistently higher proportion of cloudy pixels over the GoEA than MODIS$^T$ images over the course of a year. While this trend is consistent throughout the year, it is pronounced through the summer ablation period, at the critical time when the $\bar{\alpha}_{yr}^{min}$ is likely to be observed. A likely driver of this pattern is daytime heating of the atmosphere through the warmer summer months, resulting in convection and aiding cloud development over the course of the day. Ultimately, while MODIS$^A$ shows some promise in its ability to supplement the MODIS$^T$ dataset as discussed above, the high variability of albedo from mixed-pixels, and the daily increase in cloud cover means that its application is limited in New Zealand.

Interestingly, the cloud cover results from both MODIS$^A$ and MODIS$^T$ images are still of use, as the spatial characterisation of cloud cover over the Southern Alps is very limited (Wardle, 1986). Current research is largely limited to point based cloud data from automatic weather stations (e.g. Conway et al. 2015). Cloud cover patterns and dynamics are important for glaciers, as clouds play a key role in influencing incident solar radiation at the glacier surface (Conway and Cullen, 2016). Figure 15 displays the non-uniform distribution of monthly average cloud cover across the processed area. Over the period of study, the frequency of clouds in pixels west of the Main Divide is as high as 90% during summer months, and reaches a minimum of 35% in some areas during winter, reflecting Fig. 14. During summer, cloud spill-over is limited to within a few kilometres east

of the Main Divide, contrasting to relatively uniform conditions through winter. It is also possible to identify the presence of 'cloud hot-spots' that persist through the 19-year record.

## 5 Discussion

### 5.1 Comparison to the EOSS survey

The EOSS survey provides a well-documented and invaluable record of the changes to glaciers in the Southern Alps over the past 40 years. The data collected by the survey bridges the gap in glaciological mass balance records between the termination of the Ivory Glacier programme in 1975 and commencement of the Brewster Glacier programme in 2004. In addition, the correlation between the snowline record and recently developed monitoring methods has allowed past mass balance trends to be reconstructed (e.g. Sirguey et al., 2016). At a larger scale, the use of $EOSS_{Alps}$ has allowed the annual variability of glacier mass balance in the Southern Alps to be broadly characterised (Chinn et al., 2012; Willsman et al., 2018).

Published relationships developed between the MODIS-derived $\bar{\alpha}_{yr}^{min}$ and the EOSS $SLA_i$ departures on Brewster Glacier and Park Pass Glacier were found to be strong (Sirguey et al., 2016; Rabatel et al., 2017). Despite the large differences between the albedo and snowline methodologies, both aim to remotely capture the point of maximum ablation at the glacier surface and use it as a proxy for mass balance. $SLA_i$ – used as an estimate of the ELA – provides an effective measure of glacier mass balance (Rabatel et al., 2005), and should therefore be closely related to $\bar{\alpha}_{yr}^{min}$ (Dumont et al., 2012).

However, measuring $SLA_i$ remains difficult, in particular due to uncertainties associated with the method used to identify snowlines from oblique photos. Complex topography, avalanching, fresh snow, and cloud cover all present additional challenges for deriving consistent results of $SLA_i$. To compound these challenges, limited resources mean that the EOSS survey is required to assume that $SLA_i$ occurs annually in early-March, which involves careful timing of the observation flights (Willsman et al., 2018). In most cases, $SLA_i$ is estimated manually based on the overall appearance of the glacier and snow patches compared to previous years. When $SLA_i$ is digitized, photographs are compared to historical topographic maps or from orthorectified images. Furthermore, the methodology often involves an interpretation and assessment of the appearance of an individual glacier in relation to other glaciers in the programme, observed during the same or from different surveys.

Despite the coarse resolution of the MODIS sensor, it has been demonstrated in this study that the albedo method is capable of retrieving robust time series of $\bar{\alpha}_{yr}^{min}$ that show strong to moderate relationships to EOSS signals (glaciers of clusters 1 and 3, respectively). There are some clear advantages in using the albedo method over an EOSS approach, which include the approach being repeatable and not being temporally constrained. While the EOSS programme yields an archive of invaluable images, a number of photos are impacted by transient snow that impact the quality of the $SLA_i$ estimates. The timing of the surveys also mean that some $SLA_i$ estimates might not fully capture late summer melt (Sirguey et al., 2016). Monitoring albedo until the onset of the accumulation season (winter) allows sustained late season ablation to be recorded. The large variability in timing of $\bar{\alpha}_{yr}^{min}$, as well as the apparent trend towards a delayed occurrence for most glaciers in the GoEA (Fig. 5 and 6), demonstrates the benefit of systematically monitoring glacier surface albedo. Thus, the albedo method provides new insights about glaciers in the Southern Alps that are not captured by the EOSS programme.

The EOSS programme reports considerably high intra-correlations across the 50 index glaciers that sample the Southern Alps, with pair-wise $R^2$ ranging 0.22 to 0.96 and averaging 0.69. Overall, our systematic application of the albedo method on the limited geographical extent of the GoEA reveals a degree of variability in glacier response that challenges the highly consistent behaviour suggested by the EOSS programme across the Southern Alps. Strong and significant intra-correlations in $\bar{\alpha}_{yr}^{min}$ across glaciers of the GoEA exist but do not dominate, with pair-wise $R^2$ ranging from 0.12 to 0.88 and averaging only 0.52. Similarly, the correlation between the average GoEA albedo signal to $EOSS_{Alps}$ is only $R^2 = 0.55$ (see Fig. 7), while the average correlation

between $SLA_i$ and $EOSS_{Alps}$ is reported at 0.81. These contrasting results suggest the EOSS approach may have led to spatial and temporal auto-correlation of the SLA records, which in turn build very strong correlations across SLA records of EOSS index glaciers and to $EOSS_{Alps}$. The outcome of this is that it may have dampened or concealed the variability of mass balance response of glaciers in the Southern Alps.

**5.2 Implications of variability of albedo on mass balance**

Given the scarcity of surface mass balance measurements in the Southern Alps, and the spatial and temporal limitations of obtaining imagery from observational flights, satellite remote sensing provides a powerful tool to increase the number of glaciers being currently monitored in New Zealand. To relate the observed variability of albedo to changes in annual mass balance, the relationship between the magnitude and timing of $\bar{\alpha}_{yr}^{min}$ in each of the three glacier clusters identified on the GoEA (Table 2) needs to be further examined. There is evidence that the occurrence of $\bar{\alpha}_{yr}^{min}$ has been delayed in time (later in the year) on all glaciers other than those located on steep, south-facing slopes (Cluster 2) (Fig. 5d). Intuitively, a delay in the timing of $\bar{\alpha}_{yr}^{min}$ on 10 of the 12 glaciers (Clusters 1 and 3) might be expected to result in a more negative summer mass balance by virtue of the ablation season being extended. However, there is no visible trend in the magnitude of $\bar{\alpha}_{yr}^{min}$ in any of the clusters during the observation period (2000-2018). Also, there is only a weak relationship between a delay in timing and the magnitude of $\bar{\alpha}_{yr}^{min}$ (Fig. 5c), suggesting that glacier-wide albedo is not necessarily lower if it is delayed. We interpret this dichotomy as evidence that the atmospheric controls on mass balance are complex, and that the magnitude of ablation is in part sensitive to discrete weather events (Cullen et al., 2019), which either bring snowfall (increase albedo) or enhance ablation (decrease albedo) in summer. There is also compelling evidence that clouds play an important role in modulating the energy available for ablation over glaciers in the Southern Alps (Conway and Cullen, 2016), and are likely to influence both the timing and magnitude of $\bar{\alpha}_{yr}^{min}$ in the GoEA, thus confounding the relationship. In particular, clouds are very common to the west of the Main Divide in the GoEA region during summer (Fig. 15), and their spatial and temporal variability elsewhere may be an underlying factor in controlling the observed variability in $\bar{\alpha}_{yr}^{min}$. We believe there is a risk that some of this complexity is being missed using the single snapshot approach of the traditional EOSS programme. Arguably, the albedo method provides more certainty of the atmospheric controls on mass balance, as it provides a platform of continuous observations of albedo and cloud cover over a large number of glaciers throughout the ablation season.

Despite the observed complexity between the timing and magnitude of $\bar{\alpha}_{yr}^{min}$ over the observation period, there is evidence of a shift towards more negative annual departures of the mean $\bar{\alpha}_{yr}^{min}$ since 2008 (Fig. 7), which is consistent with the EOSS record, implying cumulative losses in mass balance for the glaciers of the GoEA. This mass loss is consistent with glaciological observations of mass balance from Brewster Glacier (Cullen et al., 2017) and the modelling of glacier response to changes in climate in the Southern Alps (Mackintosh et al., 2017; Salinger et al., 2019a, b). This shift towards mass loss appears to be widespread, with the 2018 summer heatwave reportedly the most damaging for glaciers in the Southern Alps over the last half century (Salinger et al., 2019a). We anticipate that the GoEA are particularly vulnerable to a warming climate, as the average elevation of the icefields is close to the regional snowline, which is estimated to be 1950 m a.s.l. (Chinn, 2001). Cullen and Conway (2015) showed that the majority of precipitation at the altitude of Brewster Glacier falls at an air temperature that is very close to the threshold between snow and rain. Thus, small increases in air temperature controlled by warming is likely to impact the amount of precipitation falling as snow on the GoEA, which in turn will impact surface albedo and mass balance, potentially driving a rapid decline under a sustained warming trend. While the observations we have obtained using the albedo method suggest that most glaciers are vulnerable, the contrasting response of glaciers contained on steep, south-facing slopes (Cluster 2) indicates they are likely to survive this demise longer.

Lastly, the positioning of the GoEA across the Main Divide of the Southern Alps is particularly important from a management perspective. Because the icefields straddle the Main Divide, they contribute meltwater to the catchments of the Rangitata River on the east coast, and the Wanganui and Whataroa rivers to the west. Given the important role of snow and ice in New Zealand's water resources, changes in the volume of these icefields have the potential to impact the hydrology of these rivers in the future.

## 5.3 Limitations and developments of the albedo method

The results presented in this study rely on the inherent accuracy of MODImLab. The specific error associated with the MODImLab retrieval over the GoEA is uncertain due to the lack of *in situ* data. However, Dumont et al. (2012) quantified the relative error between field measurements and the MODImLab 250 m broadband albedo to be approximately ± 10% (RMSE = 0.052), confirmed by Sirguey et al. (2016). This value is an average estimate, with error recognised to be up to twice as high around the mixed-pixel margins of glaciers compared to the clear snow/ice pixels near the centre (Dumont et al., 2012). As a result, it is expected the uncertainty will be variable between glaciers, where estimates of $\bar{\alpha}(t)$ on large glaciers (i.e. Lambert Glacier) may be more reliable than those of small glaciers with high edge effects (i.e. East Lambert Icefall) (Brun et al., 2015).

Supporting Rabatel et al. (2017) and Davaze et al. (2018), we find that the error in the albedo retrieval caused by surface shading does not affect the identification of the summer $\bar{\alpha}_{\text{yr}}^{\text{min}}$. On glaciers where shading is most prominent (steep sloping, south-facing, topographically incised), we show the $\bar{\alpha}_{\text{yr}}^{\text{min}}$ is typically reached much earlier than early-April (when the shading-induced decrease in $\bar{\alpha}(t)$ begins). This means that while New Zealand has a large number of glaciers that exhibit similar hypsometry to those glaciers in Cluster 2, in most cases, the albedo retrieval will still yield reliable values of $\bar{\alpha}_{\text{yr}}^{\text{min}}$.

The masking technique proposed in this study should simplify the process of objectively converting glacier outlines to glacier masks. As we show in Sect. 3.2 and 4.1, the new approach to masking glacier boundaries over the GoEA yields values of $\bar{\alpha}(t)$ over the ablation period within 3% of the more subjective method employed by previous studies. The success of this technique may support the rapid application of the albedo method to new glaciers using existing outlines, regardless of whether they are made up of mixed-pixels and/or debris-cover. In addition, the snow-covered-area filter may help to reduce some of the problems of using a static glacier mask (e.g. the glacier extent changing substantially over the observation period).

Finally, the use of MODIS imagery to apply the albedo method remains applicable only for glaciers that are large enough to allow albedo to be retrieved and averaged over a number of pixels. Davaze et al. (2018) show that $\bar{\alpha}_{\text{yr}}^{\text{min}}$ can be successfully retrieved on glaciers covering as little as 0.5 km$^2$ with good correlation to annual mass balance. This is consistent with the agreement we obtained between $\bar{\alpha}_{\text{yr}}^{\text{min}}$ and EOSS on Vertebrae Col 25 (0.7 km$^2$ or 11 pixels), although it is believed to stretch the reliance on MODIS for the albedo method. The temporal resolution achievable by combining multi-spectral imaging from higher resolution sensors such as Sentinel-2 and Landsat 8 OLI justifies the development of snow and ice albedo products to resolve $\bar{\alpha}_{\text{yr}}^{\text{min}}$ on smaller glaciers. Such development is desirable to advance the albedo method and support the widespread monitoring of New Zealand glaciers.

## 6 Conclusion

The results from this study represent the next step towards the use of the albedo method for widespread monitoring of glaciers in the Southern Alps. Following the successful application to Brewster Glacier and Park Pass Glacier, we have produced a 19-year long coherent seasonal signal of glacier-wide albedo on the previously unstudied Gardens of Eden and Allah (GoEA). These results have supported and advanced key aspects of the methodology that will inform future applications in the Southern Alps and beyond. The key findings can be summarised as:

1. A new objective glacier masking approach has been developed that compares favourably to a more traditional manual method of identifying suitable pixels to calculate glacier-wide albedo.

2. The annual minimum glacier-wide albedo ($\bar{\alpha}_{yr}^{min}$) for individual glaciers ranges between 0.42 and 0.70, and can occur as early as mid-January and as late as the end of April. The timing of $\bar{\alpha}_{yr}^{min}$ appears to have shifted to later in the year over the 19-year period on all glaciers other than those located on steep, south-facing slopes. However, there is only a weak relationship between the delay in timing and the magnitude of $\bar{\alpha}_{yr}^{min}$, which suggests that glacier-wide albedo is not necessarily lower if it is delayed.

3. The glacier-wide surface albedo anomaly for the 19-year period explains 55% of the variability in the average annual departure of the 50 index glaciers from the EOSS programme (EOSS$_{Alps}$). The largest negative departures in $\bar{\alpha}_{yr}^{min}$ (lower than average albedo) are consistent with high snowlines, with the 2018 departure the most negative on record. This is consistent with Brewster Glacier and was caused by a marine and terrestrial heatwave over New Zealand (Salinger et al., 2019a).

4. The MODIS record of $\bar{\alpha}_{yr}^{min}$ for Vertebrae Col 25 explains less than half of the variability observed in the EOSS SLA record ($R^2 = 0.43$, $p = 0.003$). However, the relationship is stronger when compared to other GoEA glaciers, with Angel Glacier having the strongest relationship with EOSS observations at Vertebrae Col 25, accounting for 69% of its variance ($p < 0.001$). Lambert, East Lambert, and Eve glaciers each capture half or more of the variance (52%, 50%, and 55%, respectively). The relationship between EOSS observations at Vertebrae Col 25 and $\bar{\alpha}_{yr}^{min}$ of each glacier is related in order of importance to topographic shading, slope and aspect.

5. The EOSS programme has reported on how strongly each of the 50 index glaciers behaviour is related to the mean of all remaining glaciers (known as EOSS$_{Alps}$), and pair-wise regression shows there is high intra-correlation between glaciers. However, the albedo method enables the variability in response of individual glaciers to be explored in more detail, revealing that topographic setting plays a second order control in addition to the regional climate signal (first order control). The albedo method captures enough individual glacier variability on the GoEA to firmly question the validity of the hypothesis that glaciers in the Southern Alps behave as a single climatic unit.

6. For the first time, MODIS imagery acquired by the Aqua platform (MODIS$^A$) has been used successfully to increase the temporal resolution of albedo monitoring using the MODImLab algorithm. There is some evidence to suggest it is capable of capturing diurnal variability in albedo as controlled by changes to snow and ice properties during daytime. Despite cloud being more frequent during the afternoon, especially in summer, there are advantages in using MODIS$^A$ due to higher incident radiation (less shading) on some slopes at certain times of the year.

7. Cloud cover results from MODIS imagery acquired by the Terra (MODIS$^T$) and Aqua (MODIS$^A$) platforms show the spatial and temporal variability of clouds. The frequency of cloud in pixels west of the Main Divide is as high as 90% during summer months, and reaches a minimum of 35% in some locations in winter. There is a strong gradient in cloud cover frequency between regions west and east of the Main Divide, and specific areas appear to be consistently cloudier than others. These complex cloud interactions deserve further attention as they are likely to play a considerable role in glacier surface energy and mass balance.

The key findings presented in this research have provided a platform to further develop and extend the application of the albedo method to monitor glacier behaviour in the Southern Alps. The next logical step will be to attempt an assessment of all of the main glaciated areas of the Southern Alps together at a high temporal resolution, which will complement observations made by the EOSS programme, and expand our understanding of the linkages between glaciers and the climate system. There is

some urgency to do this as our observations of the GoEA, and those obtained from traditional glaciological observations elsewhere, suggest that glaciers in the Southern Alps are undergoing an unprecedented decline at present.

*Data and code availability*. MODIS data used in this research are freely available from the Level 1 and Atmosphere Archive and Distribution System (LAADS) Web. NZSoSDEM is freely available from the koordinates.com geographical data
repository. The MODImLab software is available upon request from PS.

*Author contributions.* PS and NC initiated and coordinated the study. NC and PS obtained funding for the research. AD and PS processed and analysed the MODIS data. AD wrote the first draft of the manuscript, while PS and NC were responsible for the submission and revision of the final research.


*Competing interests.* The authors declare that they have no conflict of interest.

*Acknowledgements.* This research received funding support from the Brian Mason Scientific and Technical Trust and University of Otago Research Grants (ORG0112-0313; ORG0118-0319). The field component of this research was supported
by the Department of Conservation under the concession 52174-RES. A. Dowson was funded by a postgraduate scholarship from the University of Otago, while N. Cullen received support from the Alexander von Humboldt Foundation, Germany to help support the completion of this research. The MODIS Level-1B data were processed by the MODIS Adaptive Processing System (MODAPS) and the Goddard Distributed Active Archive Center (DAAC), and are archived and distributed by the Goddard DAAC. We thank E. Berthier for enabling the acquisition of Airbus DS Pléiades imagery within the Pléiades Glaciers
Observatory (PGO) initiative of the ISIS-CNES programme. The authors also thank both referees for their reviews and constructive comments.

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

**Tables**

Table 1: Comparison of $\bar{\alpha}(t)$ between Mask 1 and Mask 2 from Terra MODIS images between 25 February 2000 and 30 April 2018. Statistics are calculated as average values from the 12 outlet glaciers.

|  | $R^2$ | RMSD | MD |
|---|---|---|---|
| Annual | 0.902 | 0.037 | -0.012 |
| Winter (April – September) | 0.899 | 0.040 | -0.010 |
| Summer (October – March) | 0.910 | 0.030 | -0.013 |

Table 2: Physical characteristics of the 12 outlet glaciers in the GoEA identified for MODIS-albedo retrieval, calculated from the 15-m resolution NZSoSDEM. Cluster membership is defined by K-mean cluster analysis of aspect, slope and topographic shading. An estimate of the number of 250 m raster pixels in each individual glacier mask can be found by dividing the total area by 0.0625 km$^2$.

| Glacier Name | Area (km$^2$) | Aspect | Elevation (m) | Slope (°) | Topographic Shading (%) | Cluster |
|---|---|---|---|---|---|---|
| Abel | 2.57 | S | 1918.7 | 20.7 | 81.5 | 2 |
| Arethusa | 2.57 | E | 1950.3 | 21.9 | 15.6 | 1 |
| Barlow | 1.09 | W | 2027.1 | 18.7 | 51.4 | 3 |
| Beelzebub | 4.44 | W | 2045.5 | 17.5 | 43.8 | 3 |
| Colin Campbell | 3.95 | S | 2008.7 | 24.8 | 82.0 | 2 |
| Eve | 2.73 | SW | 1918.1 | 15.9 | 68.4 | 3 |
| Farrar | 0.86 | W | 1986.2 | 20.0 | 64.5 | 3 |
| Lambert | 9.44 | NE | 1921.4 | 16.7 | 25.5 | 1 |
| Perth | 2.56 | SW | 1985.2 | 18.9 | 64.9 | 3 |
| Unnamed East (East Lambert) | 0.44 | ENE | 1971.0 | 14.5 | 15.4 | 1 |
| Unnamed West (Angel Gl.) | 2.54 | ENE | 1874.3 | 18.0 | 5.4 | 1 |
| Vertebrae Col 25 | 0.70 | SW | 1882.7 | 11.7 | 50.0 | 3 |

Table 3: Correlation between the median timing and magnitude of the $\bar{\alpha}_{yr}^{min}$ and glacier hypsometry using the Spearman's rank coefficient (r) for the 12 outlet glaciers of the GoEA. 2-tailed correlation is significant at the 0.05 level (*) and the 0.01 level (**).

|  | Elevation (m) | Slope (°) | Aspect $\cos(\theta)$ | Aspect $\sin(\theta)$ | Area (km$^2$) | Topographic Shading (%) |
|---|---|---|---|---|---|---|
| $\bar{\alpha}_{yr}^{min}$ (%) | 0.406 | 0.196 | -0.489 | -0.555 | 0.251 | 0.741** |
| $\bar{\alpha}_{yr}^{min}$ (t) | 0.049 | -0.319 | 0.805** | 0.225 | -0.361 | -0.782** |


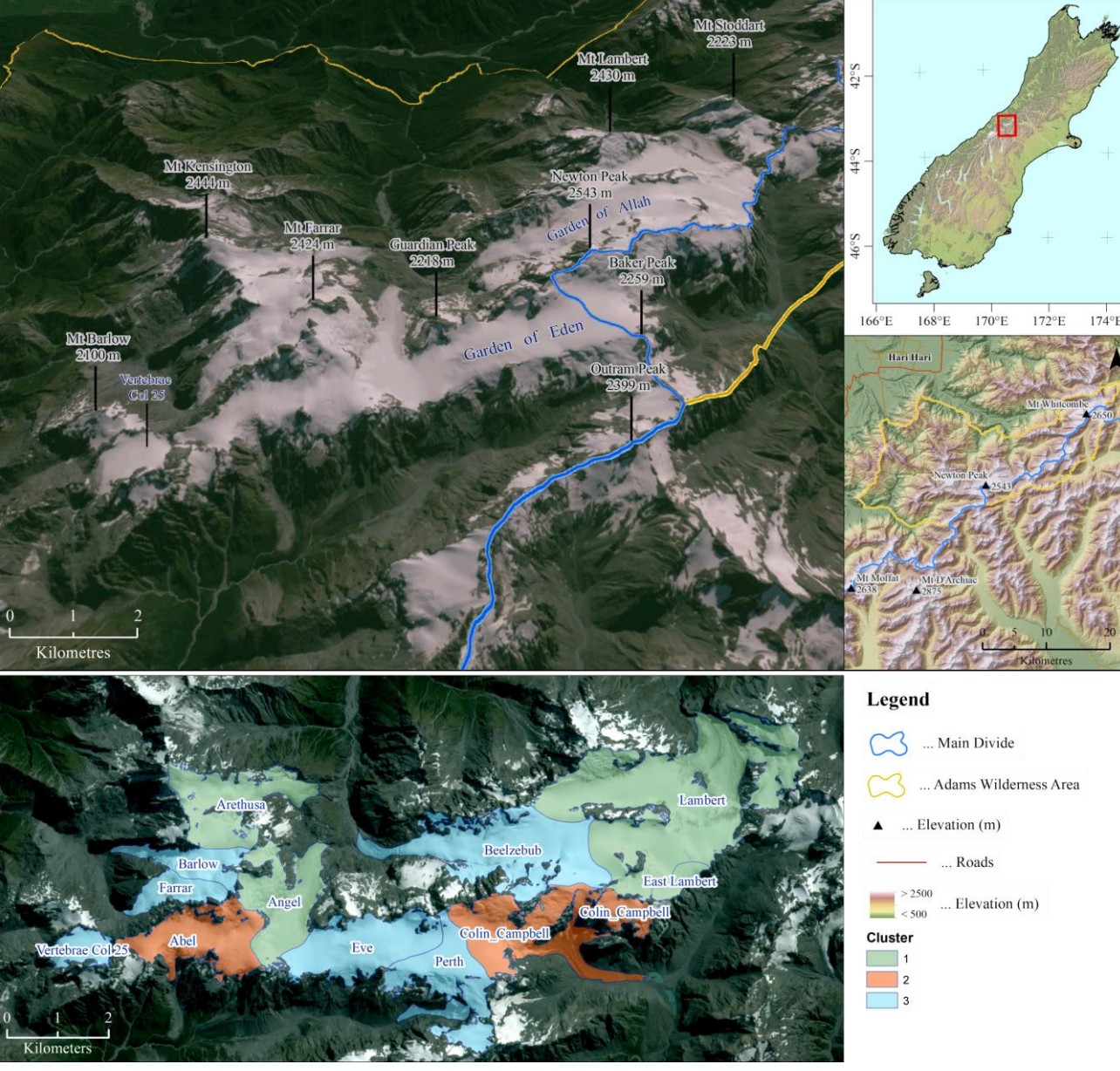

**Figure 1: (top) Location of the Garden of Eden and Garden of Allah within the Adams Wilderness Area, straddling the Main Divide of New Zealand's Southern Alps. Sentinel-2A image from 9 March 2017 draped onto a 15 m-resolution NZSoSDEM (Columbus et al., 2011). (bottom) Outlines of glaciers considered in this study coloured by clusters.**


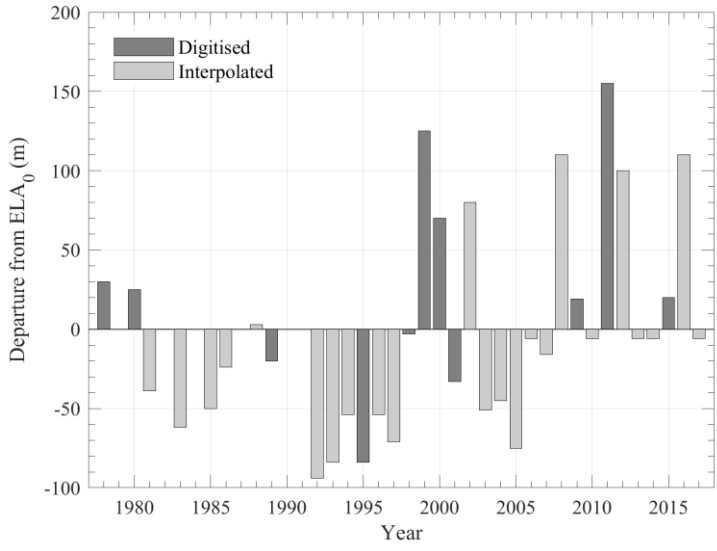


**Figure 2: Record of SLA$_i$ departure for Vertebrae Col 25 from ELA$_0$ (1840 m a.s.l.) between 1977 and 2017 (Willsman et al., 2018).**

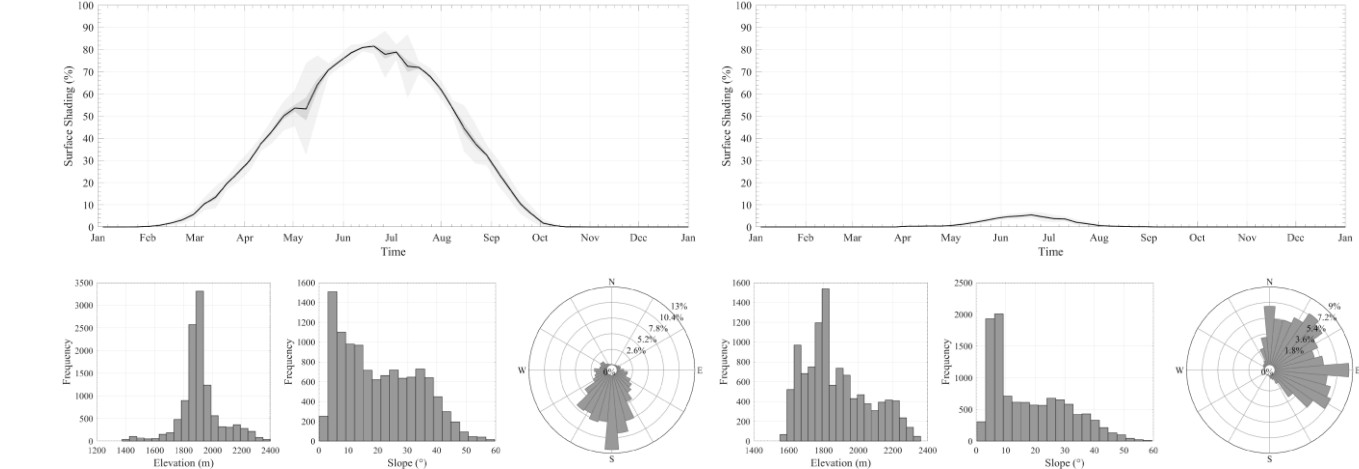

**Figure 3: Surface shading and hypsometry of (left) Abel Glacier and (right) Angel Glacier in the GoEA. Shading is displayed over one year, calculated as a weekly average between 2000 and 2018, used to indicate the nature of the surrounding topography. Topographic measures of (a) elevation, (b) slope and (c) aspect are calculated at 15 m-resolution from the NZSoSDEM.**

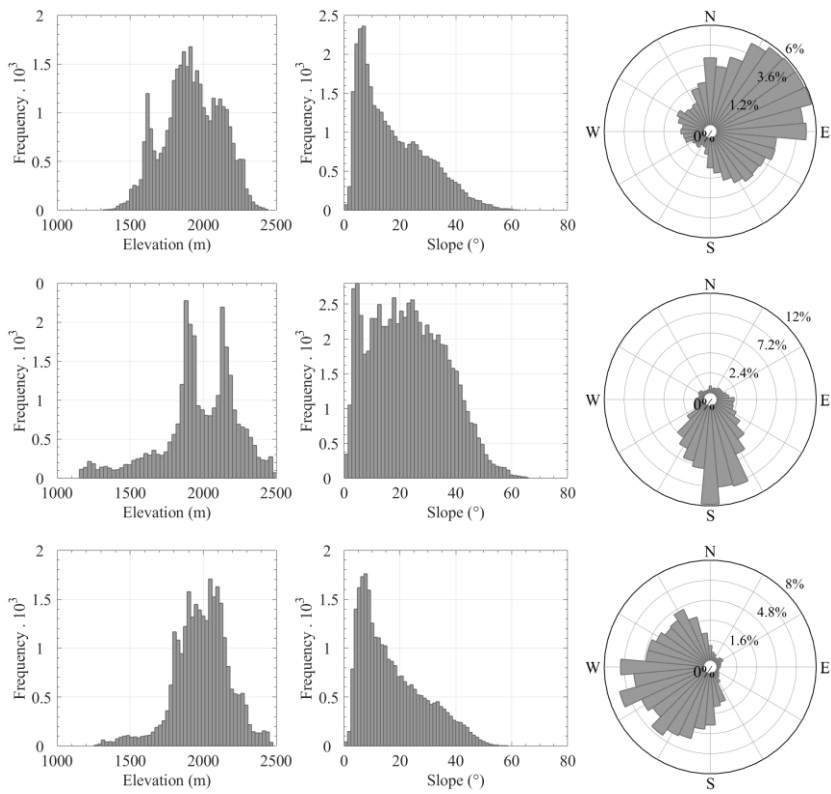

**Figure 4: Topographic measures of (a) elevation, (b) slope and (c) aspect are calculated at 15 m-resolution from the NZSoSDEM for each of the three clusters of glaciers in the GoEA (clockwise from top-left).**



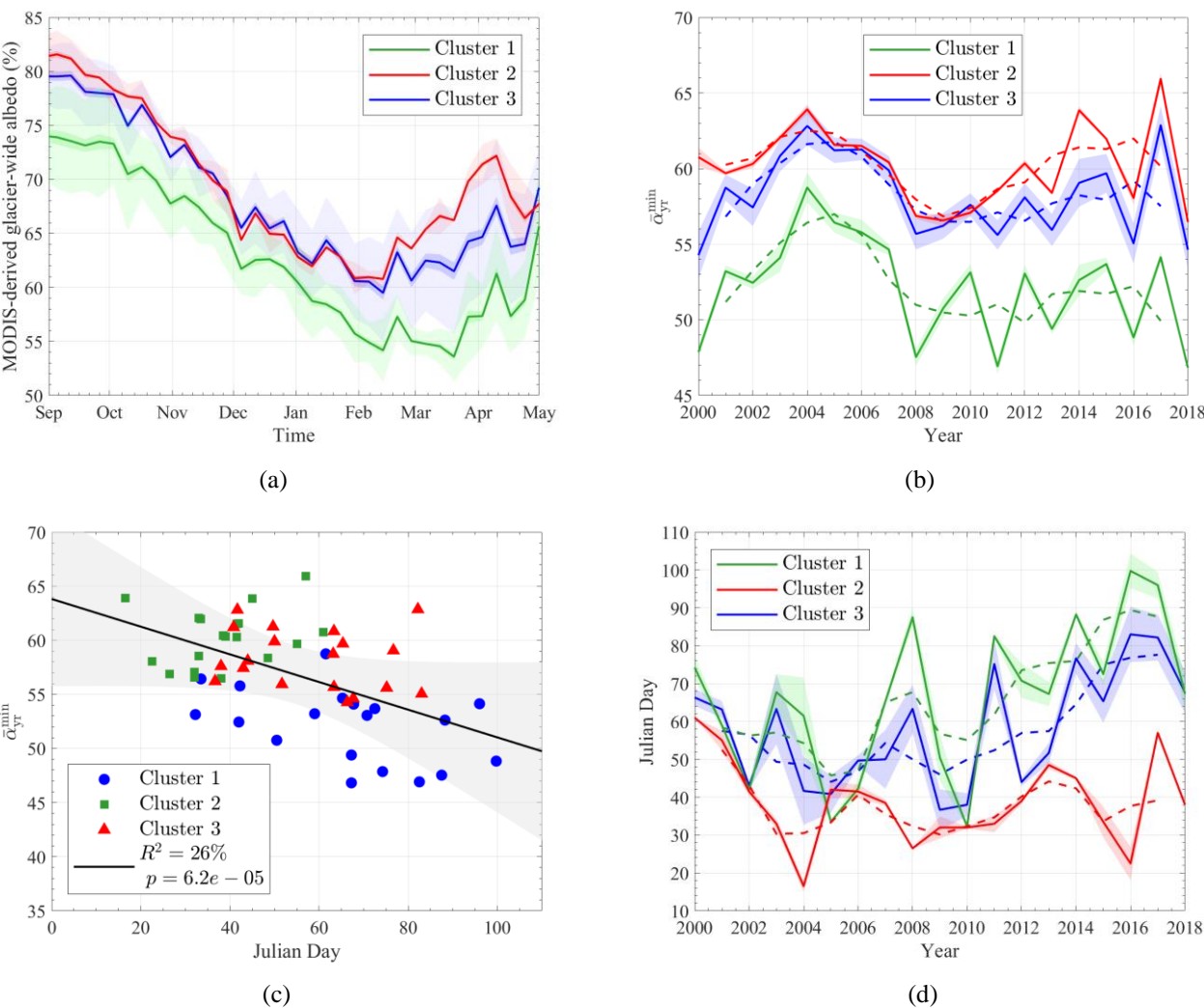

**Figure 5: (a)** MODIS-derived glacier-wide albedo $\bar{\alpha}(t)$ between September and May of the outlet glacier averaged within-cluster. Values are calculated between February 2000 and April 2018 as a mean weekly average. The dark and light shaded envelopes show the standard error and standard deviation of the mean, respectively. **(b)** Temporal changes of $\bar{\alpha}_{yr}^{min}$ averaged within-cluster. **(c)** Correlation between $\bar{\alpha}_{yr}^{min}$ and the its timing in Julian Day; the grey envelope shows the 95% confidence interval for the linear regression model. **(d)** Temporal changes of the timing of $\bar{\alpha}_{yr}^{min}$ in Julian Day averaged within-cluster. Dotted lines show 3-year moving average.

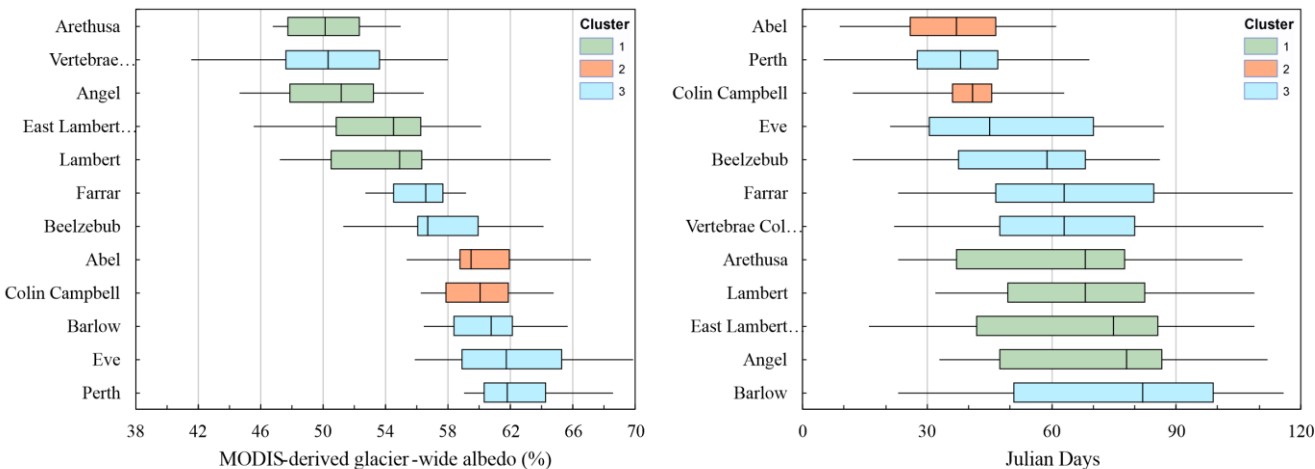

**Figure 6: The (left) magnitude and (right) timing of the $\bar{\alpha}_{yr}^{min}$ over the 12 outlet glaciers of the GoEA between 2000 and 2018. The box plots represent the 19-year median, minimum, maximum and interquartile range of each of the glaciers.**

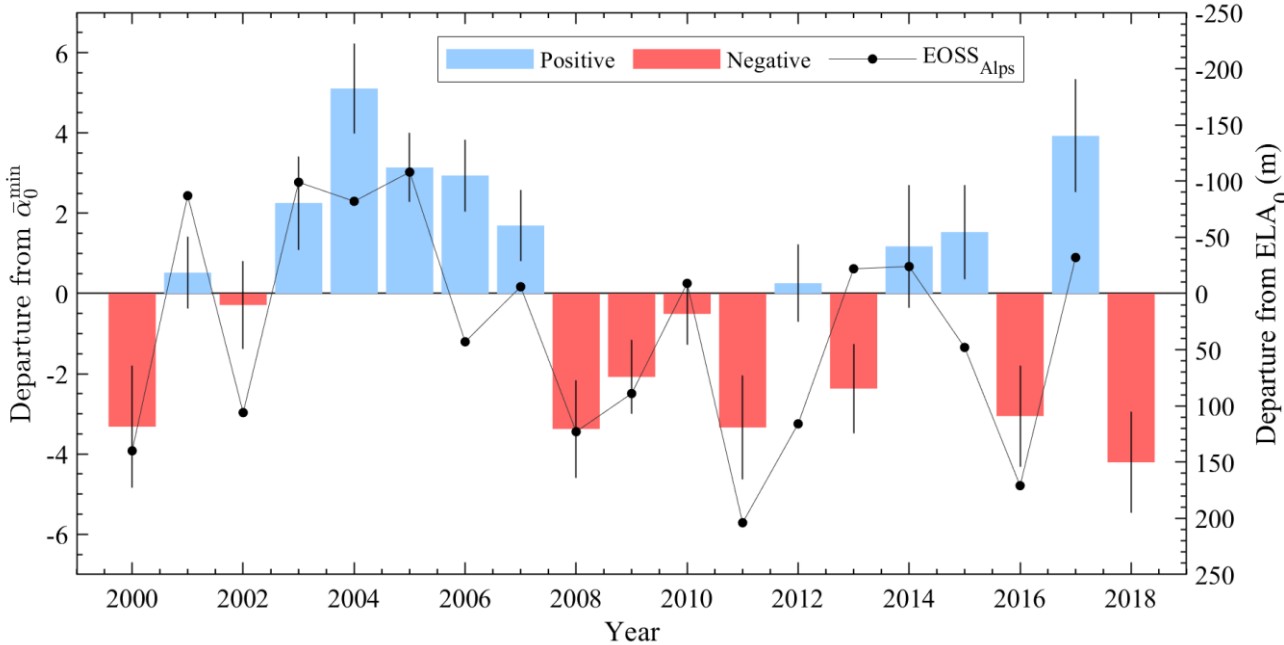


**Figure 7: Annual departure and standard error of the mean $\bar{\alpha}_{yr}^{min}$ across the 12 outlet glaciers from the long-term average $\bar{\alpha}_{yr}^{min}$ ($\bar{\alpha}_0^{min}$; 56.5%) between 2000 and 2018. The average annual departure of the EOSS index glaciers from their respective $ELA_0$ ($EOSS_{Alps}$) is also included. Linear agreement between the records is $R^2 = 0.55$.**


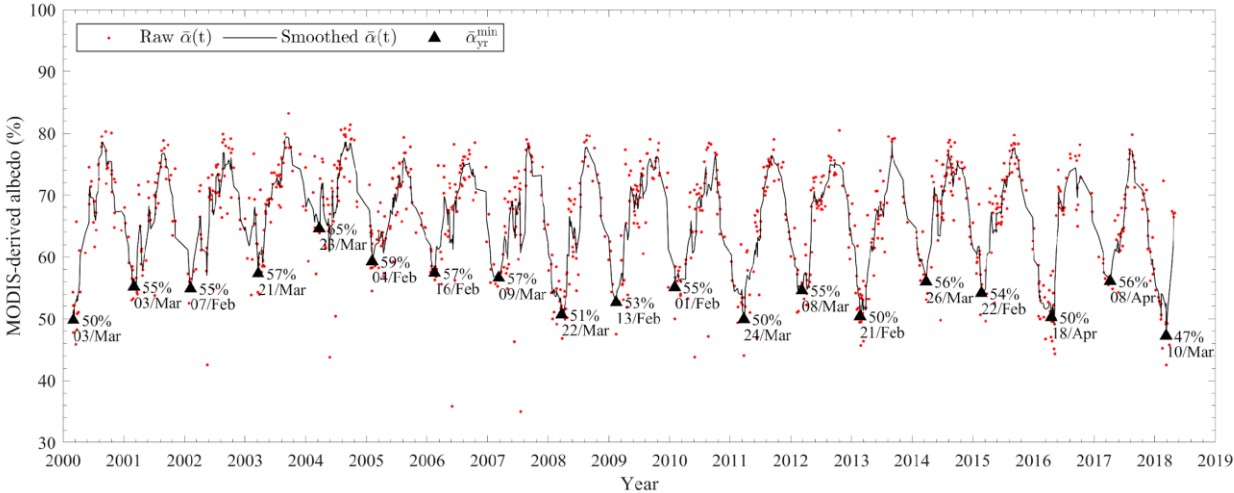

**Figure 8: Near-daily MODIS-derived glacier-wide surface albedo $\overline{\alpha}(t)$ on Lambert Glacier between February 2000 and April 2018, with the 3-period rolling average and annual minimum glacier-wide albedo $\overline{\alpha}_{yr}^{min}$ indicated in black.**

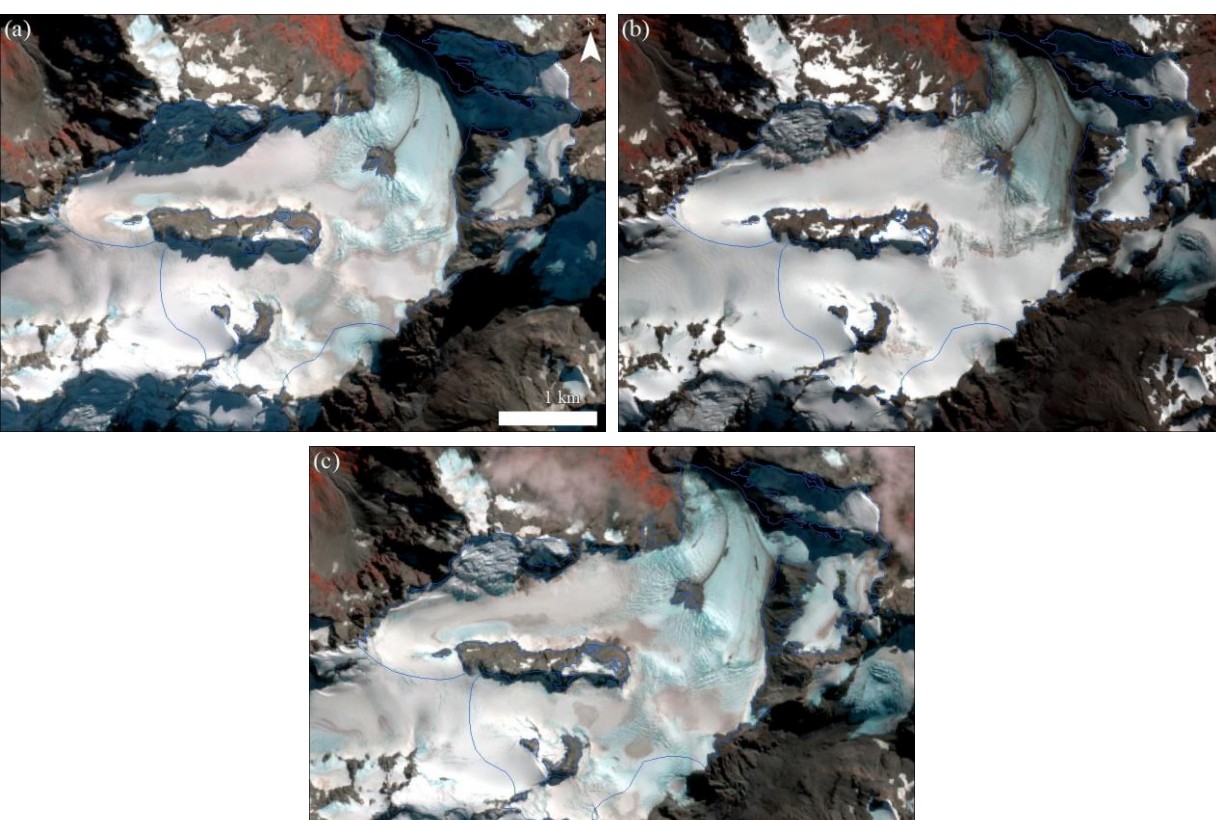


**Figure 9: Maximum elevation of the seasonal snowline (approximating the point of maximum ablation) on Lambert Glacier (outlined in blue) observable in cloud-free Sentinel-2A and 2B images from (a) 30 April 2016, (b) 29 March 2017 and (c) 29 March 2018.**


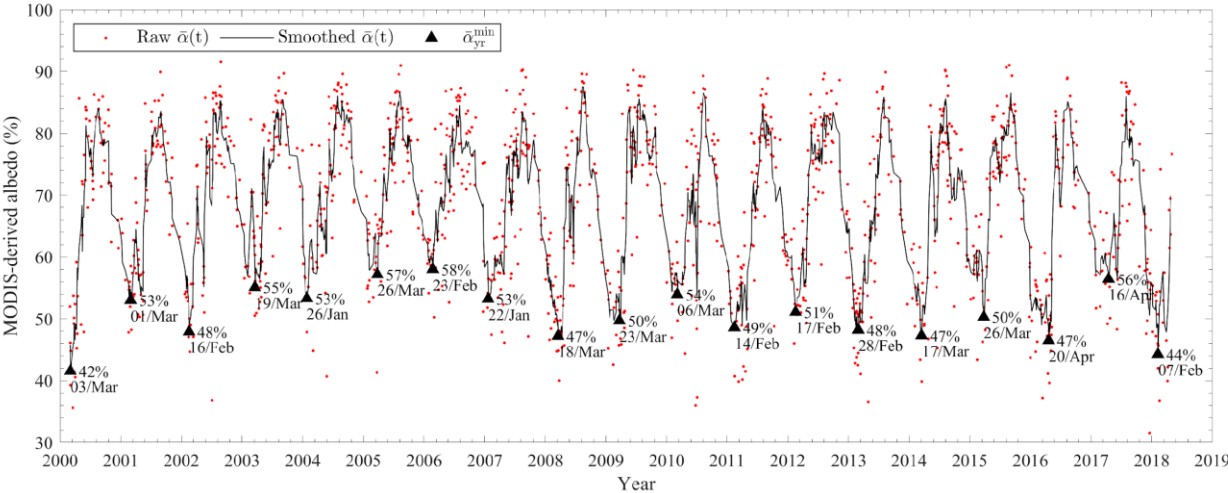

**Figure 10: MODIS-derived glacier-wide surface albedo $\bar{\alpha}(t)$ on Vertebrae Col 25 between February 2000 and April 2018, with the 3-period rolling average and annual minimum glacier-wide albedo $\bar{\alpha}_{yr}^{min}$ indicated in black.**

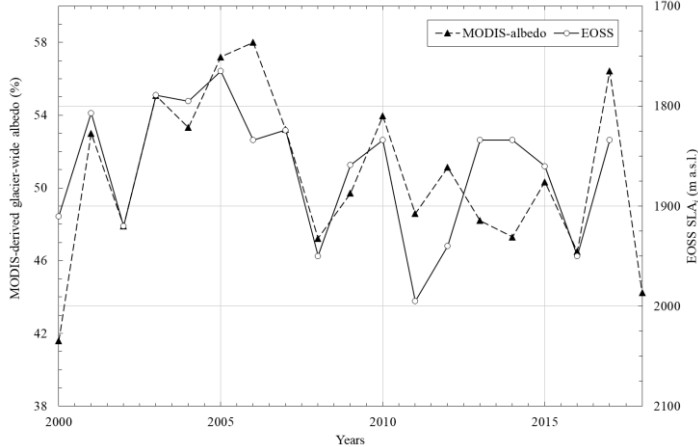

**Figure 11: Comparison of MODIS-derived $\overline{\alpha}_{yr}^{min}$ and EOSS SLA$_i$ on Vertebrae Col 25 between 2000 and 2018 ($R^2 = 0.43$).**

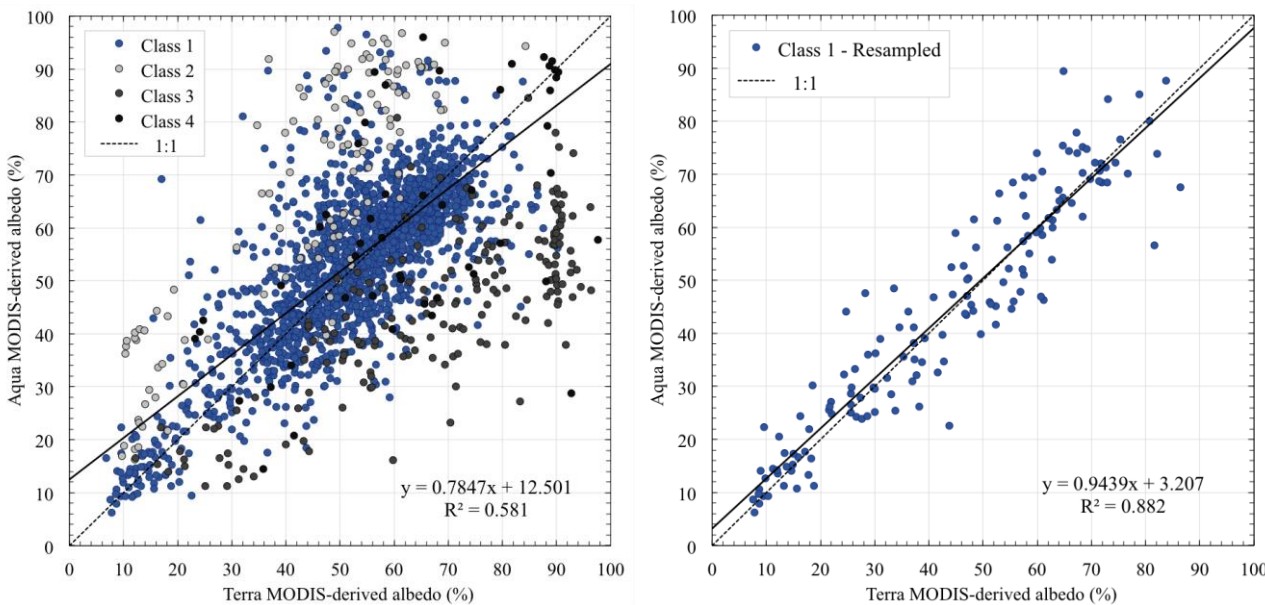


**Figure 12: Pixel-by-pixel comparison of MODIS-derived glacier surface albedo across the GoEA, from four pairs of cloud-free Terra and Aqua MODIS images (4 March 2009, 10 March 2010, 9 March 2012, 11 March 2013). Linear regression analysis was performed on (left) all Class 1 pixels ($R^2$ = 0.58, p < 0.001) and (right) a stratified random sample of 150 Class 1 pixels ($R^2$ = 0.88, p < 0.001). The dotted line represents the ideal 1:1 relationship.**


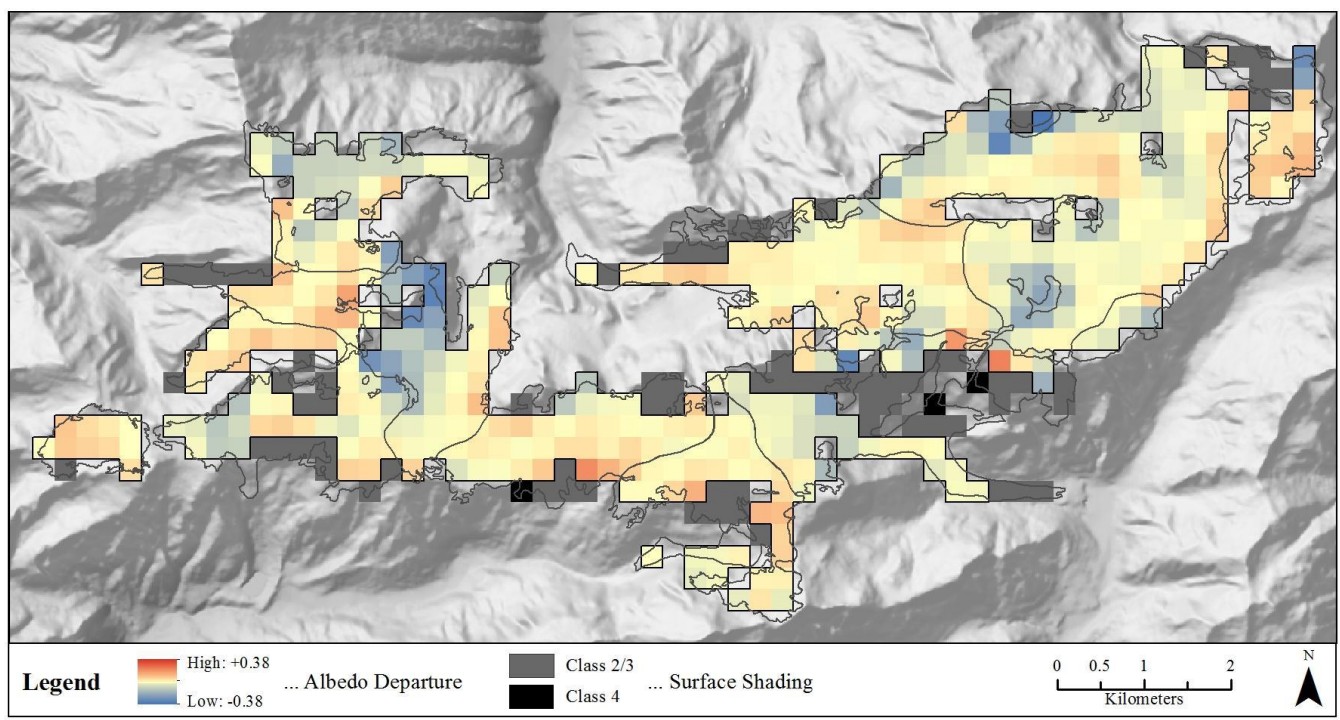

**Figure 13: Average departure of MODIS-derived glacier surface albedo for Class 1 pixels between Terra and Aqua images from the 4 March 2009, 10 March 2010, 9 March 2012 and 11 March 2013. A positive departure (red pixels) indicates a higher value of albedo in the Terra image. Pixels with shadows have been displayed in grey/black.**

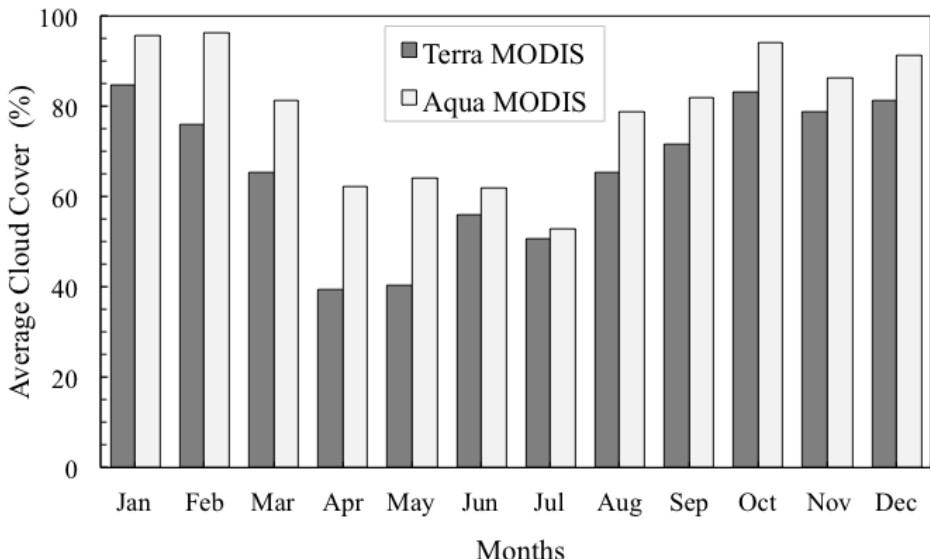

**Figure 14: Proportion of pixels over the GoEA obscured by cloud in Terra (10:30 am) and Aqua (1:30 pm) MODIS images. Monthly averages calculated from 1 January 2012 to 31 December 2012.**

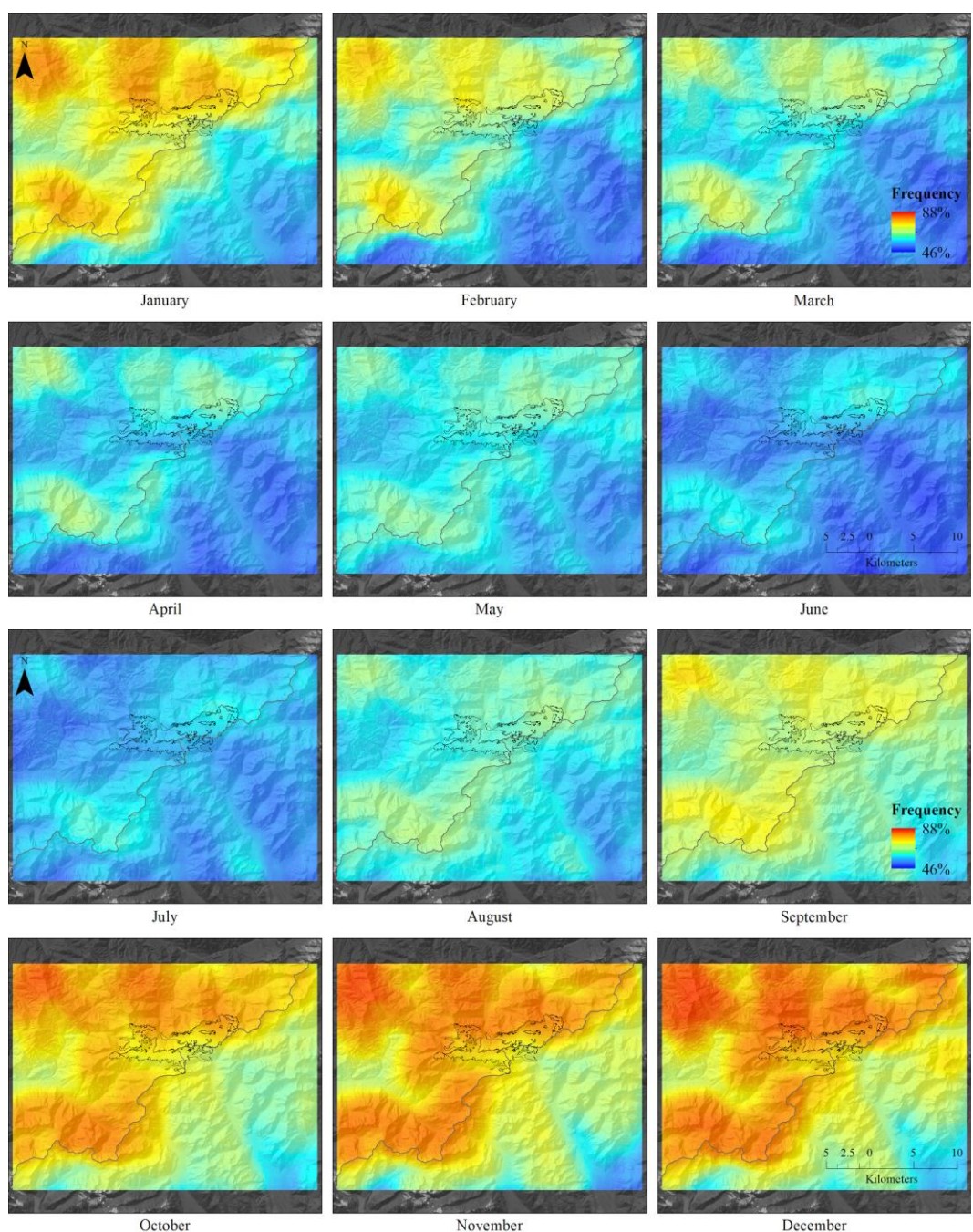

Figure 15: Frequency of monthly cloud cover per pixel in Terra MODIS images over the Southern Alps from 2000 to 2017 (frequent cloud cover is displayed in red). The Main Divide of the Southern Alps is shown running southwest to northeast, with the position of the GoEA marked in black.