# Peer review of "Variability of glacier albedo and links to annual mass balance for the Gardens of Eden and Allah, Southern Alps, New Zealand"

_The Cryosphere, 2020_

## Referee Comment (RC1) · Anonymous Referee #1 · 30 Jan 2020

This is a welcome contribution that provides details of an alternative remotely sensed method for monitoring glacier albedo as a potential mass balance proxy. Direct measurement of mass balance on mountain glaciers is resource intensive and often only provides a small number of point data that still require interpolation. Improving remote monitoring methods is essential as this will enable a more comprehensive and sustainable approach to mass balance monitoring. This is a key rationale present for this project. Increased use of remote sensing techniques is a key way scientists can reduce the carbon footprint of their climate-science. Research that progresses such techniques is timely.

[Figure]

However, despite the authors making a strong case for the benefit of remote sensing over on-site measurement it is noted in the acknowledgments (and in a media campaign) that the authors did undertake field work at this remote, protected site. The strength of this method will lie in it being able to be robustly applied to any glacier without the need for onsite calibration. So some questions arise; 1. What data was gathered at the site and was this used to tune processing? 2. Could the method calibration have been done at an already high-use glacier site (e.g. Franz, Fox, Tasman Glaciers), thereby providing more support for using RS at sensitive sites, and 3.What confidence do the authors have that this method can be applied to sensitive, protected sites without the need for onsite measurement?

Generally the paper is well written and contains sufficient background information and detailed (RS) methods section. However, the interpretation of results is compromised by inadequate information (Figure 1) of the actual glaciers used in analysis. Figure 1, the location map, does not make clear the locations of all the individual glaciers contained in the text. This becomes rather frustrating when reading results and attempting to consider them in a spatial context. Consultation of the official topographic map for the area provided some assistance, but it was still unclear exactly what ice bodies the authors were referring to for the two unnamed glaciers, and for a glacier like Colin Campbell, which has multiple branches, it is not clear what branch has been used for analysis.

The decision to separate the 12 glaciers into 3 classes would benefit from a little more explanation. For example Eve Glacier appears more topographically similar to Abel and Colin Campbell (when one makes some assumptions about which branch has been used for the Colin Campbell). It is also unclear whether statistics for elevation and slope include the upper accumulation zones or just focus on themore defined glacier trunks. Having all 12 glaciers clearly defined on a map would benefit result interpretation.

Figure 5 is a key Figure, but I found myself looking for a third panel showing the the average albedo over time, which could potentially be added to Figure 5 (right). If the

timing of the ablation minimum is getting later, does this mean that the minimum albedo is also decreasing due to a longer alation season? Or Not?

As noted above, the lack of a detailed location map hinders spatial thinking, as does the organisation of Figure 6. While clearly the authors have opted to organise both graphs in Figure 6 by the scale of the x-axis values in doing so the reader is left with no clues as to how these glaciers are actually related in space, again is there a spatial influence on the data presented? It is very difficult to compare results of the left and right graph for an individual glacier as the order of the y-axis (by giving priority to the x-axis value) are different for each graph. While it is appreciated that a 'progressional' x-axis approach might be the 'neatest' presentation, something is lost in regards to the actually physical process or characteristics that might be driving the patterns being presented. For example can something more be said about W/E or N/S trends? If one colour-codes the class sizes some patterns appear, for example class 1 glaciers tend to have lower albedo and a later minimum timing, whereas class 2 (n=2, should potentially include Eve) have higher albedo and earlier minimum timing.

While it is appreciated that this paper is 'methods' focused there is missed opportunity to engage more fully with some of the glaciological findings. In particular, the finding that the timing of the minimum albedo is occurring later in the summer, which could signal a later onset of the first winter snowfall (i.e. lengthening of the ablation season). This result also makes one wonder if there is any trend (across all the glaciers measured) of a decreasing minimum albedo over time, for if the ablation season is becoming more protracted then the snow surface would likely become more discoloured. Or alternatively, is the minimum albedo the same and the trend is associated with a later start to the ablation season (i.e. more spring snow delaying the onset of melting). It would be great to see a little more discussion of these important mass balance feedbacks.

Should the authors wish to include a reference for the Rolleston Glacier mass balance programme, they could cite Purdie, H., Rack, W., Anderson, B., Kerr, T., Chinn,

T.J., Owens, I. and Linton, M. 2015: The impact of extreme summer melt on net accumulation of an avalanche fed glacier, as determined by ground-penetrating radar. Geografiska Annaler, Series A: Physical Geography 97, 779-791.

---

## Referee Comment (RC2) · Anonymous Referee #2 · 5 Feb 2020

Manuscript Number: TC-2020-5

Title: Variability of glacier albedo and links to annual mass balance for the Gardens of Eden and Allah, Southern Alps, New Zealand

Authors: Angus J. Dowson, Pascal Sirguey, Nicolas J. Cullen

The paper proposes to use minimum MODIS albedo to approximate snow line altitudes (SLA). The paper requires a substantial effort to improve the clarity and organisation. The authors need to clearly define the objectives of the paper then use the results to support the objectives.

[Figure]

General comments: A clearly stated set of objectives is require. The authors need to put some significant thought into this as no surface mass balance (smb) records are presented. The authors need to find a way to make a better connection to smb, otherwise the analysis compares a proxy for smb to another proxy for smb. The manuscript requires substantial editing for clarity and organisation. The error analysis of MODIS albedo and the potential for sensor degradation (post 2016 on the Terra bus) need to be addressed. Please use continuous line numbers and do not restart the line numbers at every page. A series of paper citations are listed at the end of this review for you to consider. Please remove the words "important", "meaningful", "arguably", etc. from the manuscript. Let the readers decide for themselves. Julian Day should probably be referred to as Day of Year. A substantial amount of effort is required to understand the errors in SLA, and minimum albedo related to their temporal mismatch. The figures are not used to full effect in the text. A clearer description and analysis of the figures in the text is required. The word significant should be reserved to describe statistical significance, otherwise words like substantial should be used to describe a large change. If the authors are going to invoke the 50 EOSS glaciers, then a much stronger analysis of how the 12 glaciers in this study compare to the 50 is required.

Specific comments: Line 10: Using the whole glacier minimum albedo the dynamics of the glacier are somewhat smoothed over. Perhaps define what is meant by dynamics in this instance. Is it annual minimum albedo that was used, or melt season? Line 13: briefly define new approach Line 14: provide the evidence in brief Line 12: Was surface mass balance measured, or was EOSS SLA measured – the two are not the same thing? Line 16: Define "high snow line" p-value should be p<0.001, etc. depending on level of significance. The results section of the Abstract need to be rewritten to present all of the results. Furthermore, a statement of why you have done this analysis is required. Lines 26-27: Define climate units. Introduction: P2,Line 17: The minimum albedo method does not infer mass balance. It scales to ELA and AAR, which in turn scale to mass balance. Without measurements of mass balance to quantify the relationship to minimum albedo, ELA, or AAR, mass balance should not really enter into

the discussion. P2,Line 20: Define "relationships". P2,Line 20-23: There are more citations for this method – see below. P2,Line 25: why is the glacier contribution "globally significant" and define (i.e., X m sea level increase). P2,Line 29: Air temperature is important in controlling glacier mass balance – but on line 19 you said shortwave radiation played a governing role on smb. Which is it? Both I gather, so this section should be expanded to detail the nuanced nature of the controls on glacier smb. P2,Line 31: define cloud properties and how they influence smb. What is the role of longwave forcing? P2,Line 32: "global warming in the Southern Alps" is nonsensical. P3,Line 1: define "fast-responding glaciers" what does this mean and why is it important? P3,Line 3: Why was minimum albedo method not used on these two glaciers? If that anlaysis has already been conducted, then provide an analysis of ELA, AAR and minimum albedo for these two glaciers in reference to smb. Once completed then the analysis presented on the other glaciers without smb will be on a better footing. P3,Line 4: Define what was measured then tell me why it is important. P3,Line 10: There is going to be a large problem using MODIS grids on very small glaciers because of the grid size to glacier area mismatch - discuss. P3,Line 11: have meant – means P3,Line 13-21: this sounds like the missing research question that should have been addressed in the end of the Abstract. P3,Line 22-27: Provide better links to the tables and figures detailing the study site. The last paragraph of the Introduction is redundant and should be removed.

Page 4: Site description: P4,Line 20: Provide climate normals. P5,Line 3: How is this a sensitive climate situation? Sensitivity means that there is a glacier response (dynamic, or mass loss etc.) for temperature or precipitation change. A case for neither of these scenarios has been presented. P5,Line 10: How do you know that the 14 March corresponds to the minimum ELA? P5,Line 12: What is "field knowledge"? Actually the whole section should be rewritten. What are the NDSI wavelengths used? There is almost two months difference between image capture dates. How is this reconciled? P5,Line 27: What is the superscript T on MODIS? The standard usage is MOD for Terra and MYD for Aqua. P5, Section 2.2: You should read:

https://nsidc.org/data/modis/terra_aqua_differences There is some speculation that the Terra MODIS sensor is degrading. This degradation is largely correct for in the collection 6 data. There are several citations that should be considered in relation to this issue. These citations have been listed at the end of the review. P5, Line 29: orbits are usually described as ascending and descending. P6, Line 2: What does "exceptionally cloudy" mean? Provide a description with statistics. Why were only four days used for comparison between Terra and Aqua? Provide justification. The MODIS methods section does not indicate which albedo product was used, or how albedo was produced at 250 m. A much more clear description of data processing is required. O.k., I see the following section on albedo processing. Perhaps a sentence here to indicate that MODImLab will be detailed (and why) later in the paper. Section 2.3 requires a much better description of what exactly was done. P6, Line 21: Sentinel was used to "approximate" the timing and elevation of SLA. A great deal of effort should be spent on what approximate means in this instance. P8, Line 30: why was 50% used? How was this value determined? Should 100% be used if you want to remove debris-covered glaciers? Page 9, Section 3.3: The requirement of using only cloud free glaciers is very restrictive and substantially reduces the amount of data used in the analysis. Provide statistics on the amount and duration of cloud cover. When there is cloud cover will likely be as important as where there is cloud cover. Pixel is a picture element found on a display screen. Grid cell is a better usage. Page 10, L1: Artefact should be replaced by error. Section 4.3 should be presented in the Introduction section. Typical values for the different glacier states should be provided. This is an example of the systematic problem with this paper, in that there really aren't that much in terms of reportable results. Page 11, L5: The classification of glaciers into four groups is accomplished by geographical variables. Without providing climate data, it is a mystery to me how the authors determine the glaciers are not behaving as a single climate unit. Page 11, L28: Either the declines are monotonic or they are not. Page 13, L15 (and elsewhere) R^2 values should be presented with significance values. Page 13, L29-30. What is a maximum summer snow line? When exactly was this observed within the summer

period? P14,L10: Citation required. P14,L23: Using only 12 glaciers, which are argued to occupy different climactic regions, can size and elevation be reasonably discounted as predictors of the min albedo and SLA? P15,L7: define effective, precisely. P15,L23: I wouldn't have characterised the R^2 values found in the Results section as strong to moderate. Section 5.2 is results and should not be presented in the discussion section. P18,L7: "likely having as significant influence" means you don't know if it is significant or not. Section 5.3. This section should be expanded. Is the correlations between minimum albedo to SLA related to the amount of cloud cover? Figure 12 (left panel) There are obvious difference between MODIS Terra and Aqua related to Class. Why is this? Might this be related to the systematic increase in cloud cover in Aqua vs. Terra (Figure 14)?

Citations to consider:

Casey, K. A., Polashenski, C. M., Chen, J., & Tedesco, M., 2017. Impact of MODIS sensor calibration updates on Greenland ice sheet surface reflectance and albedo trends. The Cryosphere, 11, 1781–1795.

Polashenski, C.M., Dibb, J. E., Flanner, M. G., Chen, J. Y., Courville, Z. R., Lai, A. M., et al., 2015) Neither dust nor black carbon causing apparent albedo decline in Greenland's dry snow zone Implications for MODIS C5 surface reflectance. Geophys. Res. Lett., 42, 9319–9327.

Van Tricht, K. et al., 2016. Clouds enhance Greenland ice sheet meltwater runoff. Nat. Commun. 7:10266 doi: 10.1038/ncomms10266.

Williamson, S.N., Copland, L., Hik, D.S., 2016. The accuracy of satellite-derived albedo for northern alpine and glaciated land covers. Polar Sci. 10, 262-269. Bahr, D.B., Radic, V., 2012. Significant contribution to total mass from very small glaciers. The Cryosphere, 6, 763-770.

Bennartz, R., Shupe, M., Turner, D. et al., 2013. July 2012 Greenland melt extent

enhanced by low-level liquid clouds. Nature, 496, 83–86. doi:10.1038/nature12002

Zhang, Z., Jiang, L., Liu, L., Sun, Y., Wang, H., 2018. Annual Glacier-Wide Mass Balance (2000-2016) of the Interior Tibetan Plateau Reconstructed from MODIS Albedo Products. Remote Sens., 10, 1031, https://doi.org/10.3390/rs10071031.

---

## Author Comment (AC1) · 10 Jun 2020

See combined AC at the end of the discussion

---

## Author Comment (AC2) · 10 Jun 2020

See combined AC at the end of the discussion

---

## Author Comment (AC3) · 10 Jun 2020

*This is a welcome contribution that provides details of an alternative remotely sensed method for monitoring glacier albedo as a potential mass balance proxy. Direct measurement of mass balance on mountain glaciers is resource intensive and often only provides a small number of point data that still require interpolation. Improving remote monitoring methods is essential as this will enable a more comprehensive and sustainable approach to mass balance monitoring. This is a key rationale present for this project. Increased use of remote sensing techniques is a key way scientists can reduce the carbon footprint of their climate-science. Research that progresses such techniques is timely.*

**Response:** We thank the anonymous reviewer for his/her very positive review of our manuscript.

*However, despite the authors making a strong case for the benefit of remote sensing over on-site measurement it is noted in the acknowledgments (and in a media campaign) that the authors did undertake field work at this remote, protected site. The strength of this method will lie in it being able to be robustly applied to any glacier without the need for onsite calibration. So some questions arise;*

**Response:** It is true that the team completed a 2-day fieldwork trip on the study site at the end of January 2018. The data collected on this trip were not directly linked to the study of albedo presented in this paper. The paper did not suggest otherwise but, for the sake of clarification, we provide an answer to each of the referee's specific questions below:

1. *What data was gathered at the site and was this used to tune processing?*

   **Response:** As explained in the manuscript, the only data originating from the field trip that were used to inform this study were oblique aerial and terrestrial photographs that we found useful to refine glacier outlines. At the early stages of this study, glacier outlines were derived from Sentinel-2 imagery until we were fortunate enough to receive very high-resolution Pleiades imagery. Based on this, we decided to refine some glacier outlines and used photos captured during the trip to help inform this process. No data collected during the fieldwork were used to calibrate or validate the processing of MODIS albedo data.

2. *Could the method calibration have been done at an already high-use glacier site (e.g. Franz, Fox, Tasman Glaciers), thereby providing more support for using RS at sensitive sites,*

   **Response:** The albedo retrieval method presented in this study does not need field observations for calibration. The albedo retrieval implemented in Modimlab has already been validated on multiple sites around the world, including on New Zealand glaciers, as can be found in the bibliographic references provided in the manuscript. Given the principle by which the albedo is retrieved by Modimlab and its previous validation in a New Zealand glacier environment, it did not require calibration for this study. Therefore, our method can be applied to other high-use and/or sensitive glaciers sites.

*and 3. What confidence do the authors have that this method can be applied to sensitive, protected sites without the need for onsite measurement?*

**Response:** As noted above, this study did not require site-specific calibration, and no onsite measurements were used other than some additional photos to help refine a number of specific glacier outlines. The photos and on-site field observations were of course very helpful in our efforts to refine the glacier outlines from a quality standpoint. Nonetheless, they were not essential to the method per say. Given the relatively coarse resolution of MODIS data, the interpretation of glacier outlines from high to very high resolution imagery remains sufficient to derive the albedo series. Importantly, there is no significant improvement on the albedo retrieval solely arising from the use of the photos. Given the new insight this study reveals about a poorly documented protected site, we are now much more confident that this method can be applied more widely. This confidence is matched by a reviewer of one of the validation studies of this method, who noted this "*powerful observation-based approach begs to be applied elsewhere*" (Interactive comment on The Cryosphere Discuss., doi:10.5194/tc-2016-98, 2016).

Please note that as we addressed this comment we noted a mistake in the reported date of the fieldwork in the original manuscript, which has been corrected (27-28 January 2018 instead of 20-21 January 2018.)

*Generally the paper is well written and contains sufficient background information and detailed (RS) methods section. However, the interpretation of results is compromised by inadequate information (Figure 1) of the actual glaciers used in analysis. Figure 1, the location map, does not make clear the locations of all the individual glaciers contained in the text. This becomes rather frustrating when reading results and attempting to consider them in a spatial context. Consultation of the official topographic map for the area provided some assistance, but it was still unclear exactly what ice bodies the authors were referring to for the two unnamed glaciers, and for a glacier like Colin Campbell, which has multiple branches, it is not clear what branch has been used for analysis.*

**Response:** We thank the reviewer for this constructive comment. We agree that Figure 1 is a good place to show the names of each glacier mentioned in the study. We have modified Figure 1 and added a panel showing the glacier outlines and names, and also display the cluster to which they belong.

*The decision to separate the 12 glaciers into 3 classes would benefit from a little more explanation. For example Eve Glacier appears more topographically similar to Abel and Colin Campbell (when one makes some assumptions about which branch has been used for the Colin Campbell).*

**Response:** We have clarified how we determine the 3 glacier classes in Section 4.2, which is based on the interpretation of glacier characteristics from clusters identified in scatter plots. Those scatter plots are reproduced below for the reviewer and other readers (class 1: green; class 2: red; class 3: blue). While such a classification involves a trade-off between generalisation and retaining some formal structure, we believe there is enough evidence to justify having three classes. The updated Figure 1 and scatter plots below also provide clear evidence that the class of Eve Glacier is appropriate.

[Figure]

*It is also unclear whether statistics for elevation and slope include the upper accumulation zones or just focus on the more defined glacier trunks.*

**Response:** We have clarified that the statistics for the features used to support the classification are "derived from the mapped outlines of each glacier".

*Having all 12 glaciers clearly defined on a map would benefit result interpretation.*

**Response:** We modified Figure 1 to display glacier outlines and names as well as clusters.

*Figure 5 is a key Figure, but I found myself looking for a third panel showing the average albedo over time, which could potentially be added to Figure 5 (right). If the timing of the ablation minimum is getting later, does this mean that the minimum albedo is also decreasing due to a longer alation season? Or Not?*

**Response:** We thank the reviewer for this constructive comment. We added a panel showing the evolution of the minimum glacier-wide minimum albedo alongside the evolution of its timing. As we revised this figure, we decided to also display yearly values to help visualise the variability better, while maintaining the 3-year rolling average as a dotted line. Furthermore, we added a scatter plot that shows the relationship between $\alpha_{yr}^{min}$ and its timing and assessed its correlation to inform the hypothesis of the reviewer. We find a weak correlation when all clusters are considered together ($R^2$=0.26, p<0.001) and an even weaker correlation when all glaciers are considered separately ($R^2$=0.11, p<0.001). However, within an individual cluster, the value of $\alpha_{yr}^{min}$ does not exhibit a significant correlation with its timing (cluster 1: $R^2$=0.2, p=0.058; cluster 2: $R^2$=0.08, p=0.248; cluster 3: $R^2$=0.03, p=0.472). We have added some further insights in Section 4.4 to report on this new result, updated references to this figure in the text where appropriate, and added a glaciological interpretation and significance of these findings in a new discussion section (5.2 Implications of variability of albedo on mass balance).

*As noted above, the lack of a detailed location map hinders spatial thinking, as does the organisation of Figure 6. While clearly the authors have opted to organise both graphs in Figure 6 by the scale of the x-axis values in doing so the reader is left with no clues as to how these glaciers are actually related in space, again is there a spatial influence on the data presented? It is very difficult to compare results of the left and right graph for an individual glacier as the order of the y-axis (by giving priority to the x-axis value) are different for each graph. While it is appreciated that a 'progressional' x-axis approach might be the 'neatest' presentation, something is lost in regards to the actually physical process or characteristics that might be driving the patterns being presented. For example can something more be said about W/E or N/S trends? If one colour-codes the class sizes some patterns appear, for example class 1 glaciers tend to have lower albedo and a later minimum timing, whereas class 2 (n=2, should potentially include Eve) have higher albedo and earlier minimum timing.*

**Response:** Thanks for the constructive comment. We found the suggestion of colour coding glaciers by class in Figure 6 very helpful and well-thought out. Section 4.4 in the original manuscript did characterize the contrast in magnitude and timing of minimum albedo between glaciers of class 1 and 2 that now becomes more obvious with the colour coding. The reference to Fig. 6 has also been added to the point made in Section 4.4. As noted above, we have added a new section in the discussion that adds to the interpretation of our results by assessing how topography modulates the response and possible fate of the GoEA glaciers.

*While it is appreciated that this paper is 'methods' focused there is missed opportunity to engage more fully with some of the glaciological findings.*

**Response:** Our paper makes use of the albedo method to produce a comprehensive set of new glaciological observations in an area where very little data exist. The principles of this method are described for completeness, yet this section remains limited to echoing the body of literature where it was developed and validated. In particular, Sirguey et al. (2016) provides a very useful '*methods focused paper*' that is in part focused on describing, calibrating and validating the albedo method.

In contrast, this study is very much an application of the method rather than a paper about the method. It provides a large amount of new glaciological observations unavailable to date, and characterizes glacier behaviour of largely undocumented icefields. It is true that this new application of the albedo method provides an opportunity to incrementally refine it, for example by testing the use of glacier outlines to facilitate masking and the opportunity to use MODIS Aqua data. Nevertheless, results associated with these "methodological" steps only involve three out of the now eleven sections of the results and discussion. All other results present new insights about the behaviour of glaciers in the study area. This paper also provides a discussion of the implications of the new findings regarding the variability of the glacier surface compared to results from the EOSS methodology traditionally used in New Zealand. We have also explored the validity of the "unified climatic unit" theory about the response of New Zealand glaciers. In response to comments from both reviewers, we have added a new discussion section (5.2 Implications of variability of albedo on mass balance) that we hope builds knowledge about the glaciology of this remote but important region in the Southern Alps.

*In particular, the finding that the timing of the minimum albedo is occurring later in the summer, which could signal a later onset of the first winter snowfall (i.e. lengthening of the ablation season). This result also makes one wonder if there is any*

*trend (across all the glaciers measured) of a decreasing minimum albedo over time, for if the ablation season is becoming more protracted then the snow surface would likely become more discoloured. Or alternatively, is the minimum albedo the same and the trend is associated with a later start to the ablation season (i.e. more spring snow delaying the onset of melting). It would be great to see a little more discussion of these important mass balance feedbacks.*

**Response:** We thank the reviewer for this input. We found the suggestion very useful as a prompt for us to explore this issue more deeply as our data challenges this hypothesis. As demonstrated above, the comparison between the magnitude of $\underline{\alpha}_{yr}^{min}$ and its timing shows that a delayed minimum albedo does not necessarily lead to a lower albedo. We have added elements of our interpretation of this result in the new discussion section.

*Should the authors wish to include a reference for the Rolleston Glacier mass balance programme, they could cite Purdie, H., Rack, W., Anderson, B., Kerr, T., Chinn, T.J., Owens, I. and Linton, M. 2015: The impact of extreme summer melt on net accumulation of an avalanche fed glacier, as determined by ground-penetrating radar. Geografiska Annaler, Series A: Physical Geography 97, 779-791.*

**Response:** We thank the reviewer for this suggestion and have added the citation to support the statement made in the introduction about the Rolleston Glacier mass balance programme.

**RC2: 'Interactive comment on "Variability of glacier albedo and links to annual mass balance for the Gardens of Eden and Allah, Southern Alps, New Zealand" by Angus J. Dowson et al.', Anonymous Referee #2, 05 February 2020**

*The paper proposes to use minimum MODIS albedo to approximate snow line altitudes (SLA). The paper requires a substantial effort to improve the clarity and organisation. The authors need to clearly define the objectives of the paper then use the results to support the objectives.*

**Response:** We thank the reviewer for his/her comprehensive review of our manuscript.

*General comments: A clearly stated set of objectives is require. The authors need to put some significant thought into this as no surface mass balance (smb) records are presented. The authors need to find a way to make a better connection to smb, otherwise the analysis compares a proxy for smb to another proxy for smb.\*

**Response:** We have refined the aim and provided three specific objectives to meet this aim. We have clarified how this research provides another important step towards developing a robust and comprehensive approach to remotely monitoring glaciers in the Southern Alps. We have clarified in the introduction and elsewhere in the manuscript how the albedo method can be used, even when not calibrated, to assess the spatial and temporal variability of glacier mass balance. Finally, in the context of having limited direct measurements of mass balance in the Southern Alps, the albedo method provides the opportunity of having a second proxy to supplement the 40-year record of the EOSS programme. There is merit in making this comparison as it produces new insights and enabled us to revisit an important conclusion inferred from the EOSS programme.

*The manuscript requires substantial editing for clarity and organisation.*

**Response:** We have made major revisions of the manuscript. The changes we have made, as suggested by the reviewers, have improved the clarity and organisation of the manuscript as detailed in our response to each comment and the track-changes version of the new manuscript.

*The error analysis of MODIS albedo and the potential for sensor degradation (post 2016 on the Terra bus) need to be addressed.*

**Response:** The reviewer reiterated this point in another specific comment – please see our full response below.

*Please use continuous line numbers and do not restart the line numbers at every page.*

**Response:** Done.

*A series of paper citations are listed at the end of this review for you to consider.*

**Response:** We thank the reviewer for those suggestions. We used some of these citations in the revised manuscript as indicated in our response to the reviewer's comments.

*Please remove the words "important", "meaningful", "arguably", etc. from the manuscript. Let the readers decide for themselves.*

**Response:** Accepted, we have removed or replaced all instances of the words.

*Julian Day should probably be referred to as Day of Year.*

**Response:** We appreciate that Julian Day is initially defined as a datation system with an origin on 1 January 4713 BC. However, it is generally accepted that in context, a Julian Day is often understood and unambiguously defined as the number of days since an alternative origin. For example, CNES uses 1 January 1950, while NASA uses 24 May 1968 to define their Julian *date*. In fact, without historical context, Julian Day is often used to refer to the *day of year* number as we use it here. NASA provides a Julian Day Calendar that can be found at https://landweb.modaps.eosdis.nasa.gov/browse/calendar.html. In

view of this, we don't think that our use of "Julian Day" creates an ambiguity that warrants a change as suggested by the reviewer.

*A substantial amount of effort is required to understand the errors in SLA, and minimum albedo related to their temporal mismatch. The figures are not used to full effect in the text. A clearer description and analysis of the figures in the text is required.*

**Response:** We clarified the methodology used by the EOSS program to derive SLA and expanded Section 4.6 with reference to Figures 8, 9 & 10 to stress this point, as well as providing a new discussion section that reflects on these differences. We hope these substantial revisions have provided more clarity.

*The word significant should be reserved to describe statistical significance, otherwise words like substantial should be used to describe a large change.*

**Response:** We agree with the reviewer about the use of "significant". We replaced the word "significant" if it was not being used to describe, or supported by, a statistical test, except when the word was unambiguously referring to its alternate meaning as *"indicative of something sufficiently great or important to be worthy of attention"*.

*If the authors are going to invoke the 50 EOSS glaciers, then a much stronger analysis of how the 12 glaciers in this study compare to the 50 is required.*

**Response:** We have updated and provided a detailed comparison between the response of the glaciers on the GoEA and the EOSS observations from the same region. See Sections 4.6-7 and the discussion in Section 5.1 for these analyses.

*Specific comments:*

*Line 10: Using the whole glacier minimum albedo the dynamics of the glacier are somewhat smoothed over. Perhaps define what is meant by dynamics in this instance. Is it annual minimum albedo that was used, or melt season?*

**Response:** We thank the reviewer for this input. The use of the word *dynamics* was ambiguous as it suggests mechanical behaviour. We implied a more general definition of *dynamics,* namely "*a characteristic in relation to the constant change of a process or a system*". This has been removed as part of our reworking of the abstract.

*Line 13: briefly define new approach*

**Response:** We believe that describing the approach we tested and used for masking would be unnecessarily complicated and lengthen the abstract. We decided to remove this part of the sentence.

*Line 14: provide the evidence in brief*

**Response:** We rephrased the sentence to clarify this.

*Line 12: Was surface mass balance measured, or was EOSS SLA measured – the two are not the same thing?*

**Response:** We have refined and updated the abstract. Surface mass balance was not measured in this study. This sentence referred to the fact that the albedo method used in this study has been shown in earlier work to be an efficient proxy for glacier mass balance. The question raised by the reviewer is legitimate as the sentence may have been misleading given we are not in a position to provide a quantitative estimate, as there has not been a calibration of mass balance in this study.

*Line 16: Define "high snow line"*

**Response:** We added that "high snow line altitude" is defined as "relative to the long-term average"

*p-value should be p<0.001, etc. depending on level of significance.*

**Response:** Accepted, we expressed the actual p value unless p<0.001, in which case we only used the statement of inequality.

*The results section of the Abstract need to be rewritten to present all of the results.*

**Response:** We have refined and updated the abstract.

*Furthermore, a statement of why you have done this analysis is required.*

**Response:** This comment echoes the need for clearer objectives that was highlighted in the first general comment. We have modified the abstract and other parts of the manuscript to ensure the aim and objectives of this research are more explicit.

*Lines 26-27: Define climate units.*

**Response:** We are referring to the theory of the Southern Alps glaciers behaving as a "*unified climatic unit*" as proposed by Chinn et al. (2012). This is one of the main outcomes of the EOSS monitoring programme inferred from the high degree of intra-correlation found in the variability of snowline elevation between any index glacier and the average of the group. In the introduction where the term "*unified climatic unit*" is directly quoted from Chinn et al. (2012), we have clarified that this is referring to a "consistent response to climate variability inferred from the EOSS record". We have also reworded the abstract to clarify the origins and meaning of this term.

*Introduction:*

*P2,Line 17: The minimum albedo method does not infer mass balance. It scales to ELA and AAR, which in turn scale to mass balance. Without measurements of mass balance to quantify the relationship to minimum albedo, ELA, or AAR, mass balance should not really enter into the discussion.*

**Response:** We agree that the word "infer" was not appropriate in this sentence. We replaced it by "estimate". We however respectfully disagree with the reviewer that mass balance should not enter into the discussion. This paragraph is introducing the general concepts of remote sensing approaches for the very purpose of capturing glacier mass balance, whether its relative variability, or its absolute value when calibration data can be provided. The statement made remains factually correct in view of the supporting literature provided in the paragraph.

*P2,Line 20: Define "relationships".*

**Response:** We have rephrased this sentence as "The annual minimum glacier-wide albedo ($\alpha_{yr}^{min}$) retrieved from MODIS imagery has been found to scale to annual glacier mass balance…"

*P2,Line 20-23: There are more citations for this method – see below.*

**Response:** We thank the reviewer for these suggestions. Among those, we find that only Zhang et al. (2018) is directly relevant and added it to the Introduction section.

*P2,Line 25: why is the glacier contribution "globally significant" and define (i.e., X m sea level increase).*

**Response:** We must stress to the reviewer that the meaning and definition of *significant* is not unique and solely reserved to characterise a quantitative value subject to statistical significance testing. In context, *significant* means "*Sufficiently great or important to be worthy of attention; noteworthy*" (Oxford Dictionary). As such, New Zealand is invariably represented in global studies as its own glacier region. In view of this we maintain that our statement that New Zealand Glaciers "*are regarded as globally significant*" is factually correct and has adequate context to suggest the meaning of significant in this sentence as "*noteworthy*". The alternative would be that New Zealand glaciers would be otherwise ignored in such global studies, which is clearly not the case. We respectfully disagree with the reviewer that using *significant* in this context necessarily calls for a specific quantified statement.

*P2,Line 29: Air temperature is important in controlling glacier mass balance – but on line 19 you said shortwave radiation played a governing role on smb. Which is it? Both I gather, so this section should be expanded to detail the nuanced nature of the controls on glacier smb.*

**Response:** We have changed the wording in this paragraph to ensure readers understand the relative importance of solar radiation, air temperature and precipitation. Our review of these governing processes are consistent to those that are well established in the literature.

*P2,Line 31: define cloud properties and how they influence smb. What is the role of longwave forcing?*

**Response:** We have provided a benchmark reference for readers and return to the importance of clouds in the results and discussion.

*P2,Line 32: "global warming in the Southern Alps" is nonsensical.*

**Response:** We have changed the wording of this sentence. However, we respectfully disagree with the reviewer about this comment. In this paragraph we clearly provide context in reference to the findings of Mackintosh et al. (2017). The title and content of the latter is unambiguous about the regional impacts of a "*period of global warming*" on New Zealand. The abstract itself reads "*The exceptional terminus advance of some glaciers during recent global warming is thought to relate to locally specific climate conditions, such as increased precipitation.*" As such, we maintain that our statement is factually correct in reference to the point being made to the supporting literature.

*P3,Line 1: define "fast-responding glaciers" what does this mean and why is it important?*

**Response:** This is again a context statement that we believe unambiguously paraphrases Mackintosh et al. (2017) where the reviewer can read "*This combination of large mass balance gradients and steep, relatively thin and fast-moving ice makes them adjust rapidly to climate forcing. Very few glaciers on Earth are capable of responding this quickly.*"

*P3,Line 3: Why was minimum albedo method not used on these two glaciers? If that anlaysis has already been conducted, then provide an analysis of ELA, AAR and minimum albedo for these two glaciers in reference to smb. Once completed then the analysis presented on the other glaciers without smb will be on a better footing.*

**Response:** We provide information two paragraphs earlier that the albedo method has been used and related to glacier mass balance in the Southern Alps, with citations to relevant work by Sirguey et al. (2016) and Rabatel et al. (2017). In order to further address this, comment, we have specifically named the glaciers on which the albedo method was used, namely Brewster Glacier (direct calibration with in-situ mass balance) and Park Pass Glacier (indirect assessment via correlation to EOSS record, itself scaling to mass balance) to ensure readers are aware that this method has been used on glaciers in New Zealand. We hope readers who require further information will consult the cited literature. We have also provided further details about how the albedo method was calibrated using in situ mass balance data from Brewster Glacier. The albedo method has not been used on Rolleston Glacier as it is too small to be captured by MODIS imagery (0.11 km$^2$ in 2012), with the area and reasoning provided.

*P3,Line 4: Define what was measured then tell me why it is important.*

**Response:** We reworked this paragraph to state specifically that the mass balance campaign on Ivory Glacier in the 1970s was the first comprehensive glaciological mass balance study conducted in New Zealand's Southern Alps. We removed the word "important" as part of addressing an earlier comment from the reviewer.

*P3,Line 10: There is going to be a large problem using MODIS grids on very small glaciers because of the grid size to glacier area mismatch - discuss.*

**Response:** The reviewer is right about this limitation to using MODIS for the albedo method. We did stress this point in Section 4.7 of the initial manuscript with a supporting citation. Noting that albedo is mapped with MODImLab at the improved 250m resolution, Davaze et al. (2018) show that $\alpha_{yr}^{min}$ could be successfully retrieved for glaciers down to 0.5 km$^2$ (7 pixels) with good correlation to annual mass balance. The agreement we obtained between $\alpha_{yr}^{min}$ and EOSS using the 0.7 km$^2$ Vertebrae Col 25 is consistent with this assessment. In response to an earlier comment, we emphasised this limitation when introducing

Rolleston Glacier. In order to address the reviewer's comment further, we also added a discussion about this limitation in Section 5.3.

*P3,Line 11: have meant – means*

**Response:** We have modified the sentence.

*P3,Line 13-21: this sounds like the missing research question that should have been addressed in the end of the Abstract.*

**Response:** This sentence does provide in part the background and motivation for the aim and three objectives that we have refined as part of our revision of the manuscript.

*P3,Line 22-27: Provide better links to the tables and figures detailing the study site. The last paragraph of the Introduction is redundant and should be removed.*

**Response:** We added a reference to Figure 1. We accepted the suggestion of the reviewer to remove the last paragraph of the introduction.

*Page 4: Site description: P4,Line 20: Provide climate normals.*

**Response:** Accepted. Marcara (2017) estimated a long-term mean annual rainfall surface for New Zealand for the period 1972-2016 from which we retrieved a range for precipitation over the area. We used normal temperature at surrounding weather stations over the period 1980-2010 and a lapse rate of -5.7°C km$^{-1}$ determined from the data to interpolate a mean normal temperature surface over the area, and a range over the glaciated region. We stress however that this estimate remains only indicative given the distance of the GoEA from established weather stations (closest weather station on the West Coast is 22km away, while the nearest to the east is >60km) and the uncertainty associated with the interpolation.

*P5,Line 3: How is this a sensitive climate situation? Sensitivity means that there is a glacier response (dynamic, or mass loss etc.) for temperature or precipitation change. A case for neither of these scenarios has been presented.*

**Response:** We have removed reference to the "sensitive climate situation" to avoid confusion.

*P5,Line 10: How do you know that the 14 March corresponds to the minimum ELA?*

**Response:** The point the reviewer is raising is not clear, nor what she/he means by "minimum ELA" in this context. We understand ELA stands for Equilibrium Line Altitude, which has meaning in surface mass balance. Braithwaite & Raper (2009) define "*minimum ELA* values corresponding to balance years with *highly (...) positive mass balances*". In view of this, we believe the reviewer's use of "minimum ELA" seems misplaced and makes a connection between ELA and glacier outlines that is irrelevant in our context. Nonetheless, we assume the reviewer is trying to say that glacier outlines should be derived from imagery timed precisely at the maximum ablation. If so, we dispute this strict requirement because glacier outlines are not a direct consequence of the surface mass balance for that year due to glacier response time. In practice, one aims for late summer imagery mostly because it corresponds to the melting of as much transient snow as possible to facilitate interpretation of glacier boundaries. At the same time, tradeoffs must be made given the availability of imagery, and mapping glacier outlines remains often heavily reliant on photo interpretation and experience. We indicated that we used the NDSI from the S2 image on 14 March as a base to derive glacier outlines. We then refined them manually (e.g., NDSI cannot map debris-covered parts and transient snow needs to be removed) using the additional data mentioned in the text. We believe this is a very common and valid approach, yet it is not within the scope of this manuscript to discuss this in more detail. For the reviewer's information, the Sentinel-2 image from 30 April 2016 shown in Figure 9 in the original manuscript would be closer to the end of ablation and the latest before winter snowfall that year. On the one hand it exhibits less residual snow, which may help. On the other hand, it can be seen in the comparison below, how topographic shading affects the image on 30 April compared to 14 March. Mapping solely from this image is not making the task easier nor more accurate. The NDSI is sensitive to shadow with high values in the shade greatly complicating the snow/ice segmentation. We maintain that the mid-March image is a better compromise despite the remaining snow patches, to derive suitable glacier outlines for this study. The consistency between snow patches with the late April image and across a few years with other Sentinel images (e.g., those shown in Fig. 9) and Pleiades 2017 images help discriminate them from glaciers at the refinement stage.

Braithwaite, R. & Raper, S. (2009). Estimating equilibrium-line altitude (ELA) from glacier inventory data. Annals of Glaciology 50 (53), 127--132. (Doi: 10.3189/172756410790595930.)

[Figure]

| 14 March 2016 | 30 April 2016 |
|---|---|
| False colour infrared (bands 8-4-3) | False colour infrared (bands 8-4-3) |
| NDSI (B3-B11)/(B3+B11) stretched between 0 (black) and 1 (white) | NDSI (B3-B11)/(B3+B11) stretched between 0 (black) and 1 (white) |

*P5,Line 12: What is "field knowledge"? Actually the whole section should be rewritten. What are the NDSI wavelengths used?*

**Response:** We have modified this section and specified the wavelength used for the NDSI. As explained, we took advantage of a two-day field trip to take close-up aerial photos of the glacier. This provided us with valuable direct evidence of this remote area that supplemented our interpretation of satellite imagery used to refine some glacier outlines. To avoid any confusion, we removed "field knowledge" from the revised manuscript.

*There is almost two months difference between image capture dates. How is this reconciled?*

**Response:** We have reworked this section to ensure there is no confusion for readers, and we hope this addresses the reviewer's concerns about our approach to mapping glacier outlines in this study. To respond to the specific question, we acknowledge that we did not specifically mention in the manuscript that we used the 29 April 2016 image to refine our mapping, which we understand the reviewer is referring rather than the 14 March 2016 image. It should also be noted that we used 0.5m imagery from Pleiades in March 2017, as well as photos from our field work in 2018. Thus, we compiled images across two years to help interpret glacier outlines from the March 2016 Sentinel-2 imagery. For example, the Pleiades imagery and high resolution surface supported much better interpretation of debris-covered glaciers. It is also important to keep in mind that the glacier boundaries used for this study are derived to extract 250-m resolution pixels from the MODIS albedo map and that glaciers are not expected to retreat hundreds of meters per year in this region (it would need calving in terminal lakes to achieve such rates). The mapping also compounds uncertainties in the order of tens of meters due to the geolocation of Sentinel-2 imagery and the variable interpretation at pixel level. In this regard, we don't believe the two-year period between these sources of information and our approach to mapping are any cause for concern. Further evidence of the benefit of exploiting all of these data are shown below, which provides a sequence of the terminus of Lambert Glacier from 14 March 2016, 30 April 2016 (Sentinel-2), and March 2017 (Pleiades).

[Figure]

[Figure]

[Figure]

| Sentinel-2 14 March 2016 | Sentinel-2 30 April 2016 | Pleiades-1B 10 March 2017 |

*P5,Line 27: What is the superscript T on MODIS? The standard usage is MOD for Terra and MYD for Aqua.*

**Response:** T (resp. A) superscripts indeed stands for MODIS Terra (resp. Aqua). We believed this was self-explanatory. We are aware of the MOD/MYD convention for naming MODIS products, and we did refer to this usage in P6L9 of the original manuscript. In context, when clearly referring to either sensor rather than the products from them, we do not believe that using the acronyms of MOD or MYD are appropriate. We therefore maintain our use of the superscript to refer to either one or the other sensor, and clarified this convention in Section 2.2 to avoid confusion.

*P5, Section 2.2: You should read: https://nsidc.org/data/modis/terra_aqua_differences There is some speculation that the Terra MODIS sensor is degrading. This degradation is largely correct for in the collection 6 data. There are several citations that should be considered in relation to this issue. These citations have been listed at the end of the review.*

**Response:** We thank the reviewer for this reading suggestion. This also echoes the general comment made earlier about "*The error analysis of MODIS albedo and the potential for sensor degradation (post 2016 on the Terra bus) need to be addressed.*"

We have provided further information about MODIS calibration issues in Section 4.8.1. However, we raise two important points about this issue in response to the reviewer's comments.

First, the main point being made on webpage given by the reviewer is about the differences in processing MODIS snow products between Terra and Aqua sensors due to the band 6 failed detectors of Aqua. There is no mention about the degradation of MODIS detectors. We are very aware of the differences in processing MOD10/MTD10 C6 products. However, these are not relevant for our study as we are not relying on MOD10 data. Instead, we complete all processing and albedo retrieval directly from L1B data, as is clearly explained in the manuscript, and in even more detail in the cited literature about our processing technique. As far as we know, the Quantitative Image Restoration (QIR) technique to restore Aqua band 6 is only used in production of MYD10 products but not in the production of the MOD03/MOD02 C6 products we are using. Thus, it does not apply to our data (see MODIS L1B Product User's Guide for L1B Version 6.2.2 (Terra) and Version 6.2.1 (Aqua): https://mcst.gsfc.nasa.gov/sites/default/files/file_attachments/M1054E_PUG_2017_0901_V6.2.2_Terra_V6.2.1_Aqua.pdf).

Please also note that we have acknowledged the failed detector of band 6 and demonstrate that it does not significantly affect our retrieval, which we believe is a valuable outcome of our study. This was expected as band 6 in the SWIR is an absorption band for snow and ice, and therefore contributes only marginally to the albedo signal.

Second, we are aware of the sensor calibration degradation in MODIS C5 data as demonstrated by Lyapustin et al. (2014). The latter concludes very clearly that "*The new C6 calibration approach removes major calibrations trends in MODIS Level 1B data.*". From the citations suggested by the reviewer, Casey et al. (2017) and Polashenski et al. (2015) both point out and document calibration issues with C5 data and conclude that the use of C6 data is preferable, if not necessary. Sayer et al. (2015) also confirmed the better performance of the C6 calibration. The reviewer points out that the degradation is largely corrected for in the C6 data. As we use the C6 data, which are the highest quality data available and do not have any severe calibration issues, we are of the view that further justification for using them is not necessary.

In this context, we shall stress Figure 8 and 10 to the reviewer which show albedo time series. The reviewer can assess a lack of visible trends in winter albedo which we believe should lift concerns about the existence (speculated or not) of any major

calibration issue with C6 affecting our study and results. The alternative hypothesis would be that winter albedo has a trend that is precisely matched by a calibration issue. We do not believe that this hypothesis is likely.

Lyapustin, A., Wang, Y., Xiong, X., Meister, G., Platnick, S., Levy, R., Franz, B., Korkin, S., Hilker, T., Tucker, J., Hall, F., Sellers, P., Wu, A. & Angal, A. (2014). Science impact of MODIS C5 calibration degradation and C6+ improvements. Atmospheric Measurement Techniques 7 (12), 4353--4565. (Doi: 10.5194/amt-7-4353-2014.)

Sayer, A. M., Hsu, N. C., Bettenhausen, C., Jeong, M.-J. & Meister, G. (2015). Effect of MODIS Terra radiometric calibration improvements on Collection 6 Deep Blue aerosol products: Validation and Terra/Aqua consistency. Journal of Geophysical Research: Atmospheres 120 (23). (Doi: 10.1002/2015jd023878.)

*P5, Line 29: orbits are usually described as ascending and descending.*

**Response:** Agreed, we modified the revised manuscript accordingly.

*P6, Line 2: What does "exceptionally cloudy" mean? Provide a description with statistics.*

**Response:** We replaced "exceptionally" by "extraordinary" as described by Wardle (1986), and we reworked the sentence to make the origin of this statement unambiguous and invite the reader to consult the citation for more information. We also agreed with the reviewer and added specific statistics from Wardle (1986) that are specific to two sites surrounding our region of study. It should be noted that Cropp River, west from the GoEA, exhibits the most frequent cloud cover of all sites examined by Wardle (1986).

*Why were only four days used for comparison between Terra and Aqua? Provide justification.*

**Response:** We clarified that in addition to clear sky acquisition on the same day, such comparison needed similar sensor-zenith to minimise the effect of contrasting panoramic distortion on pixel footprint. As our assessment of cloud cover below demonstrates, there are not many opportunities to get clear-sky conditions for both Terra and Aqua. This opportunity is further reduced when adding the limitation of sensor zenith. We also stated the number of samples provided by the four images amounts to 2196 pixel pairs, which we believe is a sufficient sample size to complete this comparison.

*The MODIS methods section does not indicate which albedo product was used, or how albedo was produced at 250 m. A much more clear description of data processing is required. O.k., I see the following section on albedo processing. Perhaps a sentence here to indicate that MODImLab will be detailed (and why) later in the paper.*

**Response:** We added a sentence in the previous paragraph that the albedo retrieval is described in the methods section.

*Section 2.3 requires a much better description of what exactly was done.*

**Response:** This comment suggests to us the reviewer is assuming we completed the mapping of the EOSS snowlines and generated the EOSS SLA records for Vertebrea Col 25 as part of this study. This is not the case. The EOSS programme is a national programme run by the National Institute of Water and Atmospheric Research or NIWA. An annual report with SLA values for each index glacier is normally published sometime after the observations are taken. Appendices about the method used to map SLA on each year are also provided on request. They indicate what method was used to derive SLA for any particular year. We have clarified that the EOSS record was obtained rather than generated by this study, summarised the methodology used by the programme to derive SLA and supported it with relevant citations in Section 2.3. We also substantially reworked the section to stress the outcome of the EOSS programme that led to the "climatic unit" theory that our study is testing.

*P6, Line 21: Sentinel was used to "approximate" the timing and elevation of SLA. A great deal of effort should be spent on what approximate means in this instance.*

**Response:** We considerably reworked this paragraph to clarify how we used additional Sentinel-2 images to observe the evolution of the glacier surface and assess its consistency with relative variations of EOSS and albedo in those years. We also revised Section 4.6 to stress the consistency of these observations with the measured evolution of the minimum surface albedo, and related those variations to documented heatwaves that affected New Zealand glaciers.

*P8, Line 30: why was 50% used? How was this value determined? Should 100% be used if you want to remove debris-covered glaciers?*

**Response:** We agree that in theory 100% snow/ice cover should be used. This however assumes that the spectral unmixing is perfect. This is not the case and relying only on pixels truly achieving 100% snow and ice membership from unmixing excludes too many to retrieve a useful albedo signal. We use the 50% threshold because it is the one generally used for binary discrimination of snow/ice pixels (see Hall et al. 2002; Hall & Riggs, 2007; Sirguey et al., 2009; Masson et al., 2018), while being conservative enough to preserve a sufficient amount of pixels. It is true that it may include mixed pixels into the signal and it is the reason why we tested this approach with the conservative masking technique (M2). We did report and discuss in Section 4.1 that the albedo signal with the objective masking (with 50% threshold) was slightly darker on average by 1.2%, likely due to some mixing in the debris-covered pixels. The differences between both approaches however demonstrated sufficient agreement to accept this choice. We added justification for the choice of this threshold and added a citation in support.

Masson, T., Dumont, M., Mura, M., Sirguey, P., Gascoin, S., Dedieu, J.-P. & Chanussot, J. (2018). An Assessment of Existing Methodologies to Retrieve Snow Cover Fraction from MODIS Data. Remote Sensing 10 (4), 619. (Doi: 10.3390/rs10040619.)

Hall, D. K. & Riggs, G. A. (2007). Accuracy assessment of the MODIS snow products. Hydrological Processes 21 (12), 1534--1547.

Hall, D. K., Riggs, G., Salomonson, V. V., DiGirolamo, N. E. & Bayr, K. J. (2002). MODIS snow-cover products. Remote Sensing of Environment 83, 181--194.

*Page 9, Section 3.3: The requirement of using only cloud free glaciers is very restrictive and substantially reduces the amount of data used in the analysis. Provide statistics on the amount and duration of cloud cover. When there is cloud cover will likely be as important as where there is cloud cover.*

**Response:** Accepted, we included that the cloud-free criteria excluded about 66% of images.

*Pixel is a picture element found on a display screen. Grid cell is a better usage.*

**Response:** We respectfully disagree with the reviewer on such a strict use of "pixel". The use of "pixel" to denote image data is widespread in the literature and remote sensing textbooks. For example, Richards (2013) defines *"We talk about the recorded imagery as* image data*, since it is the primary data source from which we extract usable information. One of the important characteristics of the image data acquired by sensors on aircraft or spacecraft platforms is that it is readily available in digital format. Spatially it is composed of discrete picture elements, or* pixels*. Radiometrically—that is in brightness—it is quantised into discrete levels."*

Richards, J. (2013). Remote Sensing Digital Image Analysis. An Introduction. Springer-Vertag.

*Page 10, L1: Artefact should be replaced by error.*

**Response:** Accepted.

*Section 4.3 should be presented in the Introduction section. Typical values for the different glacier states should be provided. This is an example of the systematic problem with this paper, in that there really aren't that much in terms of reportable results.*

**Response:** As noted earlier we have clarified the atmospheric controls on mass balance in the Introduction, and have removed and changed some of the text in this section to provide clarity. As suggested by the reviewer, we also moved two sections of discussion (5.2.1 & 2) to the results, which now contains 9 sections of results (4.1-4.8a,b). Thus, we are of the view there is no shortage of reportable results.

*Page 11, L5: The classification of glaciers into four groups is accomplished by geographical variables. Without providing climate data, it is a mystery to me how the authors determine the glaciers are not behaving as a single climate unit.*

**Response:** We classified glaciers into three groups, rather than four as suggested. We have responded to an earlier comment to clarify that the "unified climatic unit" refers to the theory proposed by Chinn et al. (2012) in reference to the consistent

response of glaciers inferred from the EOSS record. This is based on the high degree of intra-correlation between EOSS SLA across the Southern Alps, not on the study of climate data. Since the snowline and albedo methods aim to capture a similar glacier response, our study can apply a similar approach to test this hypothesis by assessing the degree of intra-correlation between minimum albedo across the GoEA. The outcome of this test and our conclusion that it challenges the "unified climatic unit" inferred by EOSS, is unrelated to the clustering of glaciers. The classification provides additional insights that help characterise what may control the consistency or contrasts in glacier responses across this area. It does not preclude the existence of local contrasts in climate that ultimately govern glacier mass balance. Our results do raise the question about the extent of local climate variability, which itself may be complicated by the complex terrain, but can hardly be answered without reliable climate data that are not available in this area.

*Page 11, L28: Either the declines are monotonic or they are not.*

**Response:** We respectfully disagree with the reviewer, "*monotonic decreasing*" has mathematical meaning that "*decreasing*" alone has not. A signal can be decreasing over a period albeit punctuated by increases. This is the definition of an increasing or decreasing function. A monotonic decreasing function however shall not exhibit any increase within a period of decrease. We have replaced decline by decrease to better fit the mathematical definition.

*Page 13, L15 (and elsewhere) R^2 values should be presented with significance values.*

**Response:** We added the significance value.

*Page 13, L29-30. What is a maximum summer snow line? When exactly was this observed within the summer period?*

**Response:** We added the word "altitude" and the years when those observations were made. The exact date can be found in the figure caption.

*P14,L10: Citation required.*

**Response:** Done.

*P14,L23: Using only 12 glaciers, which are argued to occupy different climactic regions, can size and elevation be reasonably discounted as predictors of the min albedo and SLA?*

**Response:** Indeed, this is what our results show. We have added the correlation analyses and significance of these as part of an earlier comment, which should help clarify and avoid any further confusion.

*P15,L7: define effective, precisely.*

**Response:** We reworked this sentence to be more factual with reference to supporting citations.

*P15,L23: I wouldn't have characterised the R^2 values found in the Results section as strong to moderate.*

**Response:** To reiterate here, the R2 value between EOSS and $\alpha_{yr}^{min}$ for Vertebrae Col is 0.43, while for Angel glacier it is 0.69. Overall, Class 1 glaciers yield an average R2=0.51, while Class 3 glaciers yield R2=0.36. While we appreciate that there is no unique approach to qualifying the strength of the relationship, Akoglu (2018) compiled the following with all approaches compatible with a qualification of those R2 as indicative of moderate to strong relationship.

Adapted from Table 1 in Akoglu, H. (2018). Users guide to correlation coefficients. Turkish Journal of Emergency Medicine 18 (3), 91--93. (Doi: 10.1016/j.tjem.2018.08.001.)

| Coefficient of Determination | Dancey & Reidy (Psychology) | Quinnipiac University (Politics) | Chan YH (Medicine) |
| --- | --- | --- | --- |
| 1 | Perfect | Perfect | Perfect |

| 0.81 | Strong | Very Strong | Very Strong |
|------|--------|-------------|-------------|
| 0.64 | Strong | Very Strong | Very Strong |
| 0.49 | Strong | Very Strong | Moderate |
| 0.36 | Moderate | Strong | Moderate |
| 0.25 | Moderate | Strong | Fair |
| 0.16 | Moderate | Strong | Fair |
| 0.09 | Weak | Moderate | Fair |
| 0.04 | Weak | Weak | Poor |
| 0.01 | Weak | Negligible | Poor |
| 0 | Zero | None | None |

Dancey C.P., Reidy J. Pearson Education; 2007. Statistics without Maths for Psychology; Chan Y.H. Biostatistics 104: correlational analysis. Singap Med J. 2003;44(12):614–619.

*Section 5.2 is results and should not be presented in the discussion section.*

**Response:** Accepted. We moved these sections to results. In relation to characterising cloud cover frequency, we added a statement in Section 2.2 that explicitly indicates that we re-used the classification of clouds from the albedo processing chain of the complete inventory of MODIS imagery for this purpose. The new discussion section (5.2) maintains key points relevant to the control of clouds on the spatial variability of glacier response.

*P18,L7: "likely having as significant influence" means you don't know if it is significant or not.*

**Response:** We replaced "significant" with "substantial".

*Section 5.3. This section should be expanded. Is the correlations between minimum albedo to SLA related to the amount of cloud cover?*

**Response:** In response to an earlier comment we added a discussion about the limitation of MODIS spatial resolution with respect to the size of glaciers. The section referred to has been moved to results and we have created a new section to address in part the potential role clouds have played on albedo and SLA.

*Figure 12 (left panel) There are obvious difference between MODIS Terra and Aqua related to Class. Why is this? Might this be related to the systematic increase in cloud cover in Aqua vs. Terra (Figure 14)?*

**Response:** We did indicate in Section 2.2 that this comparison used 4 pairs of clear-sky imagery for the purpose of testing the consistency of the albedo retrieval between Aqua and Terra, it is therefore unrelated to cloud cover. We must stress to the reviewer that this point is discussed in detail in Section 4.8.1 (former 5.2.1). This is related to the contrasting shadowing configuration between the morning and afternoon image acquisitions and the difficulty of retrieving an accurate observation of albedo in the shade.

*Citations to consider:*

*Casey, K. A., Polashenski, C. M., Chen, J., & Tedesco, M., 2017. Impact of MODIS sensor calibration updates on Greenland ice sheet surface reflectance and albedo trends. The Cryosphere, 11, 1781–1795.*

*Polashenski, C.M., Dibb, J. E., Flanner, M. G., Chen, J. Y., Courville, Z. R., Lai, A. M., et al., 2015) Neither dust nor black carbon causing apparent albedo decline in Greenland's dry snow zone Implications for MODIS C5 surface reflectance. Geophys. Res. Lett., 42, 9319–9327.*

*Van Tricht, K. et al., 2016. Clouds enhance Greenland ice sheet meltwater runoff. Nat. Commun. 7:10266 doi: 10.1038/ncomms10266.*

*Williamson, S.N., Copland, L., Hik, D.S., 2016. The accuracy of satellite-derived albedo for northern alpine and glaciated land covers. Polar Sci. 10, 262-269.*

*Bahr, D.B., Radic, V., 2012. Significant contribution to total mass from very small glaciers. The Cryosphere, 6, 763-770.*

*Bennartz, R., Shupe, M., Turner, D. et al., 2013. July 2012 Greenland melt extent enhanced by low-level liquid clouds. Nature, 496, 83–86. doi:10.1038/nature12002*

*Zhang, Z., Jiang, L., Liu, L., Sun, Y., Wang, H., 2018. Annual Glacier-Wide Mass Balance (2000-2016) of the Interior Tibetan Plateau Reconstructed from MODIS Albedo Products. Remote Sens., 10, 1031, https://doi.org/10.3390/rs10071031.*

---

## Author Response (AR2)

**Variability of glacier albedo and links to annual mass balance for the Gardens of Eden and Allah, Southern Alps, New Zealand**

Angus J. Dowson[1,2], Pascal Sirguey[2], Nicolas J. Cullen[1]

[1]School of Geography, University of Otago, Dunedin, 9016, New Zealand
[2]National School of Surveying, University of Otago, Dunedin, 9016, New Zealand

*Correspondence to*: Pascal Sirguey (pascal.sirguey@otago.ac.nz)

**Status: Reconsider after major revisions**

Dear Valentina,

Thank you for your email and giving us the opportunity to respond to you directly about MS No.: tc-2020-5; Variability of glacier albedo and links to annual mass balance for the Gardens of Eden and Allah, Southern Alps, New Zealand. We have read the reviewer comments and your decision letter and as requested now provide the following rebuttal letter.

For clarity, we have addressed the reviewer's comments, including the two broader questions that are posed, followed by responses to your comments and further questions. The reviewer or your comments are italised and are responses are in normal font.

*Anonymous Referee #2*

*Most of the suggestions for revision from the first round of review were not acted on.*

This is not true. We have carefully responded to all of the feedback given by both reviewers and in almost all cases actioned the suggested changes, which is documented in our 16-page response. In this document, we also provide additional data analysis and results to clearly demonstrate that our findings are logical and robust. It is clear from this comment the reviewer has not taken the time to read our responses.

*1. The manuscript provides very little in the way of new science. No comparison to actual or modelled mass balance was conducted. A proxy for mass balance was compared to another proxy for mass balance - this can be worked out from first principles.*

We disagree that this manuscript does not provide new science. There is a fundamental question that needs to be answered here. It must be recognised that there is no published information on the behaviour of the glaciers in the Gardens of Eden and Allah, despite them being two of the largest ice fields in New Zealand. Thus, one needs to ask whether we want to provide the glaciological community with new and detailed insights about these glaciers, using a state-of-the-art remote sensing approach, or whether to accept the view of the reviewer that no new science is contained in the manuscript. If we accept the latter, is the glaciological community really better off not having any new insights or information about these glaciers?

We would argue that even documentation of the areal extent of these glaciers in Fig. 1, which is not currently available in any detail, is valuable new knowledge. However, the absolutely crucial contribution of this research is that it is a stepping stone to providing a new pathway to monitor glaciers remotely, and may be far more powerful than the current standard, which is the End of Summer Snowline (EOSS) approach. Importantly, it reveals the limitation of the timing of the EOSS observations and how variability in response at an individual glacier scale can be resolved in much greater detail and more widely using the albedo method. Both approaches are carefully compared to the only available direct observations of mass balance by Sirguey et al. (2016), which does not appear to have been read by the reviewer.

It is not clear to us how first principles can be meaningfully applied to understand glacier behaviour if no observations exist from a glacier of interest, whether those are sourced from actual, modelled or remote sensing data. Why is one approach better than the other? If one applies this logic, the question that begs asking is why do we observe the behaviour of glaciers at all if their changes can always be resolved from "first principles"?

Lastly, we are of the view that comparison is a cornerstone of the scientific method. As such, we believe that comparison between our method and that used by the EOSS programme is valuable to the scientific community and we disagree with the reviewer that there is no merit in doing so.

*2. Most of the studies original shortcomings were not addressed in the revision. No climate data was added to support the idea that the glacier basins are indeed climatically different. No systematic analysis of MODIS error between Aqua and Terra was conducted.*

As noted above, we have made significant improvements to the manuscript by following in detail the feedback from the reviewers. This is articulated clearly in our responses, which include supporting figures, new analysis and a new discussion section (see Section 5.2 Implications of variability of albedo on mass balance). We would like to point out that the reviewers did not ask for climate data to be added as part of the revisions, but importantly we have refined our discussion to ensure readers understand how the hypothesis of the Southern Alps behaving as a "climate unit" has evolved from the EOSS programme, and whether this is legitimate or not. It is this point that is central to our manuscript, not the climatic assessment of individual glacier basins.

*No systematic analysis of MODIS error between Aqua and Terra was conducted.*

We do provide careful details about the differences between Terra and Aqua (L182-190), and it is not clear why any further systematic analysis of error between these two MODIS sensors would be necessary given that this aspect of the research is not the focus of the manuscript. It is stated in the manuscript that the use of Aqua was carefully explored to see whether it could be used to "fill in" the Terra time series but as we demonstrate, it does not add much additional benefit. This is an important result as few have explored this. However, the approach did allow us to document the spatial and temporal variability of cloud cover, which is novel and has value - this is why we report on these initial findings. Our question to the reviewer would be how would the proposed systematic analysis of MODIS error improve our current methodology – what new insights would such an analysis bring or more importantly, how has our current approach failed readers?

*Please consider ways to revise your manuscript so that the issues above are at least discussed in detail in the manuscript, if it is not possible to incorporate them in your analysis. For example, in (1) what was the reason for not implementing a simple mass balance model (using a degree-day model for melting, or a simple energy balance model), with reanalysis data as forcing,*

*to get an estimate of glacier mass balance? What concrete evidence do you present for the links between the albedo changes and glacier mass balance (instead of changes in summer snow line or similar proxies)? In (2), what has prevented you from performing a systematic error analysis? Could more sensitivity tests (e.g. assumptions of random errors, or using Monte Carlo methods) be conducted to get a better estimate of uncertainties? If not, why not? Could you use any reanalysis data (e.g. ERA5 - available on 30 km resolution from 1979 to present day) to support the idea that glacier basins are climatically different? If not, why not?*

We have made substantial revisions to the manuscript and we would like our comments to the reviewers, as well as the manuscript to be assessed before making any additional changes. The former was clearly not done by the reviewer, which we believe is a necessary step before proceeding. As you are aware, there are considerable uncertainties that are introduced when modelling glacier response to climate using a simple mass balance model. These glaciers are located in a very data sparse region – there are no in situ meteorological observations, and it is highly questionable whether reanalysis data could meaningfully resolve individual glacier response in the Gardens of Eden and Allah. That is what makes using remote sensing so attractive as it provides key insights that would otherwise not be attainable.

The concrete evidence of the links between the albedo method, EOSS and mass balance is extensively documented in the literature, including glaciers in New Zealand (e.g. Sirguey et al., 2016; Rabatel et al., 2017). Importantly, the method that we use has been very carefully tested on the only glacier in the Southern Alps that has direct mass balance observations (Sirguey et al., 2016), so as we argue in the current manuscript, the next logical step is to use it on glaciers that we don't know anything about to provide new insights. We are offering the glaciological community an alternative to the EOSS approach, which clearly has some limitations. We believe there is merit in exploring a new approach to asess glacier behavioiur remotely in the Southern Alps, and to carefully document for the first time the behaviour of two of the largest ice fields in New Zealand. As clearly stated, these glaciers are located in a Wilderness Area with very limited access. We are of the view there is no better way to do this than what we offer in this manuscript – careful remote sensing observations that provide a platform to monitor glacier behaviour more widely across the Southern Alps (L691-694). It will make much more sense to use reanalysis data and a simple mass balance model once we shift to a large-scale approach of detecting glacier response more broadly across the Southern Alps.

**Variability of glacier albedo and links to annual mass balance for the Gardens of Eden and Allah, Southern Alps, New Zealand**

Angus J. Dowson[1,2], Pascal Sirguey[2], Nicolas J. Cullen[1]

[1]School of Geography, University of Otago, Dunedin, 9016, New Zealand
[2]National School of Surveying, University of Otago, Dunedin, 9016, New Zealand

*Correspondence to*: Pascal Sirguey (pascal.sirguey@otago.ac.nz)

**Status: final response**

**RC1: Interactive comment on "Variability of glacier albedo and links to annual mass balance for the Gardens of Eden and Allah, Southern Alps, New Zealand" by Angus J. Dowson et al.", Anonymous Referee #1, 30 Jan 2020**

*This is a welcome contribution that provides details of an alternative remotely sensed method for monitoring glacier albedo as a potential mass balance proxy. Direct measurement of mass balance on mountain glaciers is resource intensive and often only provides a small number of point data that still require interpolation. Improving remote monitoring methods is essential as this will enable a more comprehensive and sustainable approach to mass balance monitoring. This is a key rationale present for this project. Increased use of remote sensing techniques is a key way scientists can reduce the carbon footprint of their climate-science. Research that progresses such techniques is timely.*

**Response:** We thank the anonymous reviewer for his/her very positive review of our manuscript.

*However, despite the authors making a strong case for the benefit of remote sensing over on-site measurement it is noted in the acknowledgments (and in a media campaign) that the authors did undertake field work at this remote, protected site. The strength of this method will lie in it being able to be robustly applied to any glacier without the need for onsite calibration. So some questions arise;*

**Response:** It is true that the team completed a 2-day fieldwork trip on the study site at the end of January 2018. The data collected on this trip were not directly linked to the study of albedo presented in this paper. The paper did not suggest otherwise but, for the sake of clarification, we provide an answer to each of the referee's specific questions below:

1. *What data was gathered at the site and was this used to tune processing?*

   **Response:** As explained in the manuscript, the only data originating from the field trip that were used to inform this study were oblique aerial and terrestrial photographs that we found useful to refine glacier outlines. At the early stages of this study, glacier outlines were derived from Sentinel-2 imagery until we were fortunate enough to receive very high-resolution Pleiades imagery. Based on this, we decided to refine some glacier outlines and used photos captured during the trip to help inform this process. No data collected during the fieldwork were used to calibrate or validate the processing of MODIS albedo data.

2. *Could the method calibration have been done at an already high-use glacier site (e.g. Franz, Fox, Tasman Glaciers), thereby providing more support for using RS at sensitive sites,*

   **Response:** The albedo retrieval method presented in this study does not need field observations for calibration. The albedo retrieval implemented in Modimlab has already been validated on multiple sites around the world, including on New Zealand glaciers, as can be found in the bibliographic references provided in the manuscript. Given the principle by which the albedo is retrieved by Modimlab and its previous validation in a New Zealand glacier environment, it did not require calibration for this study. Therefore, our method can be applied to other high-use and/or sensitive glaciers sites.

*and 3. What confidence do the authors have that this method can be applied to sensitive, protected sites without the need for onsite measurement?*

**Response:** As noted above, this study did not require site-specific calibration, and no onsite measurements were used other than some additional photos to help refine a number of specific glacier outlines. The photos and on-site field observations were of course very helpful in our efforts to refine the glacier outlines from a quality standpoint. Nonetheless, they were not essential to the method per say. Given the relatively coarse resolution of MODIS data, the interpretation of glacier outlines from high to very high resolution imagery remains sufficient to derive the albedo series. Importantly, there is no significant improvement on the albedo retrieval solely arising from the use of the photos. Given the new insight this study reveals about a poorly documented protected site, we are now much more confident that this method can be applied more widely. This confidence is matched by a reviewer of one of the validation studies of this method, who noted this "*powerful observation-based approach begs to be applied elsewhere*" (Interactive comment on The Cryosphere Discuss., doi:10.5194/tc-2016-98, 2016).

Please note that as we addressed this comment we noted a mistake in the reported date of the fieldwork in the original manuscript, which has been corrected (27-28 January 2018 instead of 20-21 January 2018.)

*Generally the paper is well written and contains sufficient background information and detailed (RS) methods section. However, the interpretation of results is compromised by inadequate information (Figure 1) of the actual glaciers used in analysis. Figure 1, the location map, does not make clear the locations of all the individual glaciers contained in the text. This becomes rather frustrating when reading results and attempting to consider them in a spatial context. Consultation of the official topographic map for the area provided some assistance, but it was still unclear exactly what ice bodies the authors were referring to for the two unnamed glaciers, and for a glacier like Colin Campbell, which has multiple branches, it is not clear what branch has been used for analysis.*

**Response:** We thank the reviewer for this constructive comment. We agree that Figure 1 is a good place to show the names of each glacier mentioned in the study. We have modified Figure 1 and added a panel showing the glacier outlines and names, and also display the cluster to which they belong.

*The decision to separate the 12 glaciers into 3 classes would benefit from a little more explanation. For example Eve Glacier appears more topographically similar to Abel and Colin Campbell (when one makes some assumptions about which branch has been used for the Colin Campbell).*

**Response:** We have clarified how we determine the 3 glacier classes in Section 4.2, which is based on the interpretation of glacier characteristics from clusters identified in scatter plots. Those scatter plots are reproduced below for the reviewer and other readers (class 1: green; class 2: red; class 3: blue). While such a classification involves a trade-off between generalisation and retaining some formal structure, we believe there is enough evidence to justify having three classes. The updated Figure 1 and scatter plots below also provide clear evidence that the class of Eve Glacier is appropriate.

[Figure]

*It is also unclear whether statistics for elevation and slope include the upper accumulation zones or just focus on the more defined glacier trunks.*

**Response:** We have clarified that the statistics for the features used to support the classification are "derived from the mapped outlines of each glacier".

*Having all 12 glaciers clearly defined on a map would benefit result interpretation.*

**Response:** We modified Figure 1 to display glacier outlines and names as well as clusters.

*Figure 5 is a key Figure, but I found myself looking for a third panel showing the average albedo over time, which could potentially be added to Figure 5 (right). If the timing of the ablation minimum is getting later, does this mean that the minimum albedo is also decreasing due to a longer alation season? Or Not?*

**Response:** We thank the reviewer for this constructive comment. We added a panel showing the evolution of the minimum glacier-wide minimum albedo alongside the evolution of its timing. As we revised this figure, we decided to also display yearly values to help visualise the variability better, while maintaining the 3-year rolling average as a dotted line. Furthermore, we added a scatter plot that shows the relationship between $\alpha_{yr}^{min}$ and its timing and assessed its correlation to inform the hypothesis of the reviewer. We find a weak correlation when all clusters are considered together ($R^2$=0.26, p<0.001) and an even weaker correlation when all glaciers are considered separately ($R^2$=0.11, p<0.001). However, within an individual cluster, the value of $\alpha_{yr}^{min}$ does not exhibit a significant correlation with its timing (cluster 1: $R^2$=0.2, p=0.058; cluster 2: $R^2$=0.08, p=0.248; cluster 3: $R^2$=0.03, p=0.472). We have added some further insights in Section 4.4 to report on this new result, updated references to this figure in the text where appropriate, and added a glaciological interpretation and significance of these findings in a new discussion section (5.2 Implications of variability of albedo on mass balance).

*As noted above, the lack of a detailed location map hinders spatial thinking, as does the organisation of Figure 6. While clearly the authors have opted to organise both graphs in Figure 6 by the scale of the x-axis values in doing so the reader is left with no clues as to how these glaciers are actually related in space, again is there a spatial influence on the data presented? It is very difficult to compare results of the left and right graph for an individual glacier as the order of the y-axis (by giving priority to the x-axis value) are different for each graph. While it is appreciated that a 'progressional' x-axis approach might be the 'neatest' presentation, something is lost in regards to the actually physical process or characteristics that might be driving the patterns being presented. For example can something more be said about W/E or N/S trends? If one colour-codes the class sizes some patterns appear, for example class 1 glaciers tend to have lower albedo and a later minimum timing, whereas class 2 (n=2, should potentially include Eve) have higher albedo and earlier minimum timing.*

**Response:** Thanks for the constructive comment. We found the suggestion of colour coding glaciers by class in Figure 6 very helpful and well-thought out. Section 4.4 in the original manuscript did characterize the contrast in magnitude and timing of minimum albedo between glaciers of class 1 and 2 that now becomes more obvious with the colour coding. The reference to Fig. 6 has also been added to the point made in Section 4.4. As noted above, we have added a new section in the discussion that adds to the interpretation of our results by assessing how topography modulates the response and possible fate of the GoEA glaciers.

*While it is appreciated that this paper is 'methods' focused there is missed opportunity to engage more fully with some of the glaciological findings.*

**Response:** Our paper makes use of the albedo method to produce a comprehensive set of new glaciological observations in an area where very little data exist. The principles of this method are described for completeness, yet this section remains limited to echoing the body of literature where it was developed and validated. In particular, Sirguey et al. (2016) provides a very useful '*methods focused paper*' that is in part focused on describing, calibrating and validating the albedo method.

In contrast, this study is very much an application of the method rather than a paper about the method. It provides a large amount of new glaciological observations unavailable to date, and characterizes glacier behaviour of largely undocumented icefields. It is true that this new application of the albedo method provides an opportunity to incrementally refine it, for example by testing the use of glacier outlines to facilitate masking and the opportunity to use MODIS Aqua data. Nevertheless, results associated with these "methodological" steps only involve three out of the now eleven sections of the results and discussion. All other results present new insights about the behaviour of glaciers in the study area. This paper also provides a discussion of the implications of the new findings regarding the variability of the glacier surface compared to results from the EOSS methodology traditionally used in New Zealand. We have also explored the validity of the "unified climatic unit" theory about the response of New Zealand glaciers. In response to comments from both reviewers, we have added a new discussion section (5.2 Implications of variability of albedo on mass balance) that we hope builds knowledge about the glaciology of this remote but important region in the Southern Alps.

*In particular, the finding that the timing of the minimum albedo is occurring later in the summer, which could signal a later onset of the first winter snowfall (i.e. lengthening of the ablation season). This result also makes one wonder if there is any*

*trend (across all the glaciers measured) of a decreasing minimum albedo over time, for if the ablation season is becoming more protracted then the snow surface would likely become more discoloured. Or alternatively, is the minimum albedo the same and the trend is associated with a later start to the ablation season (i.e. more spring snow delaying the onset of melting). It would be great to see a little more discussion of these important mass balance feedbacks.*

**Response:** We thank the reviewer for this input. We found the suggestion very useful as a prompt for us to explore this issue more deeply as our data challenges this hypothesis. As demonstrated above, the comparison between the magnitude of $\underline{\alpha}_{yr}^{min}$ and its timing shows that a delayed minimum albedo does not necessarily lead to a lower albedo. We have added elements of our interpretation of this result in the new discussion section.

*Should the authors wish to include a reference for the Rolleston Glacier mass balance programme, they could cite Purdie, H., Rack, W., Anderson, B., Kerr, T., Chinn, T.J., Owens, I. and Linton, M. 2015: The impact of extreme summer melt on net accumulation of an avalanche fed glacier, as determined by ground-penetrating radar. Geografiska Annaler, Series A: Physical Geography 97, 779-791.*

**Response:** We thank the reviewer for this suggestion and have added the citation to support the statement made in the introduction about the Rolleston Glacier mass balance programme.

**RC2: 'Interactive comment on "Variability of glacier albedo and links to annual mass balance for the Gardens of Eden and Allah, Southern Alps, New Zealand" by Angus J. Dowson et al.', Anonymous Referee #2, 05 February 2020**

*The paper proposes to use minimum MODIS albedo to approximate snow line altitudes (SLA). The paper requires a substantial effort to improve the clarity and organisation. The authors need to clearly define the objectives of the paper then use the results to support the objectives.*

**Response:** We thank the reviewer for his/her comprehensive review of our manuscript.

*General comments: A clearly stated set of objectives is require. The authors need to put some significant thought into this as no surface mass balance (smb) records are presented. The authors need to find a way to make a better connection to smb, otherwise the analysis compares a proxy for smb to another proxy for smb.\*

**Response:** We have refined the aim and provided three specific objectives to meet this aim. We have clarified how this research provides another important step towards developing a robust and comprehensive approach to remotely monitoring glaciers in the Southern Alps. We have clarified in the introduction and elsewhere in the manuscript how the albedo method can be used, even when not calibrated, to assess the spatial and temporal variability of glacier mass balance. Finally, in the context of having limited direct measurements of mass balance in the Southern Alps, the albedo method provides the opportunity of having a second proxy to supplement the 40-year record of the EOSS programme. There is merit in making this comparison as it produces new insights and enabled us to revisit an important conclusion inferred from the EOSS programme.

*The manuscript requires substantial editing for clarity and organisation.*

**Response:** We have made major revisions of the manuscript. The changes we have made, as suggested by the reviewers, have improved the clarity and organisation of the manuscript as detailed in our response to each comment and the track-changes version of the new manuscript.

*The error analysis of MODIS albedo and the potential for sensor degradation (post 2016 on the Terra bus) need to be addressed.*

**Response:** The reviewer reiterated this point in another specific comment – please see our full response below.

*Please use continuous line numbers and do not restart the line numbers at every page.*

**Response:** Done.

*A series of paper citations are listed at the end of this review for you to consider.*

**Response:** We thank the reviewer for those suggestions. We used some of these citations in the revised manuscript as indicated in our response to the reviewer's comments.

*Please remove the words "important", "meaningful", "arguably", etc. from the manuscript. Let the readers decide for themselves.*

**Response:** Accepted, we have removed or replaced all instances of the words.

*Julian Day should probably be referred to as Day of Year.*

**Response:** We appreciate that Julian Day is initially defined as a datation system with an origin on 1 January 4713 BC. However, it is generally accepted that in context, a Julian Day is often understood and unambiguously defined as the number of days since an alternative origin. For example, CNES uses 1 January 1950, while NASA uses 24 May 1968 to define their Julian *date*. In fact, without historical context, Julian Day is often used to refer to the *day of year* number as we use it here. NASA provides a Julian Day Calendar that can be found at https://landweb.modaps.eosdis.nasa.gov/browse/calendar.html. In

view of this, we don't think that our use of "Julian Day" creates an ambiguity that warrants a change as suggested by the reviewer.

*A substantial amount of effort is required to understand the errors in SLA, and minimum albedo related to their temporal mismatch. The figures are not used to full effect in the text. A clearer description and analysis of the figures in the text is required.*

**Response:** We clarified the methodology used by the EOSS program to derive SLA and expanded Section 4.6 with reference to Figures 8, 9 & 10 to stress this point, as well as providing a new discussion section that reflects on these differences. We hope these substantial revisions have provided more clarity.

*The word significant should be reserved to describe statistical significance, otherwise words like substantial should be used to describe a large change.*

**Response:** We agree with the reviewer about the use of "significant". We replaced the word "significant" if it was not being used to describe, or supported by, a statistical test, except when the word was unambiguously referring to its alternate meaning as *"indicative of something sufficiently great or important to be worthy of attention"*.

*If the authors are going to invoke the 50 EOSS glaciers, then a much stronger analysis of how the 12 glaciers in this study compare to the 50 is required.*

**Response:** We have updated and provided a detailed comparison between the response of the glaciers on the GoEA and the EOSS observations from the same region. See Sections 4.6-7 and the discussion in Section 5.1 for these analyses.

*Specific comments:*

*Line 10: Using the whole glacier minimum albedo the dynamics of the glacier are somewhat smoothed over. Perhaps define what is meant by dynamics in this instance. Is it annual minimum albedo that was used, or melt season?*

**Response:** We thank the reviewer for this input. The use of the word *dynamics* was ambiguous as it suggests mechanical behaviour. We implied a more general definition of *dynamics,* namely "*a characteristic in relation to the constant change of a process or a system*". This has been removed as part of our reworking of the abstract.

*Line 13: briefly define new approach*

**Response:** We believe that describing the approach we tested and used for masking would be unnecessarily complicated and lengthen the abstract. We decided to remove this part of the sentence.

*Line 14: provide the evidence in brief*

**Response:** We rephrased the sentence to clarify this.

*Line 12: Was surface mass balance measured, or was EOSS SLA measured – the two are not the same thing?*

**Response:** We have refined and updated the abstract. Surface mass balance was not measured in this study. This sentence referred to the fact that the albedo method used in this study has been shown in earlier work to be an efficient proxy for glacier mass balance. The question raised by the reviewer is legitimate as the sentence may have been misleading given we are not in a position to provide a quantitative estimate, as there has not been a calibration of mass balance in this study.

*Line 16: Define "high snow line"*

**Response:** We added that "high snow line altitude" is defined as "relative to the long-term average"

*p-value should be p<0.001, etc. depending on level of significance.*

**Response:** Accepted, we expressed the actual p value unless p<0.001, in which case we only used the statement of inequality.

*The results section of the Abstract need to be rewritten to present all of the results.*

**Response:** We have refined and updated the abstract.

*Furthermore, a statement of why you have done this analysis is required.*

**Response:** This comment echoes the need for clearer objectives that was highlighted in the first general comment. We have modified the abstract and other parts of the manuscript to ensure the aim and objectives of this research are more explicit.

*Lines 26-27: Define climate units.*

**Response:** We are referring to the theory of the Southern Alps glaciers behaving as a "*unified climatic unit*" as proposed by Chinn et al. (2012). This is one of the main outcomes of the EOSS monitoring programme inferred from the high degree of intra-correlation found in the variability of snowline elevation between any index glacier and the average of the group. In the introduction where the term "*unified climatic unit*" is directly quoted from Chinn et al. (2012), we have clarified that this is referring to a "consistent response to climate variability inferred from the EOSS record". We have also reworded the abstract to clarify the origins and meaning of this term.

*Introduction:*

*P2,Line 17: The minimum albedo method does not infer mass balance. It scales to ELA and AAR, which in turn scale to mass balance. Without measurements of mass balance to quantify the relationship to minimum albedo, ELA, or AAR, mass balance should not really enter into the discussion.*

**Response:** We agree that the word "infer" was not appropriate in this sentence. We replaced it by "estimate". We however respectfully disagree with the reviewer that mass balance should not enter into the discussion. This paragraph is introducing the general concepts of remote sensing approaches for the very purpose of capturing glacier mass balance, whether its relative variability, or its absolute value when calibration data can be provided. The statement made remains factually correct in view of the supporting literature provided in the paragraph.

*P2,Line 20: Define "relationships".*

**Response:** We have rephrased this sentence as "The annual minimum glacier-wide albedo ($\alpha_{yr}^{min}$) retrieved from MODIS imagery has been found to scale to annual glacier mass balance…"

*P2,Line 20-23: There are more citations for this method – see below.*

**Response:** We thank the reviewer for these suggestions. Among those, we find that only Zhang et al. (2018) is directly relevant and added it to the Introduction section.

*P2,Line 25: why is the glacier contribution "globally significant" and define (i.e., X m sea level increase).*

**Response:** We must stress to the reviewer that the meaning and definition of *significant* is not unique and solely reserved to characterise a quantitative value subject to statistical significance testing. In context, *significant* means "*Sufficiently great or important to be worthy of attention; noteworthy*" (Oxford Dictionary). As such, New Zealand is invariably represented in global studies as its own glacier region. In view of this we maintain that our statement that New Zealand Glaciers "*are regarded as globally significant*" is factually correct and has adequate context to suggest the meaning of significant in this sentence as "*noteworthy*". The alternative would be that New Zealand glaciers would be otherwise ignored in such global studies, which is clearly not the case. We respectfully disagree with the reviewer that using *significant* in this context necessarily calls for a specific quantified statement.

*P2,Line 29: Air temperature is important in controlling glacier mass balance – but on line 19 you said shortwave radiation played a governing role on smb. Which is it? Both I gather, so this section should be expanded to detail the nuanced nature of the controls on glacier smb.*

**Response:** We have changed the wording in this paragraph to ensure readers understand the relative importance of solar radiation, air temperature and precipitation. Our review of these governing processes are consistent to those that are well established in the literature.

*P2,Line 31: define cloud properties and how they influence smb. What is the role of longwave forcing?*

**Response:** We have provided a benchmark reference for readers and return to the importance of clouds in the results and discussion.

*P2,Line 32: "global warming in the Southern Alps" is nonsensical.*

**Response:** We have changed the wording of this sentence. However, we respectfully disagree with the reviewer about this comment. In this paragraph we clearly provide context in reference to the findings of Mackintosh et al. (2017). The title and content of the latter is unambiguous about the regional impacts of a "*period of global warming*" on New Zealand. The abstract itself reads "*The exceptional terminus advance of some glaciers during recent global warming is thought to relate to locally specific climate conditions, such as increased precipitation.*" As such, we maintain that our statement is factually correct in reference to the point being made to the supporting literature.

*P3,Line 1: define "fast-responding glaciers" what does this mean and why is it important?*

**Response:** This is again a context statement that we believe unambiguously paraphrases Mackintosh et al. (2017) where the reviewer can read "*This combination of large mass balance gradients and steep, relatively thin and fast-moving ice makes them adjust rapidly to climate forcing. Very few glaciers on Earth are capable of responding this quickly.*"

*P3,Line 3: Why was minimum albedo method not used on these two glaciers? If that anlaysis has already been conducted, then provide an analysis of ELA, AAR and minimum albedo for these two glaciers in reference to smb. Once completed then the analysis presented on the other glaciers without smb will be on a better footing.*

**Response:** We provide information two paragraphs earlier that the albedo method has been used and related to glacier mass balance in the Southern Alps, with citations to relevant work by Sirguey et al. (2016) and Rabatel et al. (2017). In order to further address this, comment, we have specifically named the glaciers on which the albedo method was used, namely Brewster Glacier (direct calibration with in-situ mass balance) and Park Pass Glacier (indirect assessment via correlation to EOSS record, itself scaling to mass balance) to ensure readers are aware that this method has been used on glaciers in New Zealand. We hope readers who require further information will consult the cited literature. We have also provided further details about how the albedo method was calibrated using in situ mass balance data from Brewster Glacier. The albedo method has not been used on Rolleston Glacier as it is too small to be captured by MODIS imagery (0.11 km$^2$ in 2012), with the area and reasoning provided.

*P3,Line 4: Define what was measured then tell me why it is important.*

**Response:** We reworked this paragraph to state specifically that the mass balance campaign on Ivory Glacier in the 1970s was the first comprehensive glaciological mass balance study conducted in New Zealand's Southern Alps. We removed the word "important" as part of addressing an earlier comment from the reviewer.

*P3,Line 10: There is going to be a large problem using MODIS grids on very small glaciers because of the grid size to glacier area mismatch - discuss.*

**Response:** The reviewer is right about this limitation to using MODIS for the albedo method. We did stress this point in Section 4.7 of the initial manuscript with a supporting citation. Noting that albedo is mapped with MODImLab at the improved 250m resolution, Davaze et al. (2018) show that $\alpha_{yr}^{min}$ could be successfully retrieved for glaciers down to 0.5 km$^2$ (7 pixels) with good correlation to annual mass balance. The agreement we obtained between $\alpha_{yr}^{min}$ and EOSS using the 0.7 km$^2$ Vertebrae Col 25 is consistent with this assessment. In response to an earlier comment, we emphasised this limitation when introducing

Rolleston Glacier. In order to address the reviewer's comment further, we also added a discussion about this limitation in Section 5.3.

*P3,Line 11: have meant – means*

**Response:** We have modified the sentence.

*P3,Line 13-21: this sounds like the missing research question that should have been addressed in the end of the Abstract.*

**Response:** This sentence does provide in part the background and motivation for the aim and three objectives that we have refined as part of our revision of the manuscript.

*P3,Line 22-27: Provide better links to the tables and figures detailing the study site. The last paragraph of the Introduction is redundant and should be removed.*

**Response:** We added a reference to Figure 1. We accepted the suggestion of the reviewer to remove the last paragraph of the introduction.

*Page 4: Site description: P4,Line 20: Provide climate normals.*

**Response:** Accepted. Marcara (2017) estimated a long-term mean annual rainfall surface for New Zealand for the period 1972-2016 from which we retrieved a range for precipitation over the area. We used normal temperature at surrounding weather stations over the period 1980-2010 and a lapse rate of -5.7°C km$^{-1}$ determined from the data to interpolate a mean normal temperature surface over the area, and a range over the glaciated region. We stress however that this estimate remains only indicative given the distance of the GoEA from established weather stations (closest weather station on the West Coast is 22km away, while the nearest to the east is >60km) and the uncertainty associated with the interpolation.

*P5,Line 3: How is this a sensitive climate situation? Sensitivity means that there is a glacier response (dynamic, or mass loss etc.) for temperature or precipitation change. A case for neither of these scenarios has been presented.*

**Response:** We have removed reference to the "sensitive climate situation" to avoid confusion.

*P5,Line 10: How do you know that the 14 March corresponds to the minimum ELA?*

**Response:** The point the reviewer is raising is not clear, nor what she/he means by "minimum ELA" in this context. We understand ELA stands for Equilibrium Line Altitude, which has meaning in surface mass balance. Braithwaite & Raper (2009) define "*minimum ELA* values corresponding to balance years with *highly (...) positive mass balances*". In view of this, we believe the reviewer's use of "minimum ELA" seems misplaced and makes a connection between ELA and glacier outlines that is irrelevant in our context. Nonetheless, we assume the reviewer is trying to say that glacier outlines should be derived from imagery timed precisely at the maximum ablation. If so, we dispute this strict requirement because glacier outlines are not a direct consequence of the surface mass balance for that year due to glacier response time. In practice, one aims for late summer imagery mostly because it corresponds to the melting of as much transient snow as possible to facilitate interpretation of glacier boundaries. At the same time, tradeoffs must be made given the availability of imagery, and mapping glacier outlines remains often heavily reliant on photo interpretation and experience. We indicated that we used the NDSI from the S2 image on 14 March as a base to derive glacier outlines. We then refined them manually (e.g., NDSI cannot map debris-covered parts and transient snow needs to be removed) using the additional data mentioned in the text. We believe this is a very common and valid approach, yet it is not within the scope of this manuscript to discuss this in more detail. For the reviewer's information, the Sentinel-2 image from 30 April 2016 shown in Figure 9 in the original manuscript would be closer to the end of ablation and the latest before winter snowfall that year. On the one hand it exhibits less residual snow, which may help. On the other hand, it can be seen in the comparison below, how topographic shading affects the image on 30 April compared to 14 March. Mapping solely from this image is not making the task easier nor more accurate. The NDSI is sensitive to shadow with high values in the shade greatly complicating the snow/ice segmentation. We maintain that the mid-March image is a better compromise despite the remaining snow patches, to derive suitable glacier outlines for this study. The consistency between snow patches with the late April image and across a few years with other Sentinel images (e.g., those shown in Fig. 9) and Pleiades 2017 images help discriminate them from glaciers at the refinement stage.

Braithwaite, R. & Raper, S. (2009). Estimating equilibrium-line altitude (ELA) from glacier inventory data. Annals of Glaciology 50 (53), 127--132. (Doi: 10.3189/172756410790595930.)

[Figure]

| 14 March 2016 | 30 April 2016 |
|---|---|
| False colour infrared (bands 8-4-3) | False colour infrared (bands 8-4-3) |
| NDSI (B3-B11)/(B3+B11) stretched between 0 (black) and 1 (white) | NDSI (B3-B11)/(B3+B11) stretched between 0 (black) and 1 (white) |

*P5,Line 12: What is "field knowledge"? Actually the whole section should be rewritten. What are the NDSI wavelengths used?*

**Response:** We have modified this section and specified the wavelength used for the NDSI. As explained, we took advantage of a two-day field trip to take close-up aerial photos of the glacier. This provided us with valuable direct evidence of this remote area that supplemented our interpretation of satellite imagery used to refine some glacier outlines. To avoid any confusion, we removed "field knowledge" from the revised manuscript.

*There is almost two months difference between image capture dates. How is this reconciled?*

**Response:** We have reworked this section to ensure there is no confusion for readers, and we hope this addresses the reviewer's concerns about our approach to mapping glacier outlines in this study. To respond to the specific question, we acknowledge that we did not specifically mention in the manuscript that we used the 29 April 2016 image to refine our mapping, which we understand the reviewer is referring rather than the 14 March 2016 image. It should also be noted that we used 0.5m imagery from Pleiades in March 2017, as well as photos from our field work in 2018. Thus, we compiled images across two years to help interpret glacier outlines from the March 2016 Sentinel-2 imagery. For example, the Pleiades imagery and high resolution surface supported much better interpretation of debris-covered glaciers. It is also important to keep in mind that the glacier boundaries used for this study are derived to extract 250-m resolution pixels from the MODIS albedo map and that glaciers are not expected to retreat hundreds of meters per year in this region (it would need calving in terminal lakes to achieve such rates). The mapping also compounds uncertainties in the order of tens of meters due to the geolocation of Sentinel-2 imagery and the variable interpretation at pixel level. In this regard, we don't believe the two-year period between these sources of information and our approach to mapping are any cause for concern. Further evidence of the benefit of exploiting all of these data are shown below, which provides a sequence of the terminus of Lambert Glacier from 14 March 2016, 30 April 2016 (Sentinel-2), and March 2017 (Pleiades).

[Figure]

[Figure]

[Figure]

| Sentinel-2 14 March 2016 | Sentinel-2 30 April 2016 | Pleiades-1B 10 March 2017 |

*P5,Line 27: What is the superscript T on MODIS? The standard usage is MOD for Terra and MYD for Aqua.*

**Response:** T (resp. A) superscripts indeed stands for MODIS Terra (resp. Aqua). We believed this was self-explanatory. We are aware of the MOD/MYD convention for naming MODIS products, and we did refer to this usage in P6L9 of the original manuscript. In context, when clearly referring to either sensor rather than the products from them, we do not believe that using the acronyms of MOD or MYD are appropriate. We therefore maintain our use of the superscript to refer to either one or the other sensor, and clarified this convention in Section 2.2 to avoid confusion.

*P5, Section 2.2: You should read: https://nsidc.org/data/modis/terra_aqua_differences There is some speculation that the Terra MODIS sensor is degrading. This degradation is largely correct for in the collection 6 data. There are several citations that should be considered in relation to this issue. These citations have been listed at the end of the review.*

**Response:** We thank the reviewer for this reading suggestion. This also echoes the general comment made earlier about "*The error analysis of MODIS albedo and the potential for sensor degradation (post 2016 on the Terra bus) need to be addressed.*"

We have provided further information about MODIS calibration issues in Section 4.8.1. However, we raise two important points about this issue in response to the reviewer's comments.

First, the main point being made on webpage given by the reviewer is about the differences in processing MODIS snow products between Terra and Aqua sensors due to the band 6 failed detectors of Aqua. There is no mention about the degradation of MODIS detectors. We are very aware of the differences in processing MOD10/MTD10 C6 products. However, these are not relevant for our study as we are not relying on MOD10 data. Instead, we complete all processing and albedo retrieval directly from L1B data, as is clearly explained in the manuscript, and in even more detail in the cited literature about our processing technique. As far as we know, the Quantitative Image Restoration (QIR) technique to restore Aqua band 6 is only used in production of MYD10 products but not in the production of the MOD03/MOD02 C6 products we are using. Thus, it does not apply to our data (see MODIS L1B Product User's Guide for L1B Version 6.2.2 (Terra) and Version 6.2.1 (Aqua): https://mcst.gsfc.nasa.gov/sites/default/files/file_attachments/M1054E_PUG_2017_0901_V6.2.2_Terra_V6.2.1_Aqua.pdf).

Please also note that we have acknowledged the failed detector of band 6 and demonstrate that it does not significantly affect our retrieval, which we believe is a valuable outcome of our study. This was expected as band 6 in the SWIR is an absorption band for snow and ice, and therefore contributes only marginally to the albedo signal.

Second, we are aware of the sensor calibration degradation in MODIS C5 data as demonstrated by Lyapustin et al. (2014). The latter concludes very clearly that "*The new C6 calibration approach removes major calibrations trends in MODIS Level 1B data.*". From the citations suggested by the reviewer, Casey et al. (2017) and Polashenski et al. (2015) both point out and document calibration issues with C5 data and conclude that the use of C6 data is preferable, if not necessary. Sayer et al. (2015) also confirmed the better performance of the C6 calibration. The reviewer points out that the degradation is largely corrected for in the C6 data. As we use the C6 data, which are the highest quality data available and do not have any severe calibration issues, we are of the view that further justification for using them is not necessary.

In this context, we shall stress Figure 8 and 10 to the reviewer which show albedo time series. The reviewer can assess a lack of visible trends in winter albedo which we believe should lift concerns about the existence (speculated or not) of any major

calibration issue with C6 affecting our study and results. The alternative hypothesis would be that winter albedo has a trend that is precisely matched by a calibration issue. We do not believe that this hypothesis is likely.

Lyapustin, A., Wang, Y., Xiong, X., Meister, G., Platnick, S., Levy, R., Franz, B., Korkin, S., Hilker, T., Tucker, J., Hall, F., Sellers, P., Wu, A. & Angal, A. (2014). Science impact of MODIS C5 calibration degradation and C6+ improvements. Atmospheric Measurement Techniques 7 (12), 4353--4565. (Doi: 10.5194/amt-7-4353-2014.)

Sayer, A. M., Hsu, N. C., Bettenhausen, C., Jeong, M.-J. & Meister, G. (2015). Effect of MODIS Terra radiometric calibration improvements on Collection 6 Deep Blue aerosol products: Validation and Terra/Aqua consistency. Journal of Geophysical Research: Atmospheres 120 (23). (Doi: 10.1002/2015jd023878.)

*P5, Line 29: orbits are usually described as ascending and descending.*

**Response:** Agreed, we modified the revised manuscript accordingly.

*P6, Line 2: What does "exceptionally cloudy" mean? Provide a description with statistics.*

**Response:** We replaced "exceptionally" by "extraordinary" as described by Wardle (1986), and we reworked the sentence to make the origin of this statement unambiguous and invite the reader to consult the citation for more information. We also agreed with the reviewer and added specific statistics from Wardle (1986) that are specific to two sites surrounding our region of study. It should be noted that Cropp River, west from the GoEA, exhibits the most frequent cloud cover of all sites examined by Wardle (1986).

*Why were only four days used for comparison between Terra and Aqua? Provide justification.*

**Response:** We clarified that in addition to clear sky acquisition on the same day, such comparison needed similar sensor-zenith to minimise the effect of contrasting panoramic distortion on pixel footprint. As our assessment of cloud cover below demonstrates, there are not many opportunities to get clear-sky conditions for both Terra and Aqua. This opportunity is further reduced when adding the limitation of sensor zenith. We also stated the number of samples provided by the four images amounts to 2196 pixel pairs, which we believe is a sufficient sample size to complete this comparison.

*The MODIS methods section does not indicate which albedo product was used, or how albedo was produced at 250 m. A much more clear description of data processing is required. O.k., I see the following section on albedo processing. Perhaps a sentence here to indicate that MODImLab will be detailed (and why) later in the paper.*

**Response:** We added a sentence in the previous paragraph that the albedo retrieval is described in the methods section.

*Section 2.3 requires a much better description of what exactly was done.*

**Response:** This comment suggests to us the reviewer is assuming we completed the mapping of the EOSS snowlines and generated the EOSS SLA records for Vertebrea Col 25 as part of this study. This is not the case. The EOSS programme is a national programme run by the National Institute of Water and Atmospheric Research or NIWA. An annual report with SLA values for each index glacier is normally published sometime after the observations are taken. Appendices about the method used to map SLA on each year are also provided on request. They indicate what method was used to derive SLA for any particular year. We have clarified that the EOSS record was obtained rather than generated by this study, summarised the methodology used by the programme to derive SLA and supported it with relevant citations in Section 2.3. We also substantially reworked the section to stress the outcome of the EOSS programme that led to the "climatic unit" theory that our study is testing.

*P6, Line 21: Sentinel was used to "approximate" the timing and elevation of SLA. A great deal of effort should be spent on what approximate means in this instance.*

**Response:** We considerably reworked this paragraph to clarify how we used additional Sentinel-2 images to observe the evolution of the glacier surface and assess its consistency with relative variations of EOSS and albedo in those years. We also revised Section 4.6 to stress the consistency of these observations with the measured evolution of the minimum surface albedo, and related those variations to documented heatwaves that affected New Zealand glaciers.

*P8, Line 30: why was 50% used? How was this value determined? Should 100% be used if you want to remove debris-covered glaciers?*

**Response:** We agree that in theory 100% snow/ice cover should be used. This however assumes that the spectral unmixing is perfect. This is not the case and relying only on pixels truly achieving 100% snow and ice membership from unmixing excludes too many to retrieve a useful albedo signal. We use the 50% threshold because it is the one generally used for binary discrimination of snow/ice pixels (see Hall et al. 2002; Hall & Riggs, 2007; Sirguey et al., 2009; Masson et al., 2018), while being conservative enough to preserve a sufficient amount of pixels. It is true that it may include mixed pixels into the signal and it is the reason why we tested this approach with the conservative masking technique (M2). We did report and discuss in Section 4.1 that the albedo signal with the objective masking (with 50% threshold) was slightly darker on average by 1.2%, likely due to some mixing in the debris-covered pixels. The differences between both approaches however demonstrated sufficient agreement to accept this choice. We added justification for the choice of this threshold and added a citation in support.

Masson, T., Dumont, M., Mura, M., Sirguey, P., Gascoin, S., Dedieu, J.-P. & Chanussot, J. (2018). An Assessment of Existing Methodologies to Retrieve Snow Cover Fraction from MODIS Data. Remote Sensing 10 (4), 619. (Doi: 10.3390/rs10040619.)

Hall, D. K. & Riggs, G. A. (2007). Accuracy assessment of the MODIS snow products. Hydrological Processes 21 (12), 1534--1547.

Hall, D. K., Riggs, G., Salomonson, V. V., DiGirolamo, N. E. & Bayr, K. J. (2002). MODIS snow-cover products. Remote Sensing of Environment 83, 181--194.

*Page 9, Section 3.3: The requirement of using only cloud free glaciers is very restrictive and substantially reduces the amount of data used in the analysis. Provide statistics on the amount and duration of cloud cover. When there is cloud cover will likely be as important as where there is cloud cover.*

**Response:** Accepted, we included that the cloud-free criteria excluded about 66% of images.

*Pixel is a picture element found on a display screen. Grid cell is a better usage.*

**Response:** We respectfully disagree with the reviewer on such a strict use of "pixel". The use of "pixel" to denote image data is widespread in the literature and remote sensing textbooks. For example, Richards (2013) defines *"We talk about the recorded imagery as* image data*, since it is the primary data source from which we extract usable information. One of the important characteristics of the image data acquired by sensors on aircraft or spacecraft platforms is that it is readily available in digital format. Spatially it is composed of discrete picture elements, or* pixels*. Radiometrically—that is in brightness—it is quantised into discrete levels."*

Richards, J. (2013). Remote Sensing Digital Image Analysis. An Introduction. Springer-Vertag.

*Page 10, L1: Artefact should be replaced by error.*

**Response:** Accepted.

*Section 4.3 should be presented in the Introduction section. Typical values for the different glacier states should be provided. This is an example of the systematic problem with this paper, in that there really aren't that much in terms of reportable results.*

**Response:** As noted earlier we have clarified the atmospheric controls on mass balance in the Introduction, and have removed and changed some of the text in this section to provide clarity. As suggested by the reviewer, we also moved two sections of discussion (5.2.1 & 2) to the results, which now contains 9 sections of results (4.1-4.8a,b). Thus, we are of the view there is no shortage of reportable results.

*Page 11, L5: The classification of glaciers into four groups is accomplished by geographical variables. Without providing climate data, it is a mystery to me how the authors determine the glaciers are not behaving as a single climate unit.*

**Response:** We classified glaciers into three groups, rather than four as suggested. We have responded to an earlier comment to clarify that the "unified climatic unit" refers to the theory proposed by Chinn et al. (2012) in reference to the consistent

response of glaciers inferred from the EOSS record. This is based on the high degree of intra-correlation between EOSS SLA across the Southern Alps, not on the study of climate data. Since the snowline and albedo methods aim to capture a similar glacier response, our study can apply a similar approach to test this hypothesis by assessing the degree of intra-correlation between minimum albedo across the GoEA. The outcome of this test and our conclusion that it challenges the "unified climatic unit" inferred by EOSS, is unrelated to the clustering of glaciers. The classification provides additional insights that help characterise what may control the consistency or contrasts in glacier responses across this area. It does not preclude the existence of local contrasts in climate that ultimately govern glacier mass balance. Our results do raise the question about the extent of local climate variability, which itself may be complicated by the complex terrain, but can hardly be answered without reliable climate data that are not available in this area.

*Page 11, L28: Either the declines are monotonic or they are not.*

**Response:** We respectfully disagree with the reviewer, "*monotonic decreasing*" has mathematical meaning that "*decreasing*" alone has not. A signal can be decreasing over a period albeit punctuated by increases. This is the definition of an increasing or decreasing function. A monotonic decreasing function however shall not exhibit any increase within a period of decrease. We have replaced decline by decrease to better fit the mathematical definition.

*Page 13, L15 (and elsewhere) R^2 values should be presented with significance values.*

**Response:** We added the significance value.

*Page 13, L29-30. What is a maximum summer snow line? When exactly was this observed within the summer period?*

**Response:** We added the word "altitude" and the years when those observations were made. The exact date can be found in the figure caption.

*P14,L10: Citation required.*

**Response:** Done.

*P14,L23: Using only 12 glaciers, which are argued to occupy different climactic regions, can size and elevation be reasonably discounted as predictors of the min albedo and SLA?*

**Response:** Indeed, this is what our results show. We have added the correlation analyses and significance of these as part of an earlier comment, which should help clarify and avoid any further confusion.

*P15,L7: define effective, precisely.*

**Response:** We reworked this sentence to be more factual with reference to supporting citations.

*P15,L23: I wouldn't have characterised the R^2 values found in the Results section as strong to moderate.*

**Response:** To reiterate here, the R2 value between EOSS and $\alpha_{yr}^{min}$ for Vertebrae Col is 0.43, while for Angel glacier it is 0.69. Overall, Class 1 glaciers yield an average R2=0.51, while Class 3 glaciers yield R2=0.36. While we appreciate that there is no unique approach to qualifying the strength of the relationship, Akoglu (2018) compiled the following with all approaches compatible with a qualification of those R2 as indicative of moderate to strong relationship.

Adapted from Table 1 in Akoglu, H. (2018). Users guide to correlation coefficients. Turkish Journal of Emergency Medicine 18 (3), 91--93. (Doi: 10.1016/j.tjem.2018.08.001.)

| Coefficient of Determination | Dancey & Reidy (Psychology) | Quinnipiac University (Politics) | Chan YH (Medicine) |
| --- | --- | --- | --- |
| 1 | Perfect | Perfect | Perfect |

| 0.81 | Strong | Very Strong | Very Strong |
|------|--------|-------------|-------------|
| 0.64 | Strong | Very Strong | Very Strong |
| 0.49 | Strong | Very Strong | Moderate |
| 0.36 | Moderate | Strong | Moderate |
| 0.25 | Moderate | Strong | Fair |
| 0.16 | Moderate | Strong | Fair |
| 0.09 | Weak | Moderate | Fair |
| 0.04 | Weak | Weak | Poor |
| 0.01 | Weak | Negligible | Poor |
| 0 | Zero | None | None |

Dancey C.P., Reidy J. Pearson Education; 2007. Statistics without Maths for Psychology; Chan Y.H. Biostatistics 104: correlational analysis. Singap Med J. 2003;44(12):614–619.

*Section 5.2 is results and should not be presented in the discussion section.*

**Response:** Accepted. We moved these sections to results. In relation to characterising cloud cover frequency, we added a statement in Section 2.2 that explicitly indicates that we re-used the classification of clouds from the albedo processing chain of the complete inventory of MODIS imagery for this purpose. The new discussion section (5.2) maintains key points relevant to the control of clouds on the spatial variability of glacier response.

*P18,L7: "likely having as significant influence" means you don't know if it is significant or not.*

**Response:** We replaced "significant" with "substantial".

*Section 5.3. This section should be expanded. Is the correlations between minimum albedo to SLA related to the amount of cloud cover?*

**Response:** In response to an earlier comment we added a discussion about the limitation of MODIS spatial resolution with respect to the size of glaciers. The section referred to has been moved to results and we have created a new section to address in part the potential role clouds have played on albedo and SLA.

*Figure 12 (left panel) There are obvious difference between MODIS Terra and Aqua related to Class. Why is this? Might this be related to the systematic increase in cloud cover in Aqua vs. Terra (Figure 14)?*

**Response:** We did indicate in Section 2.2 that this comparison used 4 pairs of clear-sky imagery for the purpose of testing the consistency of the albedo retrieval between Aqua and Terra, it is therefore unrelated to cloud cover. We must stress to the reviewer that this point is discussed in detail in Section 4.8.1 (former 5.2.1). This is related to the contrasting shadowing configuration between the morning and afternoon image acquisitions and the difficulty of retrieving an accurate observation of albedo in the shade.

*Citations to consider:*

*Casey, K. A., Polashenski, C. M., Chen, J., & Tedesco, M., 2017. Impact of MODIS sensor calibration updates on Greenland ice sheet surface reflectance and albedo trends. The Cryosphere, 11, 1781–1795.*

*Polashenski, C.M., Dibb, J. E., Flanner, M. G., Chen, J. Y., Courville, Z. R., Lai, A. M., et al., 2015) Neither dust nor black carbon causing apparent albedo decline in Greenland's dry snow zone Implications for MODIS C5 surface reflectance. Geophys. Res. Lett., 42, 9319–9327.*

*Van Tricht, K. et al., 2016. Clouds enhance Greenland ice sheet meltwater runoff. Nat. Commun. 7:10266 doi: 10.1038/ncomms10266.*

Williamson, S.N., Copland, L., Hik, D.S., 2016. The accuracy of satellite-derived albedo for northern alpine and glaciated land covers. Polar Sci. 10, 262-269.

Bahr, D.B., Radic, V., 2012. Significant contribution to total mass from very small glaciers. The Cryosphere, 6, 763-770.

Bennartz, R., Shupe, M., Turner, D. et al., 2013. July 2012 Greenland melt extent enhanced by low-level liquid clouds. Nature, 496, 83–86. doi:10.1038/nature12002

[revised manuscript text omitted]